# The rise and transformation of Bronze Age pastoralists in the Caucasus

Ayshin Ghalichi[1,32✉], Sabine Reinhold[2,32✉], Adam B. Rohrlach[1,3], Alexey A. Kalmykov[4], Ainash Childebayeva[1,5], He Yu[1,6], Franziska Aron[1], Lena Semerau[1], Katrin Bastert-Lamprichs[2], Andrey B. Belinskiy[7], Natalia Y. Berezina[8], Yakov B. Berezin[8], Nasreen Broomandkhoshbacht[9], Alexandra P. Buzhilova[8], Vladimir R. Erlikh[10], Lars Fehren-Schmitz[9,11], Irina Gambashidze[12], Anatoliy R. Kantorovich[13], Konstantin B. Kolesnichenko[7], David Lordkipanidze[14,15], Rabadan G. Magomedov[16], Katharina Malek-Custodis[17], Dirk Mariaschk[2], Vladimir E. Maslov[18], Levon Mkrtchyan[19], Anatoli Nagler[2], Hassan Fazeli Nashli[20], Maria Ochir[21], Yuri Y. Piotrovskiy[22], Mariam Saribekyan[19], Aleksandr G. Sheremetev[23], Thomas Stöllner[24,25], Judith Thomalsky[2], Benik Vardanyan[19,26], Cosimo Posth[1,27,28], Johannes Krause[1,29,30], Christina Warinner[1,29,30,31], Svend Hansen[2] & Wolfgang Haak[1✉]

The Caucasus and surrounding areas, with their rich metal resources, became a crucible of the Bronze Age[1] and the birthplace of the earliest steppe pastoralist societies[2]. Yet, despite this region having a large influence on the subsequent development of Europe and Asia, questions remain regarding its hunter-gatherer past and its formation of expansionist mobile steppe societies[3–5]. Here we present new genome-wide data for 131 individuals from 38 archaeological sites spanning 6,000 years. We find a strong genetic differentiation between populations north and south of the Caucasus mountains during the Mesolithic, with Eastern hunter-gatherer ancestry[4,6] in the north, and a distinct Caucasus hunter-gatherer ancestry[7] with increasing East Anatolian farmer admixture in the south. During the subsequent Eneolithic period, we observe the formation of the characteristic West Eurasian steppe ancestry and heightened interaction between the mountain and steppe regions, facilitated by technological developments of the Maykop cultural complex[8]. By contrast, the peak of pastoralist activities and territorial expansions during the Early and Middle Bronze Age is characterized by long-term genetic stability. The Late Bronze Age marks another period of gene flow from multiple distinct sources that coincides with a decline of steppe cultures, followed by a transformation and absorption of the steppe ancestry into highland populations.

The Caucasus region and surrounding areas lie at the interface of Europe and Asia. By the mid-Holocene, the Greater Caucasus Mountain range functioned as a semipermeable barrier through which ideas, technologies, languages and people moved[1]. The wide variety of climate zones in the topographically complex South Caucasus supported a high level of biodiversity, whereas the mountain highlands and hilly piedmont zones in the North Caucasus transitioned into the flat open grasslands of the West Eurasian steppe belt[9]. With its diverse ecologies and rich metal resources, the Caucasus region became a crucible of the Bronze Age (BA) and the birthplace of the earliest steppe pastoralist societies during the fourth millennium BC (ref. 2). The subsequent continental expansions of these steppe pastoralist groups over the next two millennia ultimately reshaped the genetic make-up, languages and cultural trajectories of much of Eurasia[10,11]. However, their emergence out of local hunter-gatherer groups and connections to nascent farming communities in the Fertile Crescent remain poorly understood, as does their ultimate disappearance in the second millennium BC.

## Genetic structure

We report new genome-wide data for 131 individuals from 38 archaeological sites and 84 new radiocarbon dates across and around the Caucasus region, including the piedmont and steppe zones, tripling the available genomic data (Fig. 1 and Supplementary Tables 1 and 2). The genetic time transect covers about 6,000 years, ranging from the Mesolithic and Neolithic (seventh and sixth millennia BC, $n = 7$), Eneolithic (fifth millennium BC, $n = 11$), Late Eneolithic and Early BA (EBA; fourth millennium BC, $n = 20$), EBA and Middle BA (MBA; third millennium BC, $n = 51$), to the final MBA and Late BA (LBA; second millennium BC, $n = 42$; Supplementary Table 1). Individuals ($n = 26$) who did not meet the quality criteria were excluded (Methods). The final dataset for population genetic analyses included 102 unrelated individuals, who were combined with published ancient and modern-day individuals (Supplementary Tables 3–5).

First we carried out principal component analysis (PCA) and ADMIXTURE analysis to qualitatively assess the genetic affinities of the ancient

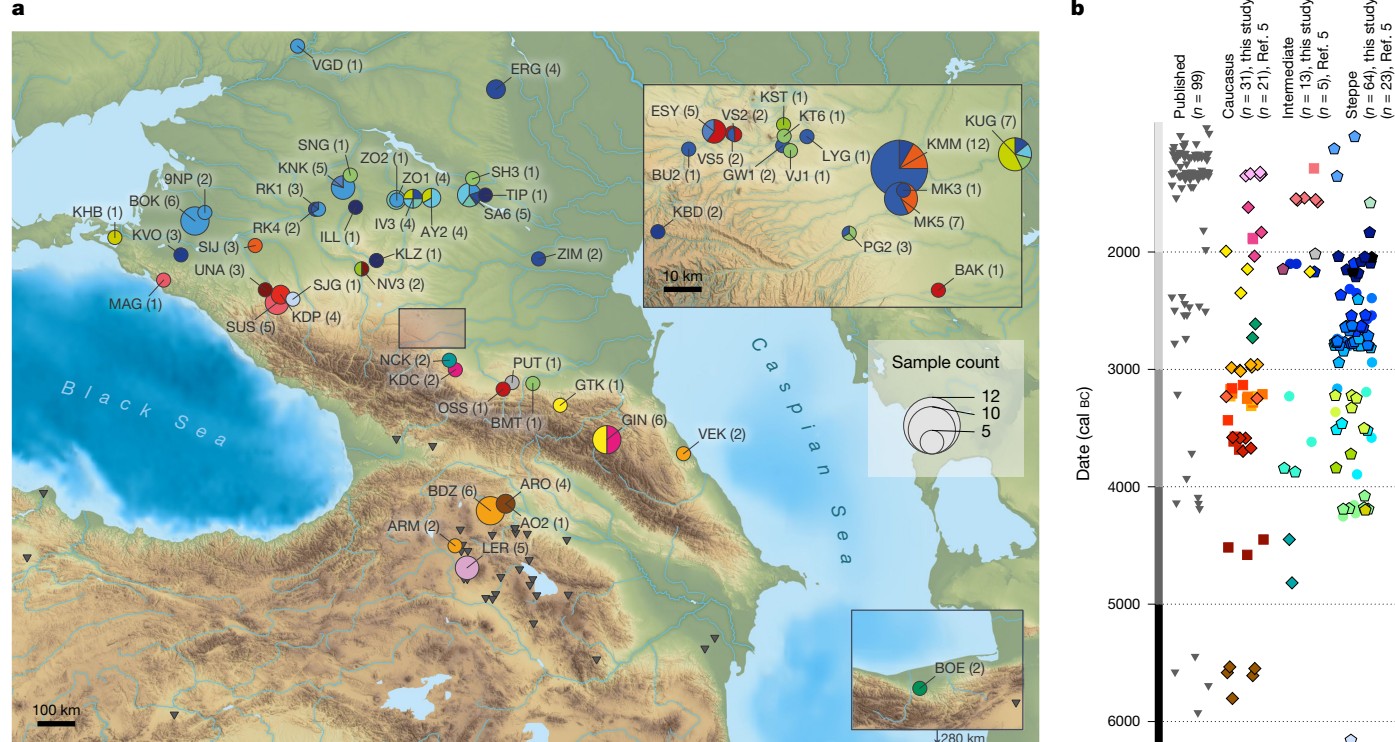

**Fig. 1 | Geographical and chronological overview of individuals analysed in this study. a**, Map of the wider Caucasus region showing the locations of the 38 sampled sites and 131 individuals from this study and published data[5] (Supplementary Tables 1 and 5). The number in parentheses indicates the studied individuals per site. The inset shows a magnified view of the area marked by the rectangle in the main image. Scale bars, 100 km (main image) and 10 km (inset). **b**, Timeline of the median [14]C ages of newly reported (black outline) and published individuals (no outline), separated on the *x* axis into main genetic and geographic clusters (Steppe, Intermediate and Caucasus), as well as other published individuals from south of the Greater Caucasus for comparison. Colours in the piecharts (**a**), and colours and symbols in the timeline (**b**) represent sampled individuals associated with different archaeological cultural complexes, and these are used consistently across all figures (see Figs. 2 and 3, Extended Data Fig. 1, Supplementary Table 1 and Supplementary Fig. 1 for full site names and individual symbols). Grey bars correspond to the chronological chapters in the results section. The map was generated using Base Relief: Mapzen, OpenStreetMap, and rivers, lakes and borders were added using free vector and raster map data from Natural Earth (https://www.naturalearthdata.com). OpenStreetMap is open data, licensed under the Open Data Commons Open Database Licence by the OpenStreetMap Foundation.

individuals (Methods and Extended Data Figs. 1 and 2). We substantiate the genetic differentiation observed between the steppe and mountain groups[5], hereafter termed the Steppe and Caucasus clusters, and describe the formation and persistence of ancestries, including mixed Intermediate groups, which reflect dynamic phases of biological and cultural interaction resulting in the establishment and spread of pastoralism, first in the Pontic–Caspian and subsequently in the entire Eurasian steppe zone (Fig. 1). Finally, during the LBA period, we observe the dissolution of the main BA ancestry clusters and the formation of the ancestry found today in the people(s) of the North Caucasus region[12,13].

## The Mesolithic–Neolithic transition

The oldest individuals in this study are from Satanaj cave in Russia (SJG001, 6221–6082 cal BC), and from the early Neolithic site of Arukhlo in Georgia (5885–5476 cal BC, *n* = 4; Fig. 1). SJG001 predates the arrival of the Near Eastern Neolithic into the Caucasus and overlaps with Eastern European hunter-gatherers (EHGs) in PC space, despite being geographically close (about 50 km) to Caucasus hunter-gatherer (CHG) sites in the South Caucasus, whose individuals carry a different genetic ancestry profile[7] (Fig. 2a). We find that SJG001 and EHG individuals from Karelia form a clade with respect to CHG individuals and other test populations (Extended Data Fig. 3a and Supplementary Tables 6 and 7). Using formal ancestry modelling with qpAdm, we were able to successfully model SJG001 with either Karelia_EHG or Sidelkino_EHG ancestry as a single source (Fig. 2c and Supplementary Table 8).

By contrast, the four Neolithic individuals from Arukhlo (ARO and AO2, Georgia_Neolithic) together with Armenia_N (with N denoting Neolithic) and Azerbaijan_LN (with LN denoting Late Neolithic) form a genetic cline between CHGs and central Anatolian Neolithic individuals (for example, Çatalhöyük), who themselves fall on a cline between Anatolian and Levantine ancestries[14–16] (Fig. 2a and Extended Data Figs. 3b and 4). Thus, we modelled all Neolithic groups from the region using Anatolia_PPN (with PPN denoting pre-pottery Neolithic), Levant_PPN and CHG-Iran_N as distal ancestry sources (Fig. 2c, Extended Data Fig. 5 and Supplementary Table 8), and find that Georgia_Neolithic and Armenia_Aknashen_N carry the highest proportion of CHG-like ancestry, whereas Armenia_MasisBlur_N and Azerbaijan_LN carry more Levant_PPN-related ancestry. However, we also tested two-way mixture models between Anatolian or Levantine Neolithic groups and CHG and find that Georgia_Neolithic can be modelled as a two-way mixture between CHG and Çatalhöyük_N (shown as Anatolia Neolithic in Fig. 2a) or Tell Kurdu (Fig. 2c), but not with PPN groups from central Anatolia, Levant and Mesopotamia or Neolithic Northwest Anatolia.

## Eneolithic

During the Eneolithic period (4900/4700–3900 BC), a settled Near Eastern Neolithic lifestyle was introduced on the northern flanks of the Caucasus in association with the Darkveti–Meshoko culture (4500–4000 BC)[5]. Subsequently, the steppe zone further north was populated by transitional forager-pastoralist groups of the lower Volga

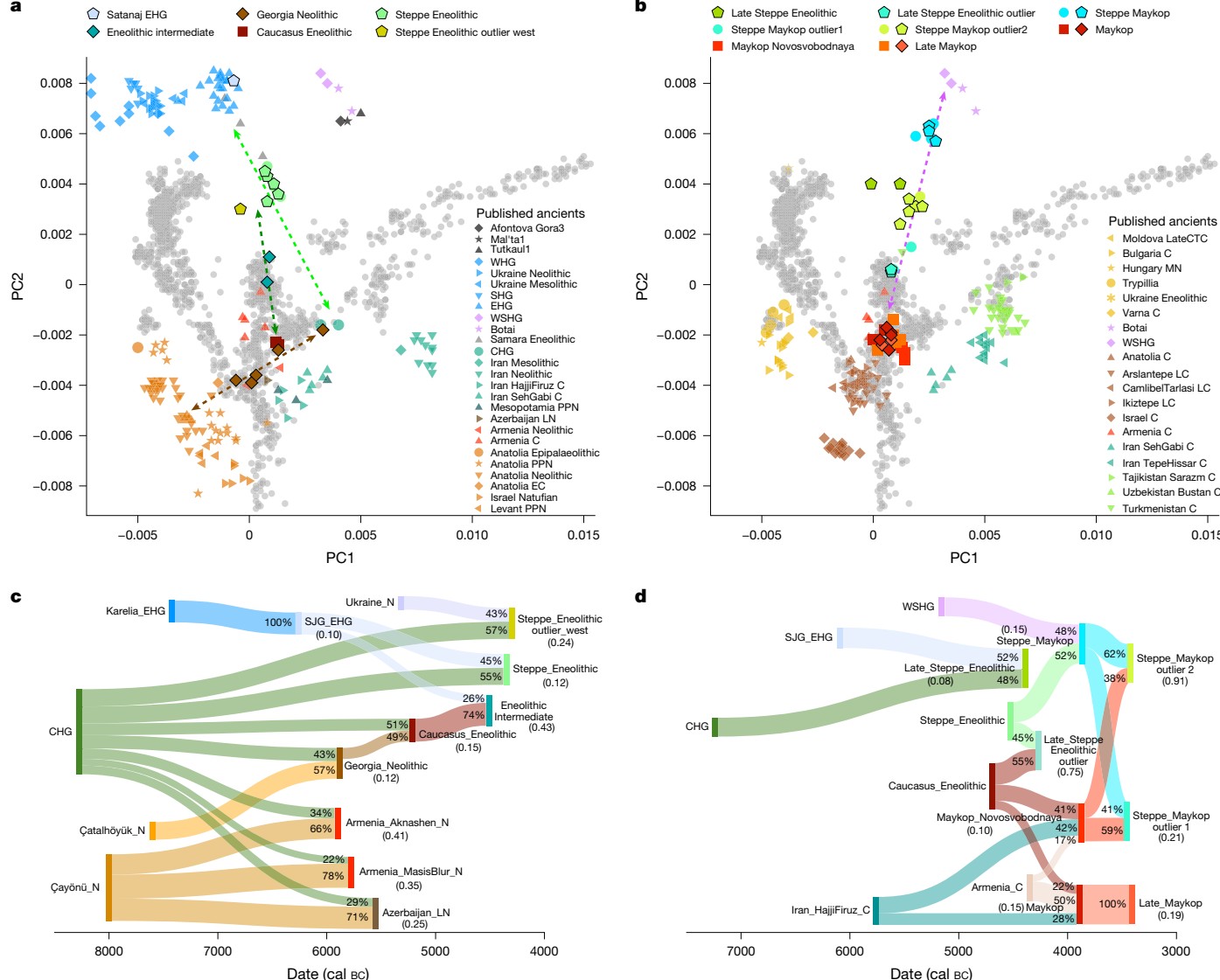

**Fig. 2 | Genetic overview of the seventh to fourth millennium BC. a,b,** PCA of newly produced ancient individuals (with outline) and individuals from previous publications (no outline) from the seventh and fifth millennium BC (**a**) and from the fourth millennium BC (**b**), projected onto 102 modern-day populations (grey dots). The dashed arrows in **a** represent the observed admixture clines between central Anatolian Neolithic and CHG (brown), between CHG and EHG (light green), and between Eneolithic_Caucasus and EHG ancestry-carrying Steppe groups (dark green), respectively. The dashed pink arrow in **b** represents an observed cline of mixture between Maykop-associated Caucasus groups and those carrying WSHG ancestry in the steppe. The corresponding labels and groupings are listed in Supplementary Table 5. **c,d,** Sankey diagram of genetic ancestry modelling for the seventh to fifth

millennium BC (**c**) and fourth millennium BC (**d**) individuals from the Caucasus region with temporally and geographically proximal sources. The admixture proportions (as percentages) are indicated on each ancestry flow, with sources on the left and target populations on the right, and *P* values for each model in brackets under the population names (Supplementary Tables 8, 10 and 13). The suffixes in the group labels present archaeological time periods and geographical regions: WHG, Western hunter-gatherer; SHG, Scandinavian hunter-gatherer; MN, Middle Neolithic; C, Chalcolithic; EC, Early Chalcolithic; LC, Late Chalcolithic. Çatalhöyük_N is shown as Anatolia Neolithic, Tell Kurdu is shown as Anatolia EC and Jordan_PPNB and Jordan_PPNC are shown as Levant PPN.

Khvalynsk Eneolithic culture[17]. Adding genome-wide data from eight new individuals allows us to describe the formation of the Eneolithic groups in the North Caucasus steppe zone. We observe the earliest formation of Steppe ancestry, resulting from the gradual mixture of EHG-like ancestry with CHG-like ancestry from the south. Together with published Eneolithic and Khvalynsk individuals[5,18], the newly reported Steppe_Eneolithic individuals form a genetic cline in PC space between EHG and CHG (Fig. 2a). $f_4$-statistics show that Steppe_Eneolithic individuals from the North Caucasus have a higher affinity to CHG than individuals from Khvalynsk (Extended Data Figs. 4 and 6 and Supplementary Table 9), and can be modelled as 55% CHG-like and 45% EHG-like ancestry (Supplementary Table 10).

We thus confirm the emergence of two distinct genetic clusters on the basis of PCA and ADMIXTURE, the Steppe cluster along the EHG–CHG cline and the mountain-oriented Caucasus cluster on the cline from Anatolia Neolithic to CHG-Iran_N (Fig. 2a, Extended Data Figs. 1 and 2 and ref. 5). Using the new Georgia_Neolithic data, we reassess the genetic ancestry of agropastoral Darkveti–Meshoko Caucasus_Eneolithic individuals from the Northwest Caucasus, which we model as a two-way mixture of Georgia_Neolithic (51% ± 6) and CHG (49% ± 6) ancestries (Fig. 2c, Extended Data Fig. 4 and Supplementary Table 10).

The individuals from Nalchik (NCK001 4531–4359 cal BC; NCK002 4930–4686 cal BC), labelled as Eneolithic_intermediate, fall between the

Steppe and Caucasus groups in the PCA, suggesting gene flow between these groups. Indeed, ADMIXTURE and $f_4$-statistics show that Nalchik individuals carry Caucasus ancestry, but also EHG-like ancestry (Fig. 2d, Extended Data Fig. 2 and Supplementary Tables 9 and 10). This means that the EHG–CHG cline and Caucasus ancestry[16] must have already been formed by the time the ancestors of Nalchik met. Further, it limits the time frame of this mixture to the early fifth millennium BC and anticipates an axis of interaction that intensifies in the fourth millennium BC. Using DATES[19], we estimate admixture dates for groups of both clusters and find that the Anatolia Neolithic to CHG-Iran_N cline formed around 6300–6000 BC, consistent with previous estimates[16], and the EHG–CHG cline formed around 5800–5300 BC (Extended Data Fig. 5c and Supplementary Table 11). Of note, KHB003 (4318–4057 cal BC) from the western-most site has a higher genetic affinity to Western hunter-gatherer (WHG) and Anatolia Neolithic-like ancestry ($|Z| > 3$), and can be modelled as a two-way mixture between CHG and Ukraine_Neolithic (Fig. 2d and Supplementary Tables 9 and 10).

## Late Eneolithic and EBA

The fourth millennium BC signifies a time period of dynamic population interaction and cultural transitions, which are visible in the material culture associated with Maykop traditions. Among the set of Late Eneolithic and EBA Maykop individuals, to which we added 20 new individuals, we confirm three previously defined genetic groups[5]: Maykop_main, Steppe_Maykop and Steppe_Maykop_outlier1, but also identify three new groups: Late_Steppe_Eneolithic, Late_Steppe_Eneolithic_outlier and Steppe_Maykop_outlier2. The genetic profiles of Maykop_main individuals can be further distinguished as three subgroups: Maykop, Late_Maykop and Maykop_Novosvobodnaya[5].

The individuals KST001 and NV3003 (3781–3652 cal BC), labelled as the Late_Steppe_Eneolithic group, fall also on the EHG–CHG cline (Fig. 2a,b). However, we find significantly higher affinity of Late_Steppe_Eneolithic individuals to EHG than Steppe_Eneolithic (Supplementary Table 12), although qpAdm reveals a similar ancestry profile with 52% SJG001 and 48% CHG ancestry (Fig. 2d and Supplementary Table 13). The individuals ZO1002 and ZO1004 (3953–3713 cal BC) are shifted towards the Caucasus cluster in PCA and ADMIXTURE (Fig. 2b and Extended Data Fig. 2), and carry less EHG-related ancestry compared to the Late_Steppe_Eneolithic individuals, and are thus labelled Late_Steppe_Eneolithic_outlier. Using proximal sources, we could model both as a two-way mixture of Caucasus_Eneolithic (55% ± 6.4) and Steppe_Eneolithic (45% ± 6.4) ancestries (Fig. 2d and Supplementary Table 13). Together with the Nalchik and Steppe_Maykop_outlier1 individuals, this reflects gene flow between Eneolithic groups living in the steppe and the Caucasus foothills (Fig. 1b).

The Maykop-associated individuals form a tight third Caucasus Maykop_main cluster, which resembles the preceding Caucasus_Eneolithic individuals, suggesting genetic continuity between these groups (Supplementary Table 12). Two-way mixture models between the preceding Caucas_Eneolithic and Armenia_C (with C denoting Chalcolithic) or Anatolia_C lack statistical support ($P < 0.05$), but the addition of Iran_C groups as a third source resulted in well-fitted models for all Maykop_main individuals (Fig. 2d and Supplementary Table 13). Late_Maykop individuals can be modelled with earlier Maykop ancestry as a single, locally preceding source, suggesting that Iran_C-related gene flow had occurred during the early Maykop phase.

The remaining two groups fall along a different genetic cline between West Siberian hunter-gatherers (WSHGs) and Caucasus_Eneolithic individuals. The first group (AY2004, IV3005 and KUG001) from sites in the Pontic–Caspian steppe dates to the Maykop period, but is genetically positioned between Steppe_Eneolithic and individuals from Botai in Central Asia and WSHG east of the Ural Mountains, who carry increased Ancestral North Eurasian (ANE) ancestry (Fig. 2b and Extended Data Fig. 2). As this group shares archaeological features attributed to the Maykop culture, it was originally described as Steppe_Maykop[5]. We show that WSHG-like ancestry contributes up to 48% to the genetic make-up of Steppe_Maykop individuals, arguing for gene flow from regions further northeast, whereas this component is absent from all other contemporaneous groups in the Caucasus and Steppe clusters (Fig. 2d, Supplementary Tables 10 and 13 and Extended Data Fig. 2).

The last group ($n = 6$; KUG002-005, IV3010 and AY2001) falls in the space of preceding Steppe_Eneolithic groups (Fig. 2b), but $f_4$-statistics and single-source qpAdm models reject direct population continuity with the preceding Steppe_Eneolithic individuals (Supplementary Tables 12 and 13). However, as this group post-dates the horizon of Maykop and Steppe_Maykop interaction, and three out of four male individuals carry the Y-chromosome haplogroup Q1b-M346 more commonly found in Steppe_Maykop and North Siberian populations, we explored alternative models involving Steppe_Maykop ancestry. This group can indeed be modelled as a two-way mixture of Steppe_Maykop (62 ± 1.6%) and Maykop_Novosvobodnaya (38 ± 1.6%) ancestries, and was consequently labelled Steppe_Maykop_outlier2.

## From EBA to MBA

We report new genome-wide data from ten individuals associated with the Yamnaya cultural complex that we refer to as Yamnaya_North_Caucasus (Yamnaya_NC; 3300–2800 BC; Supplementary Table 1). They broadly fall on the EHG–CHG cline of the Steppe groups in PC space (Fig. 3a, Extended Data Fig. 1 and Supplementary Table 14), forming a tight cluster with published data from the Black Sea, Samara and North Caucasus regions. $f_4$-statistics confirm their close genetic similarity, albeit with subtle geographic differences. Responding to previous studies that explored the regions of contact between Yamnaya pastoralists and farming groups[5,20,21], we tested a series of possible two-way qpAdm models. Using various Steppe groups as the baseline ancestry and Cucuteni–Trypillia and Globular Amphora (CTC-GAC) or Maykop_main individuals as a second 'farming-associated' source, only western Ukraine_Yamnaya can be modelled as a two-way mixture of Steppe_Eneolithic and CTC-GAC, whereas these models are rejected for Yamnaya_NC and Yamnaya_Samara individuals (Supplementary Table 15). However, adding Ukraine_Neolithic and Mesolithic as a third source improved the model fit for almost all groups. Thus, we can model Yamnaya_NC as a three-way mixture of the local proximal sources Steppe_Eneolithic, Maykop and Ukraine_Neolithic (Fig. 3c), although models with CTC-GAC as alternative source(s) are also supported.

Later MBA individuals associated with the North Caucasus culture (NCC; 2800–2400 cal BC, $n = 12$) and Catacomb culture (2800–2200 cal BC, $n = 8$) also fall within the cluster of Yamnaya-associated individuals in PCA (Fig. 3a), and $f_4$-symmetry tests reveal only subtle genetic differences among them (Supplementary Table 14), which suggests genetic continuity. Indeed, Catacomb individuals can be modelled with Yamnaya_NC as a single, locally preceding source, whereas NCC individuals can be modelled only with Ukraine_Yamnaya instead (Supplementary Table 15). All Steppe groups from the third millennium BC overlap in their admixture date estimates, ranging from about 4800 BC to 4000 BC, and thus differ from the early Eneolithic formation of steppe-related ancestry, but coincide with the presence and early interaction of both Steppe and Caucasus Eneolithic groups north of the Caucasus (Supplementary Table 11).

A similar pattern of genetic continuity and homogenization is also observed in the Caucasus cluster. Individuals associated with the Kura–Araxes culture in Georgia (3600/3300–2400 cal BC, $n = 6$) fall close in PC space and ADMIXTURE with published Kura–Araxes individuals from Armenia and Dagestan, as well as Maykop individuals (Figs. 2b and 3a and Extended Data Fig. 2), suggesting continuity of the Caucasus ancestry profile during the MBA, but with heterogeneity[20] among different Kura–Araxes groups (Supplementary Table 14). Using Maykop

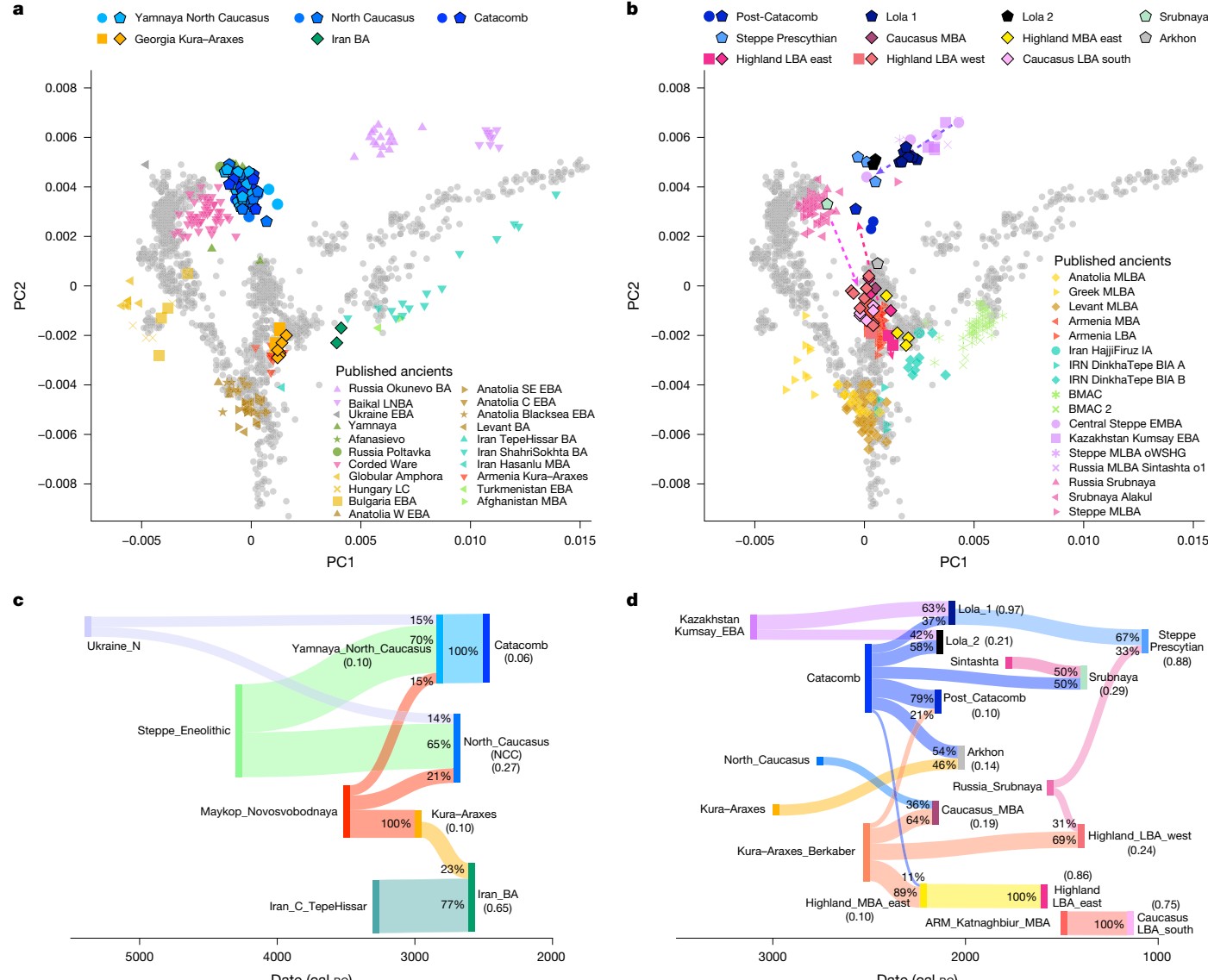

**Fig. 3 | Genetic overview of the third and second millennium BC. a,b,** PCA of newly produced ancient individuals (black outline) and individuals from previous publications (no outline) from the third millennium BC (**a**), and the second millennium BC (**b**), projected onto 102 modern-day populations (grey dots). The dashed arrows represent observed mixture clines between the Caucasus and Steppe groups and re-emerging gene flow from the northeast. The correspondent labels and groupings are listed in Supplementary Table 5. **c,d,** Sankey diagram of genetic ancestry modelling for third millennium BC (**c**)

and second millennium BC (**d**) individuals from this study based on temporally and geographically proximal sources. The admixture proportions (as percentages) are indicated on each ancestry flow, with sources on the left and target populations on the right, and *P* values for each model in brackets under the population names (Supplementary Tables 15 and 17). The suffixes in the group labels present archaeological time periods and geographical regions: MLBA, Middle–Late BA; BIA, Bronze–Iron Age; IA, Iron Age.

groups as a single source results in well-fitted models (from *P* = 0.09 to *P* = 0.7; Supplementary Table 15), with Maykop_Novosvobodnaya as the best source for Kura–Araxes individuals from Georgia and Armenia (Berkaber, Kalavan, Karnut and Shengavit), and Maykop as the best source for Talin (*P* = 0.2). By contrast, individuals from Kaps in Armenia or Velikent in Dagestan require additional ancestry from either Armenia_C or Iran_C, or both.

The Iran_BA individuals BOE001 (2861–2489 cal BC) and BOE003 (2881–2623 cal BC) fall close to those from the nearby Chalcolithic site Tepe Hissar[11] on a Southwest Asian cline. $f_4$-statistics indicate that Iran_BA has a higher genetic affinity with Chalcolithic groups from Anatolia and Kura–Araxes individuals from Karnut (Armenia), and ancestry modelling using qpAdm resulted in well-fitted models with Iran_TepeHissar_C and Kura–Araxes groups as sources (Supplementary Tables 14 and 15).

## Final MBA and LBA

The final MBA (2200–1650 BC), represented by the post-Catacomb cultural horizon[22] and LBA (1800–1200/1000 BC)[23] phases, marks another period of increased population interaction and transformation, as evidenced by 33 new individuals from this period. The PC space previously occupied by Steppe cluster individuals is now largely void, with only four individuals of different genetic ancestries (KVO009, KNK006, ESY007 and ESY009) falling in this position. All other kurgan burials in the central steppe zone are shifted towards the Caucasus cluster owing to significantly increased affinity to South Caucasus populations (*Z* = 3.352; Fig. 3b and Supplementary Table 16). Consequently, these post-Catacomb individuals can be modelled successfully as a mixture of preceding Catacomb (79%) and Kura–Araxes (21%) groups (Fig. 3d and Supplementary Table 17).

Among contemporaneous Steppe populations, individuals of the Lola culture ($n$ = 9) represent the predominant ancestry pattern, which falls close in PC space to earlier Steppe_Maykop individuals[5] (Fig. 3b and Extended Data Fig. 2). We separate them into two groups (Lola_1 and Lola_2), on the basis of differing amounts of ANE ancestry indicated by PCA and $f_4$-statistics (Supplementary Table 16). We next tested whether Lola represented a continuation of Steppe_Maykop ancestry that had returned to the North Caucasus, but find that models with Steppe_Maykop as a single ancestry source are rejected, whereas two-way mixtures of Steppe_Maykop and either NCC or Catacomb-associated ancestry are supported (Supplementary Table 17). Considering the 2,000-year time gap between Steppe_Maykop and Lola, we also tested multiple qpAdm models with North Caucasus MBA Steppe groups (for example, Catacomb or NCC) as the local substrate and central steppe Early and Middle BA[11] groups as a non-local source of ANE ancestry, and find well-fitted models with Kazakhstan Kumsay EBA. The individual KVO013 from the Northwest Caucasus falls close in PC space to BA Srubnaya individuals from the eastern European forest steppe (Fig. 3b) and can be modelled as a mixture of preceding BA Sintashta[11] and Steppe groups (Fig. 3d and Supplementary Table 17). The LBA Prescythian_steppe individuals ESY006, ESY007, ESY009 and KNK006 represent the last signal of local Steppe ancestry in the North Caucasus (Fig. 3b). However, $f_4$-symmetry tests show that these individuals carry traces of ANE ancestry similar to Lola and Steppe_Maykop, and can be modelled as a two-way mixture of Srubnaya and Lola_1 ancestry (Supplementary Tables 16 and 17).

The final MBA and LBA individuals from the Caucasus cluster are markedly shifted upwards on PC2 towards the Steppe cluster (Fig. 3b). This marks the first time in which ancient individuals fall within the same PC space as present-day populations. Individuals from the western and eastern Caucasus highlands are spread along PC2, implying varying levels of admixture with the Steppe cluster (Fig. 3b and Extended Data Fig. 2). The individuals KVO008 (MBA_Caucasus) and PUT001 (Arkhon) carry similar ancestry profiles, suggesting that a convergence of both clusters might not have been restricted to the Caucasus highlands. Using $f_4$-statistics, we find that most MBA and LBA individuals show an affinity to Steppe groups (Supplementary Table 16). Further, compared to the MBA_Highland_east group, KVO008 and PUT001 show an even higher genetic affinity to EHG–WSHG and BA Steppe groups, and both can be modelled successfully with Kura–Araxes as a proxy for Caucasus ancestry and BA Steppe groups as sources (Supplementary Tables 16 and 17). This two-way model is also supported for MBA and LBA eastern highland individuals, albeit with a higher proportion of Kura–Araxes-related ancestry, but can also be modelled with the preceding MBA_Highland_east group as a single source (Supplementary Table 17), implying that this gene flow had occurred already before the final MBA. Moreover, we observe a genetic cline from east to west, suggesting geographic structure in LBA highlander ancestry (Fig. 3b). Here, individuals from Shushuk and Marchenkova Gora in the northwest share more genetic drift with BA Srubnaya and Steppe groups compared to eastern individuals from Ginchi and Gatyn-Kale, which is consistent with the finding of the Srubnaya-associated individual KVO013 from the Northwest Caucasus (Figs. 1a and 3b). Western highlanders also show a greater genetic affinity with Maykop individuals, whereas eastern highlanders are more similar to Kura–Araxes individuals (Supplementary Table 16). Using Maykop and Kura–Araxes groups as the respective locally preceding ancestry source and Srubnaya as the second source resulted in well-fitted models (Supplementary Table 17). The five LBA individuals from the site of Lernakert in Armenia (1411–1266 cal BC) occupy a similar position in the PCA as the published MBA and LBA individuals from other sites in Armenia, suggesting local genetic continuity[20,24]. Using Armenia_MBA as the local baseline in $f_4$-statistics and qpAdm, we find support for a single local source.

## Time transects and demographic snapshots

The most conspicuous mortuary features on the Eurasian steppe are kurgans, earthen mounds that marked graves in the North Caucasus since the fifth millennium BC, throughout the BA and beyond. Kurgans were often built incrementally, spanning many centuries and cultural periods[25], and thus provide a perspective on genetic continuity or discontinuity at individual sites. In many mounds, the deceased were buried in a non-continuous series of events. Previous research has assumed close genetic or genealogical relations between individuals buried in such mounds[26]. To test this, we estimated genetic relatedness between pairs of individuals. Among 105 individuals from 21 multi-burial kurgans[5], we find 15 first- or second-degree relationships among all possible pairs ($n$ = 5,460, 0.27%). Even when filtering for pairs from the same site ($n$ = 272; 5.5%) or chrono-cultural overlap ($n$ = 1,147; 1.3%), we find predominantly unrelated or only distantly related pairs (Supplementary Tables 3, 4 and 18). Moreover, we observe a significant sex ratio bias towards male burials in the Steppe ($P$ = 0.035) but not in the Caucasus ($P$ = 0.850; Extended Data Fig. 7) cluster. Overall, this suggests that kurgans were generally not pedigree- or lineage-based burial grounds.

The multiphase mounds of Komsomolec 1-Marfa (KMM) and Marinskaya 5 (MK5) represent two well-dated examples. Focusing on the primary EBA and MBA occupation phases, with 12 and 7 individuals, respectively, we observe shifting cultural and genetic affinities among individuals in chronological succession (Extended Data Fig. 8a,b). We find a Late_Maykop brother–sister pair at KMM, and a grandfather–grandson pair at MK5[5], whereas all other individuals associated with later cultures are unrelated.

Using ancIBD[27], we estimated genetic relationships up to the sixth degree between all individuals from the Caucasus. We confirm the first-degree relationship between the Maykop individuals VIN001 and VS5001 from neighbouring mounds, and also identify more distant relationships (for example, between the grandfather–grandson pair at MK5 and the individual ESY005, buried 60 km apart; Extended Data Fig. 9 and Supplementary Table 18). However, we observe no shared identity by descent between Steppe and Caucasus cluster individuals in the entire dataset. Examining runs of homozygosity (ROH)[28], we observe that Steppe cluster individuals have a higher number of short ROH tracts (4–8 cM and 8–12 cM) compared to Caucasus cluster individuals (Extended Data Fig. 10a,c), indicating a smaller effective population size of Steppe communities. We also detect five cases of consanguinity (ROH >20 cM) in Maykop_main individuals, including offspring of second-cousin unions (AY2001 and AY2003), a first-cousin union (ESY005 and SIJ003) and a full- or half-sibling union (VS5001; Extended Data Fig. 10b,d).

## Discussion

We observe two genetically distinct populations in the Caucasus region before the Neolithic transition. SJG001 shares a close genetic affinity to hunter-gatherer groups from Karelia and the Samara region, as opposed to the geographically closer Ukraine_Mesolithic and Neolithic individuals, attesting to a lasting legacy of EHG ancestry across a large area of eastern Europe[6]. The lack of genetic admixture from the south opposes ideas of immigration of Epipalaeolithic groups from the Fertile Crescent, as proposed on the basis of similar lithic industries[29,30]. By contrast, groups carrying CHG-related ancestry must have persisted in the South Caucasus, as this ancestry is a source for Georgia_Neolithic. This cline between Anatolian Neolithic and CHG-like groups echoes the purported origin of the Neolithic expansion from the Fertile Crescent to the intermontane valleys in the Lesser Caucasus[31,32] and shows that expanding Neolithic groups interacted early and intensively with local groups. The earliest sites in the Kura and Araxes valleys date to 6000/5900 BC. The individuals

from Arukhlo reflect the rapid assimilation of the initial immigrants, which contrasts with the limited interaction between expanding farmers and WHG in Europe. The local CHG-like ancestry is also still detectable in individuals from Menteshtepe (Azerbaijan)[33] and Akanashen (Armenia)[15].

Neolithic lifeways emerged on the northern flanks of the Greater Caucasus among Eneolithic Darkveti–Meshoko agropastoral pioneers who share ancestry with South Caucasus populations. About 4300 BC, a different population, culturally related to the Khvalynsk Eneolithic[17,34], arose in the Pontic–Caspian steppe further to the north and built the first burial mounds. This marks the first appearance of Steppe ancestry in the region, formed through pre-Neolithic hunter-gatherer interactions before the emergence of the Darkveti–Meshoko groups, whose ancestry contains Anatolian Neolithic ancestry not found in Steppe groups. The initial formation of Steppe ancestry dates to the mid-sixth millennium BC and is thus consistent with the described sequence of events. KHB003 from Khutor Belyy reflects genetic interaction of Steppe groups with peoples along the eastern parts of the Black Sea[35], a trajectory that later became more important during the Yamnaya period. The oldest Eneolithic and genetically intermediate individuals from the cemetery of Nalchik provide a temporal constraint for the preceding CHG-related gene flow to the north. However, placing these Eneolithic events in sequence is challenged by radiocarbon dating reservoir effects, a problem that to some degree also affects BA pastoralist groups[36,37]. The earliest interaction between Steppe and Caucasus groups also extended to the south (for example, Areni 1 in Armenia)[24]. The consumption of dairy products by the Steppe_Eneolithic individuals KUG007 and PG2001 (ref. 2) and the presence of caprine teeth ornaments at Nalchik graves[34] suggest that incipient pastoral economies may have facilitated these contacts in the Eurasian steppe[2,38]. Future genetic studies of domesticated animals will probably clarify the origins of the animals and related pastoral technologies that had already advanced to the mountains of Central Asia and Mongolia in the early third millennium BC (refs. 39,40).

The fourth millennium BC emergence of Maykop traditions in the Caucasus mountains and piedmonts signifies a clear cultural transition, whereas Eneolithic traditions persisted in the neighbouring steppe[41]. We observe five genetically distinguishable and largely contemporaneous groups. All piedmont-associated Maykop individuals carry Caucasus ancestry inherited from southern Neolithic and Eneolithic groups, which was probably maintained by keeping close kinship ties in cohesive communities, also reflected in shared architectural construction features in some Maykop mounds. By contrast, the genetic variability and scarcity of close biological relationships in the four Steppe Eneolithic groups suggest different and more flexible kin structures, which echo the persistence of varying cultural practices[41].

For all groups, however, the archaeological record supports the transfer of Maykop material culture and social practices from the piedmonts into the steppe[25,38], and the alternating use of the same mounds by different genetic groups suggests cultural interconnectedness. As sheep dairying practices became more prominent during this period[2], isotope data from about 3500 BC onwards show a separation of grazing lands between communities in piedmont and steppe environments[42]. Maykop contexts show that the cultural interactions persisted for about 1,000 years, whereas the actual Caucasus cluster population (Maykop_main) did not spread and largely avoided intermarriage with Steppe groups. Innovations such as cattle-drawn wheeled transport and initial steps towards horse domestication gradually boosted mobility and herd management[43,44]. The clockwise tilt in the genetic cline of Late_Steppe_Eneolithic individuals from an EHG–CHG axis to an WSHG–Maykop axis is thus notable, as it encompasses additional WSHG ancestry, which is also found at Botai in Central Asia, another area of incipient equid domestication[11,44,45]. The genetic affinity of the Steppe_Maykop to eastern groups reflects the opening of the Eurasian steppe, even though this link is enigmatic and not yet related to any known archaeological phenomenon. Other technological innovations also started to spread, such as grassland-adapted sheep for dairying[2], wheels and wagons[43,46], and possibly wool as a material for insulating clothes and mobile architecture[47]. The display of wealth in graves was also soon transferred westwards, as indicated by close cultural links to Usatovo groups in the northwestern Black Sea area[48]. Indeed, the genetic profiles of several Steppe_Eneolithic individuals attest to contact with groups at the region's western periphery and form the genetic substrate from which the third millennium BC Pontic–Caspian steppe pastoralists later emerged[21]. The horizon of combined innovations in the North Caucasus enabled the emergence of pastoralism and the dynamic modes of interaction, connectivity and mobility that subsequently spread across larger geographic regions, bridging the Caucasus region, the Pontic–Caspian steppe and Europe with lands in Central and Inner Asia[10].

Genetic and archaeological evidence combined offer a perspective on the consolidation of pastoral economies in the third millennium BC, including homogenization of the Steppe ancestry profile and the emergence of Yamnaya groups. Notably, members of culturally distinct NCC and Catacomb communities[25] also fall into this homogeneous genetic group. Moreover, we find individuals carrying Steppe ancestry at sites in the Caucasus mountains[5], which suggests that groups with Caucasus ancestry had retreated higher into the mountains. The scarcity of closely related Steppe individuals is remarkable, given that some stem from narrow burial sequences within one mound. Combined with an elevated parental background relatedness, this suggests a form of social organization and kinship that regulates exogamy within a relatively small effective population. The Western Eurasian steppe pastoralists, best represented by the Yamnaya culture, stabilized and expanded their economy based on multispecies dairy products[2,49,50] and wheeled vehicle mobility[46], and spread their sustainable, permanent and self-supporting mobile lifestyle across the Eurasian steppe[39,40,51].

By contrast, the contemporaneous Caucasus population is represented only by individuals of the Kura–Araxes culture at present. In parallel to the pastoralist expansions across the Eurasian steppe, Kura–Araxes groups expanded into the Levant and today's Iran[52], which is reflected in the genetic shift towards Iranian BA populations. We observe no genetically intermediate individuals, but also acknowledge sampling gaps for this epoch.

The second millennium BC is marked by a decline of activity and population density in the North Caucasus steppe, and a total abandonment by about 1700 BC (ref. 2), possibly brought about by climatic shifts and overexploitation of ecologically fragile steppe habitats following the 4.2 kyr BP environmental crisis[22,53]. The homogeneous Steppe ancestry cluster of the preceding third millennium BC dissolves into several post-Catacomb groups, who show genetic affinities to Central Asia, as seen among the Lola, or to the northwestern Srubnaya. Interaction between mountain and steppe cultures also intensified during this period, evident in material culture and the genetic shift of the Caucasus groups towards the Steppe cluster and vice versa. This was previously interpreted as an expansion of mountain populations into the Pontic–Caspian steppe between 2200 BC and 1700 BC (ref. 22), but we observe instead an absorption of Steppe groups into Caucasus groups, probably driven by the steppe habitats becoming increasingly inhospitable.

From these developments, a pan-Caucasian mountain interaction sphere from dolmens in the northwest (Shushuk) to Dagestan (Ginchi) emerged that culturally represents the LBA[23] and also a genetic ancestry profile that still persists in the North Caucasus today[12,13]. This process anticipated the integration of the mountain populations in the succeeding LBA and Early Iron Age cultures[23], and marked the transition from a mobile BA steppe pastoralist economy to a more sedentary and complex agropastoral mountain economy[54].

For two millennia, mobile pastoralism dominated lifeways on the great expanses of steppe extending northwards from the Caucasus mountains. Fuelled by technological innovations such as wheeled transport and dairy pastoralism, as well as emerging horse husbandry, steppe populations from the Caucasus–Steppe interface exerted a large influence on the Eurasian landmass, leaving far-flung genetic and cultural footprints that remain even today. Understanding the dynamic and complex population interactions that shaped the region's most influential BA groups, such as the Maykop, Yamnaya and Kura–Araxes, is key to reconstructing the population history of both Europe and Asia. Here we reveal the genetic events that led to the formation of these groups and trace the region's history through its ultimate decline and abandonment about 1700 BC, even as the region's mobile pastoralism legacy continued to spread and flourish elsewhere[55].

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

[1]Department of Archaeogenetics, Max Planck Institute for Evolutionary Anthropology, Leipzig, Germany. [2]Eurasia Department, German Archaeological Institute, Berlin, Germany. [3]School of Computer and Mathematical Sciences, University of Adelaide, Adelaide, South Austalia, Australia. [4]Independent researcher, Stavropol, Russian Federation. [5]Department of Anthropology, University of Texas at Austin, Austin, TX, USA. [6]State Key Laboratory of Protein and Plant Gene Research, School of Life Sciences, Peking University, Beijing, China. [7]'Nasledie' Cultural Heritage Unit, Stavropol, Russian Federation. [8]Research Institute and Museum of Anthropology of Lomonosov Moscow State University, Moscow, Russian Federation. [9]UCSC Paleogenomics Lab, Department of Anthropology, University of California, Santa Cruz, Santa Cruz, CA, USA. [10]State Museum of Oriental Art, Moscow, Russian Federation. [11]UCSC Genomics Institute, University of California, Santa Cruz, Santa Cruz, CA, USA. [12]Otar Lordkipanidze Centre of Archaeological Research, Georgian National Museum, Tbilisi, Georgia. [13]Department of Archaeology, Faculty of History, Lomonosovsky Moscow State University, Moscow, Russian Federation. [14]Archaeology Department, Tbilisi State University, Tbilisi, Georgia. [15]Georgian National Museum, Tbilisi, Georgia. [16]Institute of History, Archaeology and Ethnography DFRC, Russian Academy of Sciences, Makhachkala, Russian Federation. [17]Brandenburg Authorities for Heritage Management and State Archaeological Museum, Zossen, Germany. [18]Institute of Archaeology, Russian Academy of Sciences, Moscow, Russian Federation. [19]Institute of Archaeology and Ethnography, National Academy of Sciences of the Republic of Armenia, Yerevan, Armenia. [20]Department of Archaeology, University of Tehran, Tehran, Iran. [21]Kalmyk Scientific Center of the Russian Academy of Sciences, Elista, Russian Federation. [22]Archaeological Department, The State Hermitage Museum, St Petersburg, Russian Federation. [23]Research Center for the Preservation of Cultural Heritage, Saratov, Russian Federation. [24]Institut für Archäologische Wissenschaften, Ruhr-Universität Bochum, Bochum, Germany. [25]Forschungsstelle Archäologie und Materialwissenschaften, Abteilung Forschung, Deutsches Bergbau-Museum Bochum, Bochum, Germany. [26]Shirak Armenology Research Center, National Academy of Sciences of the Republic of Armenia, Gyumri, Armenia. [27]Archaeo- and Palaeogenetics, Institute for Archaeological Sciences, Department of Geosciences, University of Tübingen, Tübingen, Germany. [28]Senckenberg Centre for Human Evolution and Palaeoenvironment, University of Tübingen, Tübingen, Germany. [29]Max Planck–Harvard Research Center for the Archaeoscience of the Ancient Mediterranean (MHAAM), Jena, Germany. [30]Max Planck–Harvard Research Center for the Archaeoscience of the Ancient Mediterranean (MHAAM), Cambridge, MA, USA. [31]Department of Anthropology, Harvard University, Cambridge, MA, USA. [32]These authors contributed equally: Ayshin Ghalichi, Sabine Reinhold. ✉e-mail: ayshin_ghalichi@eva.mpg.de; sabine.reinhold@dainst.de; wolfgang_haak@eva.mpg.de

## Methods

### Permission statement

Permission to work on the archaeological samples was granted by the respective excavators, archaeologists, curators and museum directors of the sites, who are co-authoring the study. Excavation licence numbers are provided in the Supplementary Information.

### Radiocarbon dating

We obtained new direct [14]C dates for 84 individuals. Radiocarbon dating was carried out using accelerated mass spectrometry at the Curt-Engelhorn-Zentrum Archäometrie in Mannheim, Germany (Fig. 1a and Supplementary Table 1). All new and published dates from the Caucasus were calibrated on the basis of the IntCal20 database using OxCal v4.4.2.

### Ancient DNA laboratory work

All of the laboratory work was carried out in dedicated ancient DNA facilities of the Archaeogenetics Department of the Max Planck Institute for the Science of Human History in Jena and that for Evolutionary Anthropology in Leipzig, Germany, and at the University of California, Santa Cruz. We have mainly sampled petrous bones and teeth (Supplementary Table 1) using a minimally invasive method described in the archived protocols https://doi.org/10.17504/protocols.io.bdyvi7w6 and https://doi.org/10.17504/protocols.io.bqebmtan. DNA extraction for all samples was carried out with a modified protocol[56], and the details of each step are described at https://doi.org/10.17504/protocols.io.baksicwe. DNA double-stranded libraries[57] were prepared from some of these extracts using a partial uracil-DNA glycosylase[58] (UDG-half) treatment, followed by Illumina dual indexing[59]. A detailed description of the steps is available at https://doi.org/10.17504/protocols.io.bmh6k39e and https://doi.org/10.17504/protocols.io.4r3l287x3l1y/v3. For a proportion of the samples, we used an automated protocol for producing single-stranded libraries[60,61], to improve sequence retrieval from ancient DNA. The single-stranded libraries for seven individuals (Dzedzvebi, Georgia) was prepared with the Santa Cruz reaction protocol[62] and UDG-half treatment. We initially screened 253 samples from 211 individuals for ancient human DNA preservation by preparing double-stranded genetic libraries for 112 samples, single-stranded libraries for 115 samples, and both library types for 26 samples. Genomic libraries passing quality control thresholds (>0.1% endogenous DNA and >3% damage pattern) were enriched for about 1.24 million targeted single nucleotide polymorphisms (SNPs) across the human genome[18]. Details of the libraries produced for each individual are reported in Supplementary Table 2. All prepared libraries were initially sequenced on an Illumina HiSeq 4000 platform to an average of 5 million reads. Raw FastQC files were processed through the EAGER pipeline[63], for assessment of human DNA content and DNA damage profiles. After quality assessment, all of the libraries with 0.1% human endogenous DNA or more were enriched for around 1.2 million SNPs in a targeted in-solution capture (1240k SNP capture)[18]. Additional in-house capture assays were prepared for the complete mitogenome[64], for which the sequence reads were not sufficient for mitochondrial haplogroup calls, and for mappable regions of the Y chromosome[65] (YMCA), for selected male individuals (Extended Data Fig. 7b). Captured libraries were sequenced for 20 to 40 million reads, using either a single-end (1 × 75 base pair (bp)) or paired-end (2 × 50 bp) configuration.

### Data processing and genotyping

The captured sequences were demultiplexed, and then further processed using EAGER[63] (v2.4.0) and nf-core/eager (v2.3.2). AdapterRemoval (v2.3.2) was used to remove Illumina adaptors. Subsequently, BWA (v0.7.17) was used to map reads to the human reference genome hs37d5, and duplicates were removed using MarkDuplicates (v2.26.0). mapDamage (v2.2.1) was used to determine the deamination rate pattern (G to A and C to T substitutions) in the libraries. For the UDG-half-treated double-stranded libraries, the trimbam function of bamUtils v1.0.13[66] was used to trim 2 bp from terminal ends of reads. Thereafter, we used the pileupCaller (v1.5.2) (https://github.com/stschiff/sequenceTools) tool for genotyping, which generates pseudo-haploid genotypes by randomly choosing one of the alleles at every SNP position. For single-stranded libraries we used the --singleStrandMode parameter to remove post-mortem ancient DNA damage (at C>T SNPs, we discard forward-mapping reads, and at G>A SNPs, we discard reverse-mapping reads). In cases in which both single-stranded and double-stranded libraries were prepared for the same individual, the libraries were genotyped separately as described above. Subsequently, the genotypes from these different libraries were merged using a custom script by randomly picking alleles from available genotype calls.

### Ancient DNA authentication

The genetic sex of the individuals was determined with the Sex.DetERRmine tool (v1.1.2) through EAGER[63]. The genetic sex of the 120 individuals could be determined confidently, of which 71 were male and 49 female. Following this, the ANGSD[67] (v0.935) tool was used to calculate the rate of heterozygosity on the X chromosome to determine contamination in genetically male individuals, applying a contamination threshold of 5% in individuals with at least 100 X-SNP positions covered twice. Furthermore, we used Schmutzi[68] and ContamMix[69] to quantify heterozygosity on the individual mitochondrial reads. In cases in which the coverage is not sufficient or the mitochondrial to nuclear DNA ratios are very high (>200), contamination estimates are not reliable[70]. Hence, when appropriate, we depended on alternative methods and/or the behaviour of these samples in population genetic analyses. In cases in which there are multiple individuals available from the same genetic group, the downstream analyses were carried out on non-post mortem damage (PMD)-filtered genotypes as minor levels of individual contamination would be diluted within the group. In total, 26 individuals who did not meet the quality and authentication criteria (<5% nuclear contamination, <10% mitochondrial DNA contamination and >30,000 SNPs covered on the 1240k panel) were excluded.

### Uniparentally inherited markers

Sequences from mitogenome capture were aligned to the complete human mitochondrial genome, and Schmutzi[68] was used to infer the consensus sequence for each individual. The mitochondrial haplogroups were assigned to each consensus sequence using HaploGrep2 (v2.4.0)[71]. The Y-chromosome reads from the 1240k SNP capture and YMCA were genotyped according to a SNP list from the International Society of Genetic Genealogy dataset[65]. This allows for the manual inspection of assigned ancestral and derived alleles and their correction in cases in which, owing to residual ancient damage (C to T or G to A mismatches), a more derived haplogroup was called.

### Genetic relatedness

We used BREADR[72] and READv2[73] to calculate the pairwise mismatch rate and to estimate genetic relatedness between pairs of individuals belonging to the same site or the same genetic grouping. For the cases with few individuals, an unrelatedness baseline was provided based on the value observed in groups with similar genetic ancestry and/or chrono-cultural group. In the event of first-degree relative pairs, we excluded the individual with the lower number of SNPs from downstream population genetic analyses. For summary statistics, we refer to the chrono-cultural group-based estimates from READv2. Genetic data for samples that were identified as identical (that is, coming from the same individual; $n$ = 6) were merged for downstream analysis. Among the newly reported individuals, there were three pairs from Essentukskiy 1, as well as one pair and one triplet from Balitshi-Dzedzvebi II.

In addition, sample NV3002 was found to be from the same published individual NV3001 from the same site[5].

### Sex bias significance tests

To test for a significantly different sex bias ratio in Steppe and Caucasus cluster burials compared to expectation, we applied a one-sample test for proportions. We compared the proportion of reliably assigned genetically male individuals in each group to a male-to-female birth ratio of 1.06:1 (ref. 74), and calculated a $P$ value using a two-sided test, and adjusted the $P$ values using Bonferroni[75] correction to correct for multiple hypothesis tests.

### Power analysis for biological relatedness in multiphase kurgans

To investigate whether we had the statistical power required to carry out rigorous hypothesis testing for a change in relatedness within multiphase kurgans (Supplementary Table 3), we calculated the statistical power of a $\beta$-regression using the pwr package[76]. Using the standard significance level of $\alpha = 0.05$, and power of $\beta = 0.8$, we can reliably detect only an effect size of $f^2 = 1.64$ (greater than the possible effect size of 1.0 for $\beta$-regression). Similarly, if we use a $\chi^2$ test, we can reliably detect only an effect size[77] of greater than 0.305, which is considered greater than a medium effect size.

### Reference dataset

The genotyped data were merged with the Human Origins[78,79] and published ancient data[3,5–7,11,14,16,18,20,24,45,80–107] using AADR[108] and Poseidon databases[109] (Supplementary Table 5). The Human Origins dataset with 597,573 SNPs was used for analysis comparing ancient to modern groups, such as PCA and ADMIXTURE, whereas the 1240k dataset of 1,233,013 SNPs was used for comparison between ancient groups in $f$-statistics and ancestry modelling.

### PCA

We used smartpca (v16000) from the EIGENSOFT (7.2.1) software package[110] to carry out PCA with the 'lsqproject: YES', 'shrinkmode: YES' and 'numoutlieriter: 0' parameters to project ancient individuals onto PCs calculated on different subsets of the genotype data of modern individuals from the Human Origins dataset. We used genotype data of 1,243 individuals from 82 modern populations for the West Eurasian PCA (Extended Data Fig. 1a), and an extended West Eurasian PCA with genotype data of 1,522 individuals from 102 populations (Extended Data Fig. 1b and Supplementary Table 20), including 20 populations from Tajikistan, Pakistan and India that enhance the geographic resolution of the studied genetic groups.

### ADMIXTURE analysis

We carried out an unsupervised ADMIXTURE[111] (v1.3.0) analysis together with groups from the Human Origins dataset. Before analysis, we filtered the dataset for linkage disequilibrium with the parameters --indep-pairwise 200 25 0.4. Additionally, we also removed any low-coverage individual with a 97% missing rate with the parameter --mind 0.97. In the end, 239,791 SNPs passed these quality filters. We used the default fivefold cross-validation (--cv=5), ranging the number of ancestral components from $K = 1$ to $K = 17$.

### $f$-statistics

Both the ADMIXTOOLS (7.0.2) package[79] and its wrapper program admixr (v0.9.1)[112] were used to calculate all of the $f_3$- and $f_4$-statistics analyses with 'inbreed: YES' and 'f4mode: YES' parameters settings, respectively. The default block jackknife approach was used to calculate standard errors (Supplementary Tables 6, 7, 9, 12, 14 and 16).

### Ancestry modelling and admixture proportion estimation

We used the qpWave and qpAdm (v1520) software packages available in the ADMIXTOOLS[4,113] package to test various scenarios of continuity and admixture by identifying possible ancestry streams and to quantify these admixture proportions, with the 'allsnps: NO, details: YES' options. We used the 'allsnps: NO' parameter so that all underlying $f$-statistics were calculated with the same set of overlapping SNPs between all groups. In the cases in which the target individual or groups had <200,000 overlapping SNPs, we used the 'allsnps: YES' parameter to test the validity of the models. For all qpAdm models, we started with a base set of outgroups (Mbuti. DG, Russia_MA1_HG.SG, Italy_North_Villabruna_HG, Anatolia_Epipalaeolithic, Levant_PPN) and added certain groups or individuals as we progressed in time with our modelling. In other words, some of the groups that were used as a source in previous models were added to the outgroup set later. We are interested in possible proximal sources that contributed to the target groups under investigation. We used the available high-coverage individuals with ancestries relevant for our targeted regions and time periods for the outgroup set. However, we tried to keep the number of outgroup sets minimal, as large numbers eventually result in a reduction of $P$ values to below the threshold[113]. In addition, we did not use anachronistic sources in our qpAdm models. Our threshold for feasible models was $P > 0.05$. In the case of multiple feasible models, we report the most parsimonious models (that is, with the closest spatially and temporally sources and with fewer source groups). Details on the outgroups used in each model and results of all attempted models are reported in Supplementary Tables 8, 10, 13, 15 and 17.

### Admixture date estimation

We used the software DATES (v4010)[19] to estimate the date of admixture events. DATES measures decay of ancestry covariance patterns in target ancient groups, given two ancestral sources. The rate of decay is informative about the time of admixture. We used the following parameters settings: binsize 0.001; maxdis 1; qbin 10; lovalfit 0.45. To convert the estimated dates in generations to calendar years BC, we assumed 28 years per generation (Supplementary Table 11).

### Imputation

We used GLIMPSE (v1.0.1) with the default parameters to impute the individuals in this study together with published data from the Caucasus. Genotype likelihoods were determined from trimmed bam files (2 bp) using bcftools with the 1000 Genome Phase 3 release as a reference. Imputation was carried out using GLIMPSE_impute on genomic chunks of 2,000,000 bp with a buffer size of 200,000 bp. The chunks were subsequently ligated using GLIMPSE_ligate, and the most likely haplotypes were identified with GLIMPSE_sample. Individuals with >500,000 SNPs (0.5× coverage on the 1240k positions) post-imputation were included in the identity-by-descent (IBD) analysis.

### ROH

To estimate parental relatedness from regions with long ROHs for which an individual is homozygous, that is, has identical haplotypes inherited from each parent, we used hapROH (v1.0)[28] for individuals with >400,000 SNPs.

### IBD sharing

We used ancIBD (v0.4)[27] software to analyse IBD sharing between individuals with more than 500,000 SNPs and genotype probabilities higher than 0.99 after imputation using GLIMPSE[114,115]. The HapBLOCK function was used to estimate IBD sharing. After merging imputed samples, we used the vcf_to_1240K_hdf command to convert the vcf files to the recommended hdf5 format. Using the default parameters of the hapBLOCK_chroms command, we carried out IBD sharing analysis for each chromosome. For each pair, a summary statistics table was generated, keeping only the pairs with at least one IBD recorded in 20 cM for plotting.

## Reporting summary

Further information on research design is available in the Nature Portfolio Reporting Summary linked to this article.

## Data availability

The DNA sequences reported in this paper have been deposited in the European Nucleotide Archive under the accession number PRJEB73987. The genotype data are available as a Poseidon package in the Poseidon Community Archive (https://www.poseidon-adna.org/#/archive_explorer). Maps were created using QGIS 3.36.0-Maidenhead, Adobe Adobe Illustrator 28.3 and Adobe Photoshop 25.5.

## Code availability

All software used in this work is publicly available and versions are listed in the Nature Portfolio Reporting Summary. Corresponding publications are cited in the Article and Supplementary information. Custom code for the power analyses is available via Zenodo at https://doi.org/10.5281/zenodo.13709775 (ref. 116).

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

**Acknowledgements** We thank E. Kukral, A. Wissgott, G. Brandt, S. Nagel and E. Essel for support in the ancient DNA analyses, and K. Prüfer for the processing of the raw sequence data; the teams at (former) MPI-SHH-Archaeogenetics and MPI-EVA-Archaeogenetics, the Population Genetics and PALEoRIDER groups, especially E. Skourtanioti, M. Rivollat and V. Villalba-Mouco, for continued support and discussion, and A. Evteev for sampling support; all excavation teams that produced and stored the archaeological material and the radiocarbon laboratory team of Curt-Engelhorn-Centre for Archaeometry for their support. This research was financed by the Max Planck Society, the European Research Council under the European Union's Horizon 2020 research and innovation programme (grant agreement numbers

771234-PALEoRIDER, W.H.; 834616-ARCHCAUCASUS, S.H.) and the ERA.Net RUS Plus initiative (S&T-277-BIOARCCAUCASUS, S.H. and A.P.B.).

**Author contributions** W.H., S.H., S.R. and A.G. designed the study. S.R., A.A.K., K.B.-L., A.B.B., N.Y.B., Y.B.B., A.P.B., V.R.E., L.F.-S., I.G., A.R.K., K.B.K., D.L., R.G.M., K.M.-C., D.M., V.E.M., L.M., A.N., H.F.N., M.O., Y.Y.P., M.S., A.G.S., T.S., J.T., B.V., C.P. and J.K. provided materials and resources. F.A., L.S. and N.B. carried out laboratory experiments. A.G. analysed the data with the help of A.B.R. and H.Y., and A.C., W.H., S.R., C.W., C.P. and S.H. assisted with data interpretation. A.G., S.R., A.A.K., S.H., A.B.R., C.W. and W.H. wrote the manuscript with contributions from all co-authors.

**Funding** Open access funding provided by Max Planck Society.

**Competing interests** The authors declare no competing interests.

**Additional information**
**Correspondence and requests for materials** should be addressed to Ayshin Ghalichi, Sabine Reinhold or Wolfgang Haak.

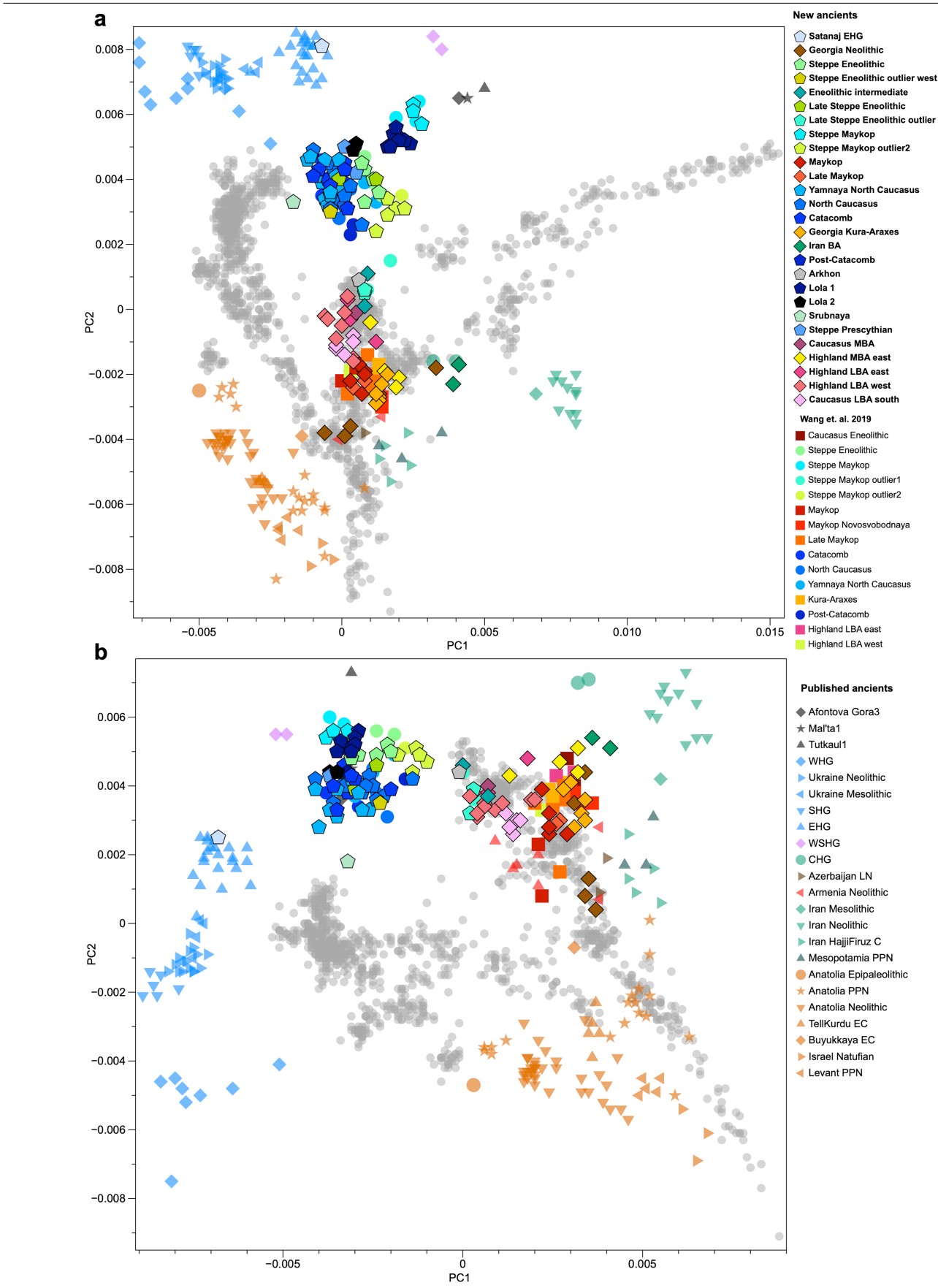

**Extended Data Fig. 1 | Principal component analysis (PCA) of prehistoric individuals from the Caucasus region. a**, extended West Eurasian PCA with 1522 individuals from 102 populations. **b**, West Eurasian PCA with 1243 individuals from 82 populations. The new individuals are shown with black outline.

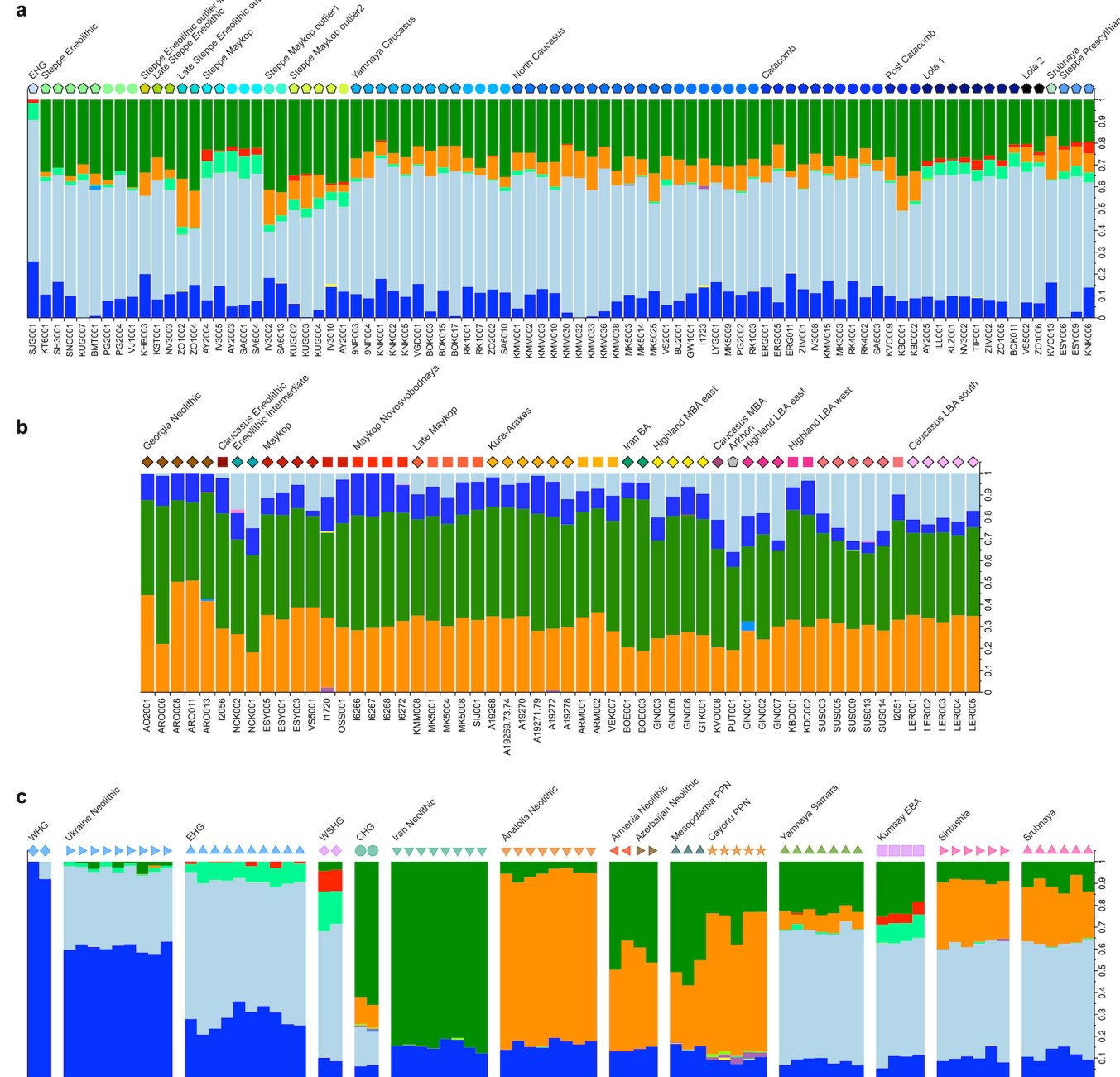

**Extended Data Fig. 2 | Results of ADMIXTURE analysis (k = 10) of the individuals from the Caucasus. a**, Individuals belonging to the *Steppe* cluster, **b**, Individuals belonging to the *Caucasus* cluster, and **c**, a representative selection of published reference individuals. Newly genotyped individuals (black outline) and published individuals are sorted by archaeological or genetic groups in chronological order from left to right.

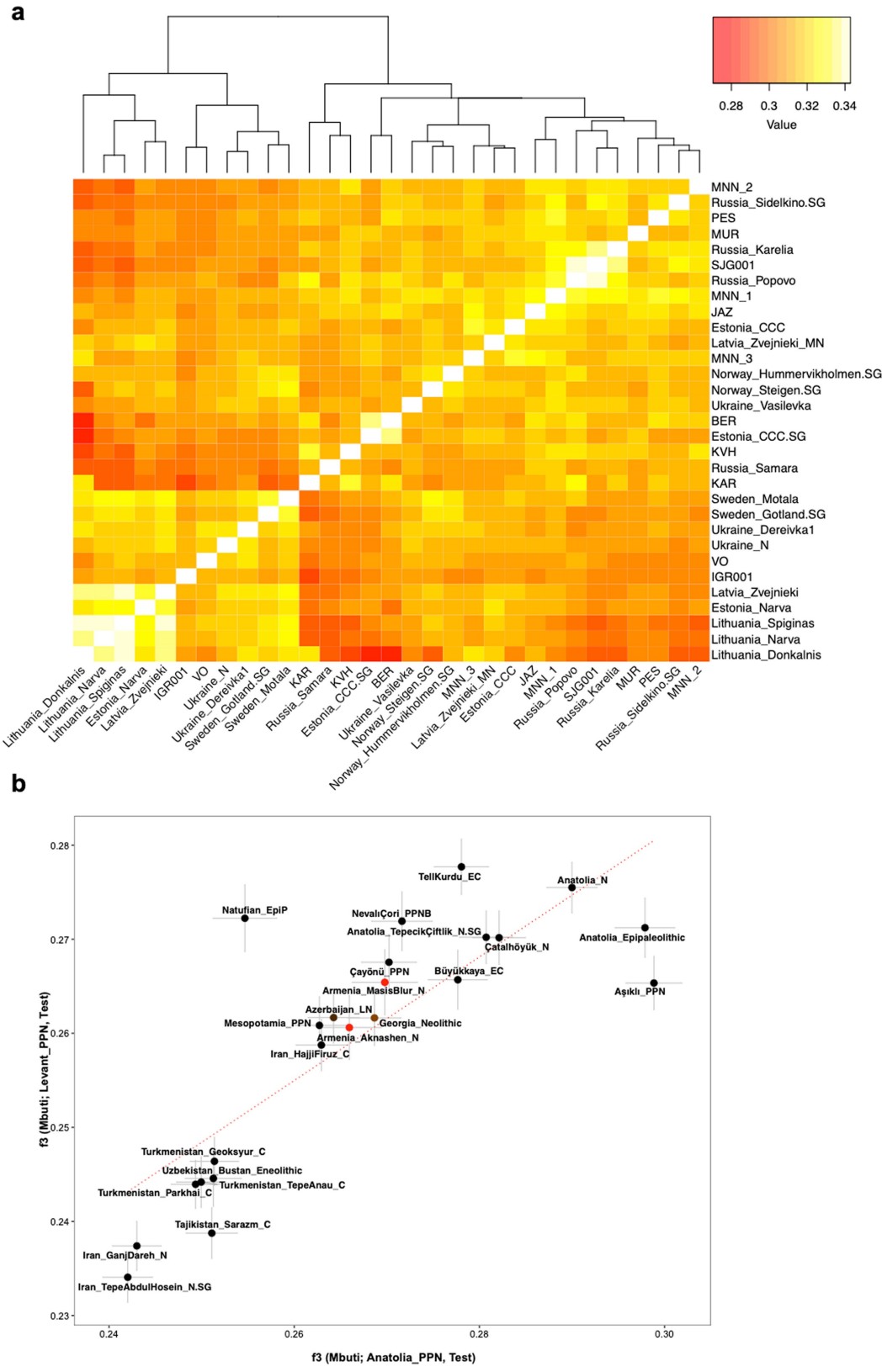

**Extended Data Fig. 3 | Outgroup $f_3$-statistics comparing genetic affinities of Mesolithic and Neolithic groups. a**, Heatmap showing shared genetic drift estimated by outgroup $f_3$-statistics of the form $f_3$(Mbuti; X, Y) between all EHG-related groups published to date. Lighter colours indicate higher $f_3$ values which corresponds to higher shared genetic drift. **b**, Scatterplot of outgroup $f_3$-statistics measuring shared drift between Neolithic and Chalcolithic groups to Anatolia and Levant PPN. Groups from the *Caucasus* cluster are shown in red and brown symbols. The trend line is given by the red dashed line. Error bars indicate ±1 s.e. and were calculated using a weighted block jackknife across all autosomes on the 1,240,000 panel (nSNPs = 1,150,639) and a block size of 5 Mb.

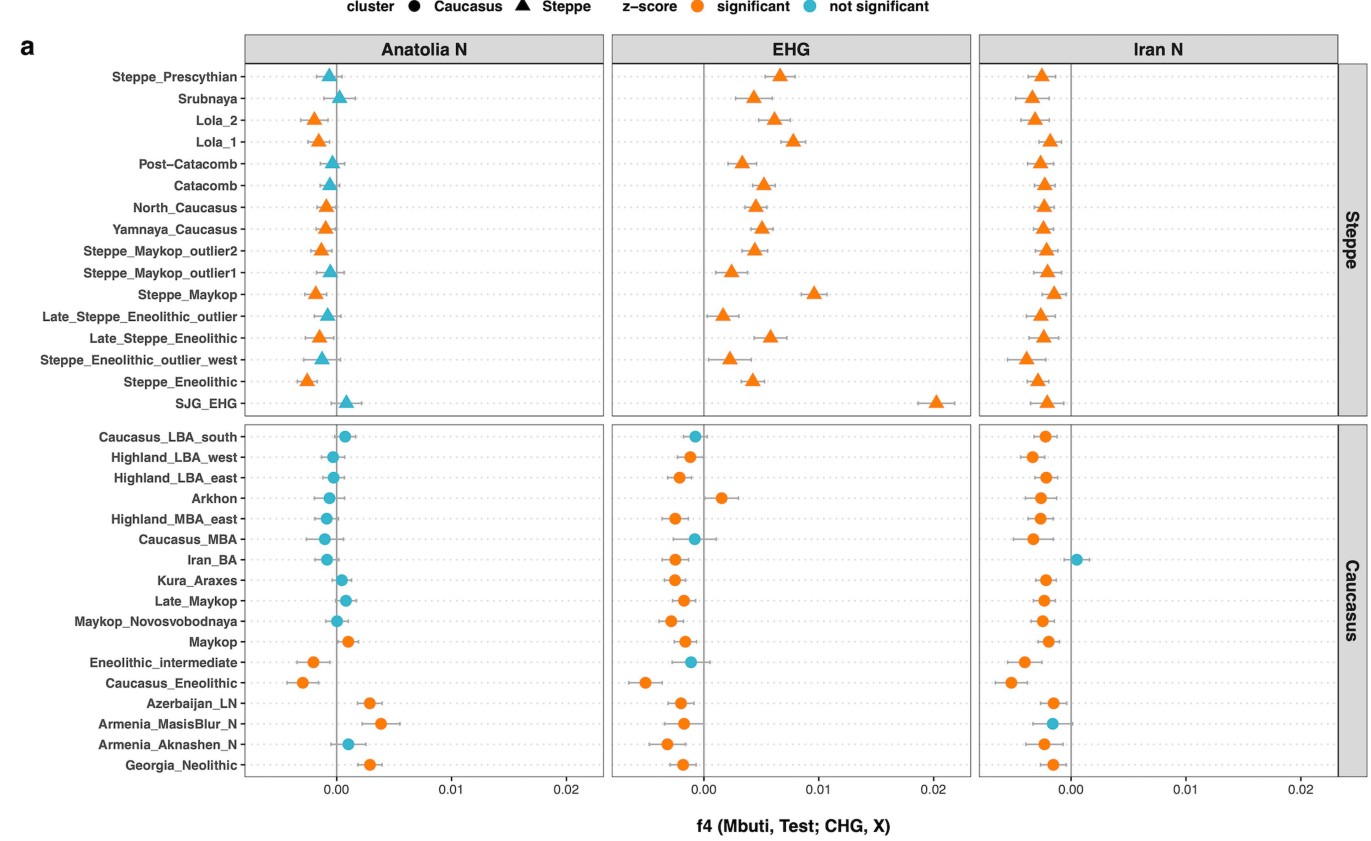

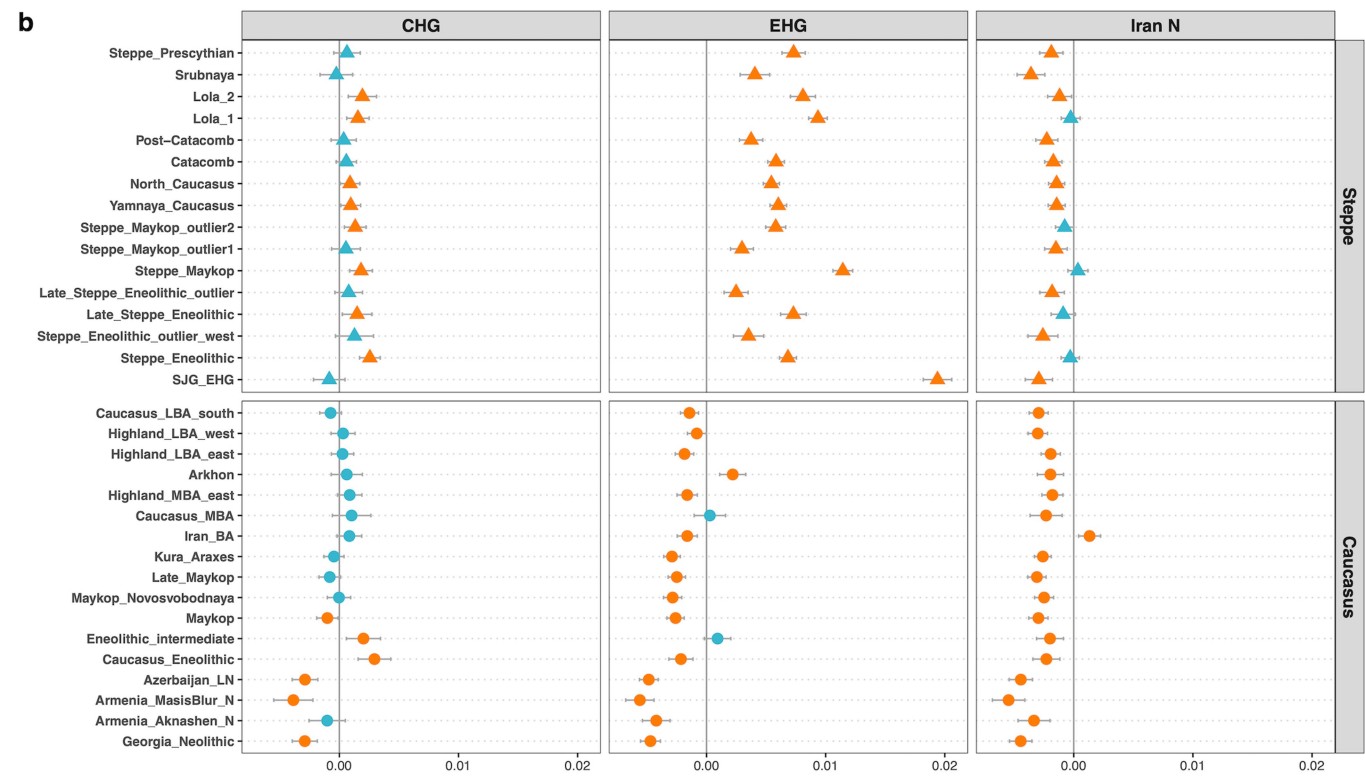

**Extended Data Fig. 4 | Formal test for temporal genetic changes in the Caucasus region. a**, $f_4$-statistic tests where $X$ denotes Anatolia N, EHG, or Iran_N and **b**, where $X$ denotes CHG, EHG, and Iran_N, and *Test* denotes various genetic groups from the *Caucasus* and *Steppe* clusters on the *y* axis. Significant Z-scores ($|Z| > 3$) are highlighted in orange and error bars indicate ± 3 s.e. and were calculated using a weighted block jackknife across all autosomes on the 1,240,000 panel (nSNPs = 1,150,639) and a block size of 5 Mb.

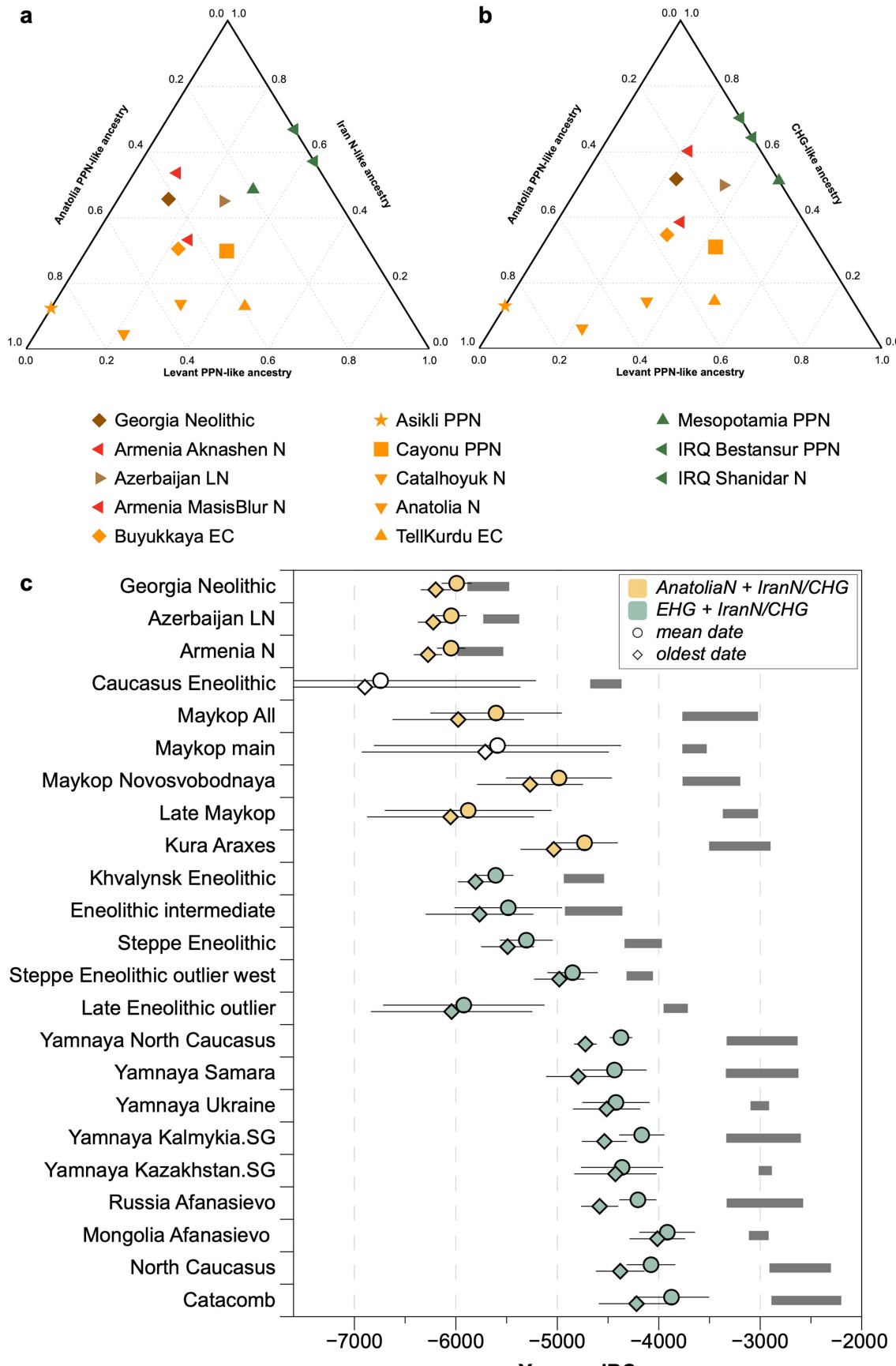

**Extended Data Fig. 5** | See next page for caption.

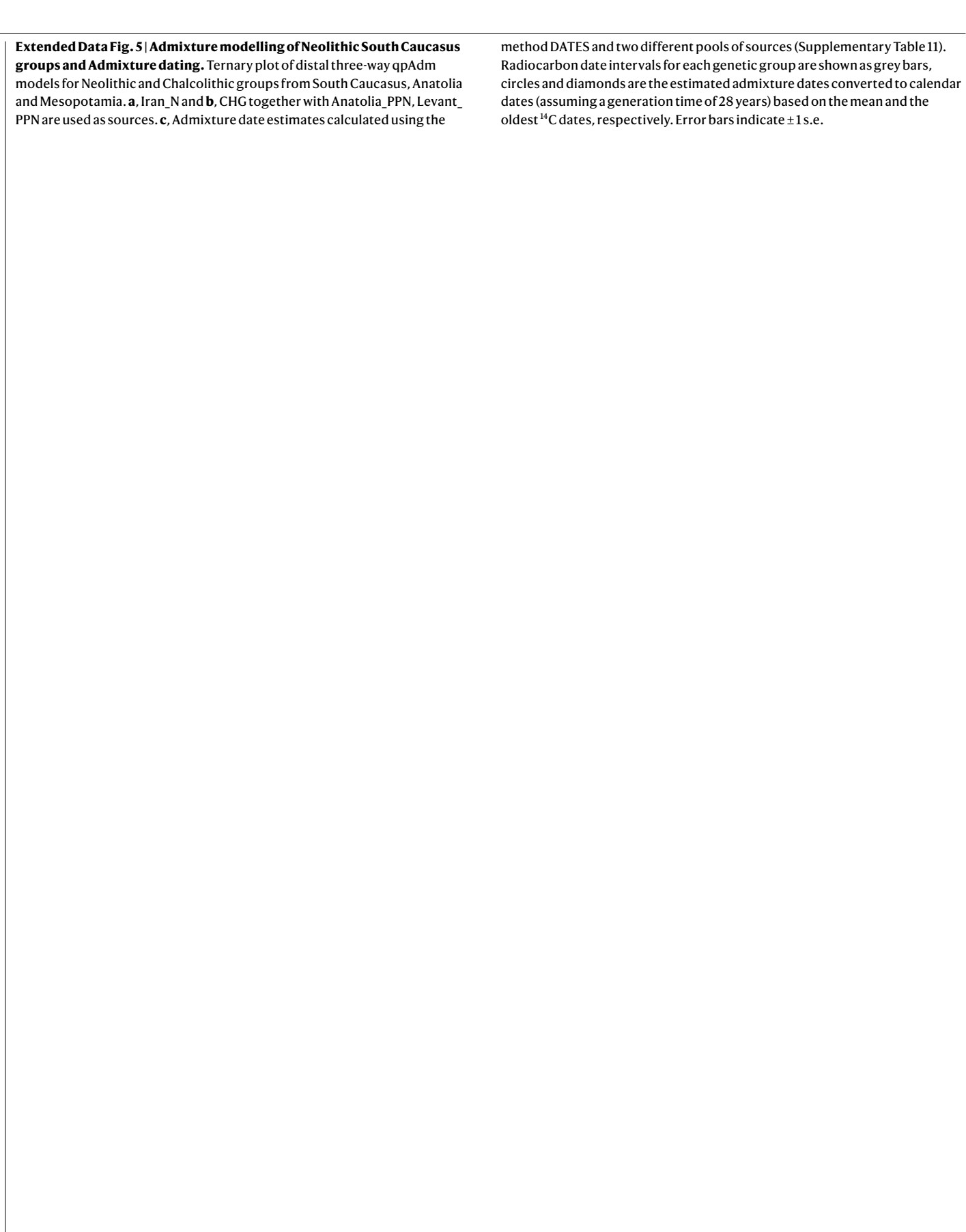

**Extended Data Fig. 5 | Admixture modelling of Neolithic South Caucasus groups and Admixture dating.** Ternary plot of distal three-way qpAdm models for Neolithic and Chalcolithic groups from South Caucasus, Anatolia and Mesopotamia. **a**, Iran_N and **b**, CHG together with Anatolia_PPN, Levant_PPN are used as sources. **c**, Admixture date estimates calculated using the method DATES and two different pools of sources (Supplementary Table 11). Radiocarbon date intervals for each genetic group are shown as grey bars, circles and diamonds are the estimated admixture dates converted to calendar dates (assuming a generation time of 28 years) based on the mean and the oldest [14]C dates, respectively. Error bars indicate ±1 s.e.

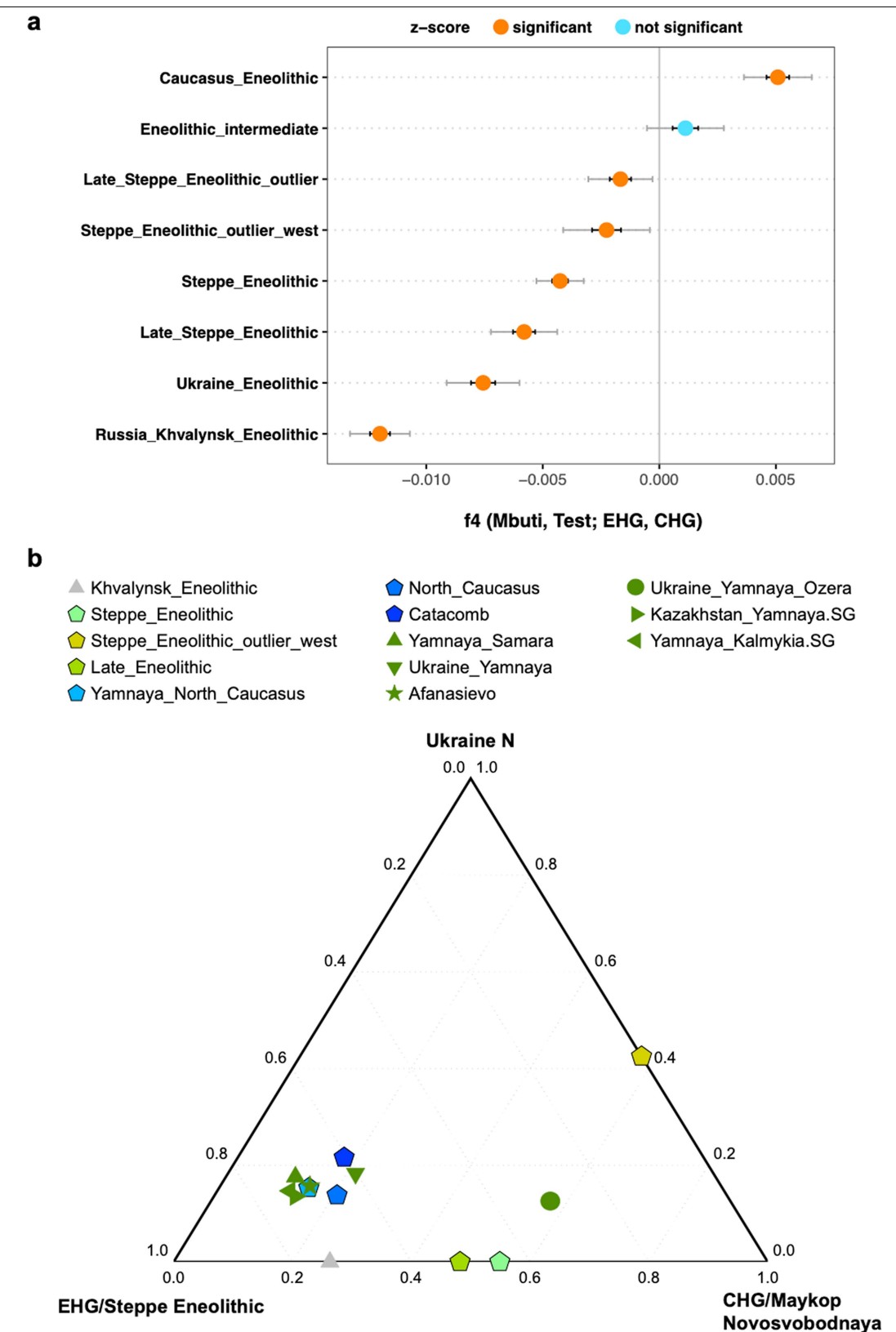

**Extended Data Fig. 6 | Key $f_4$-statistics and qpAdm results of Eneolithic and Yamnaya groups. a**, Variation in genetic affinities of Eneolithic groups to EHG or CHG. Significant Z-scores (|Z| > 3) are highlighted in orange and error bars indicate ±1 (black) and ±3 (light gray) s.e. and were calculated using a weighted block jackknife across all autosomes on the 1,240,000 panel (nSNPs = 1,150,639) and a block size of 5 Mb. **b**, Ternary plot of qpAdm modelling for *Steppe* Eneolithic and Yamnaya groups (Supplementary Tables 10 and 15).

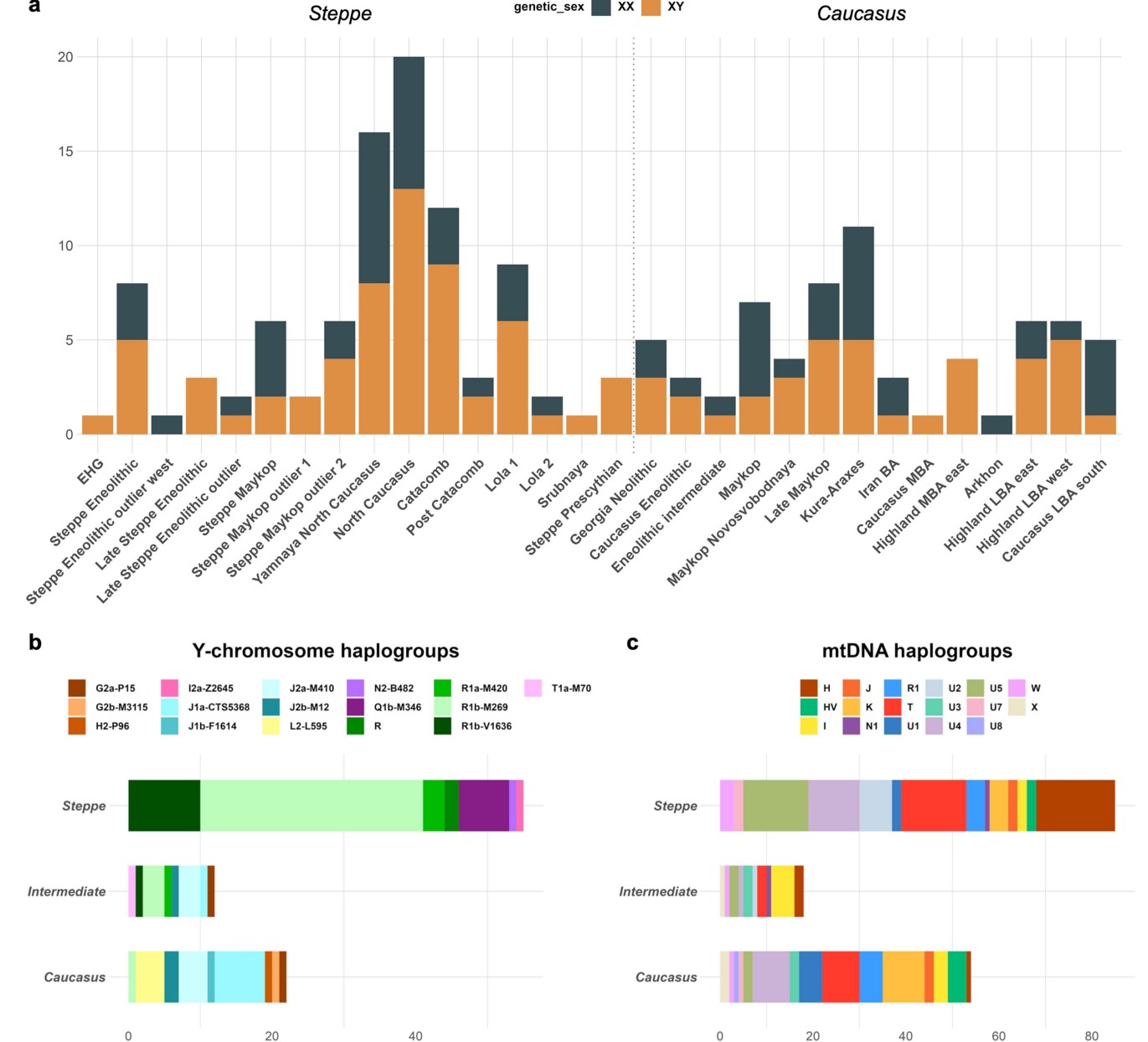

**Extended Data Fig. 7 | Results of genetic sex analysis and uniparentally inherited markers. a**, Total count of XX and XY individuals per genetic grouping from this study and Wang et al.[5]. The chrono-cultural and genetic groups are chronologically ordered within the *Steppe* and *Caucasus* clusters.

**b**, Y-chromosome haplogroups and **c**, mitochondrial haplogroups compared across the three main genetic clusters of *Caucasus*, *Intermediate* and *Steppe*. The *x* axis shows the total number of individuals per genetic cluster.

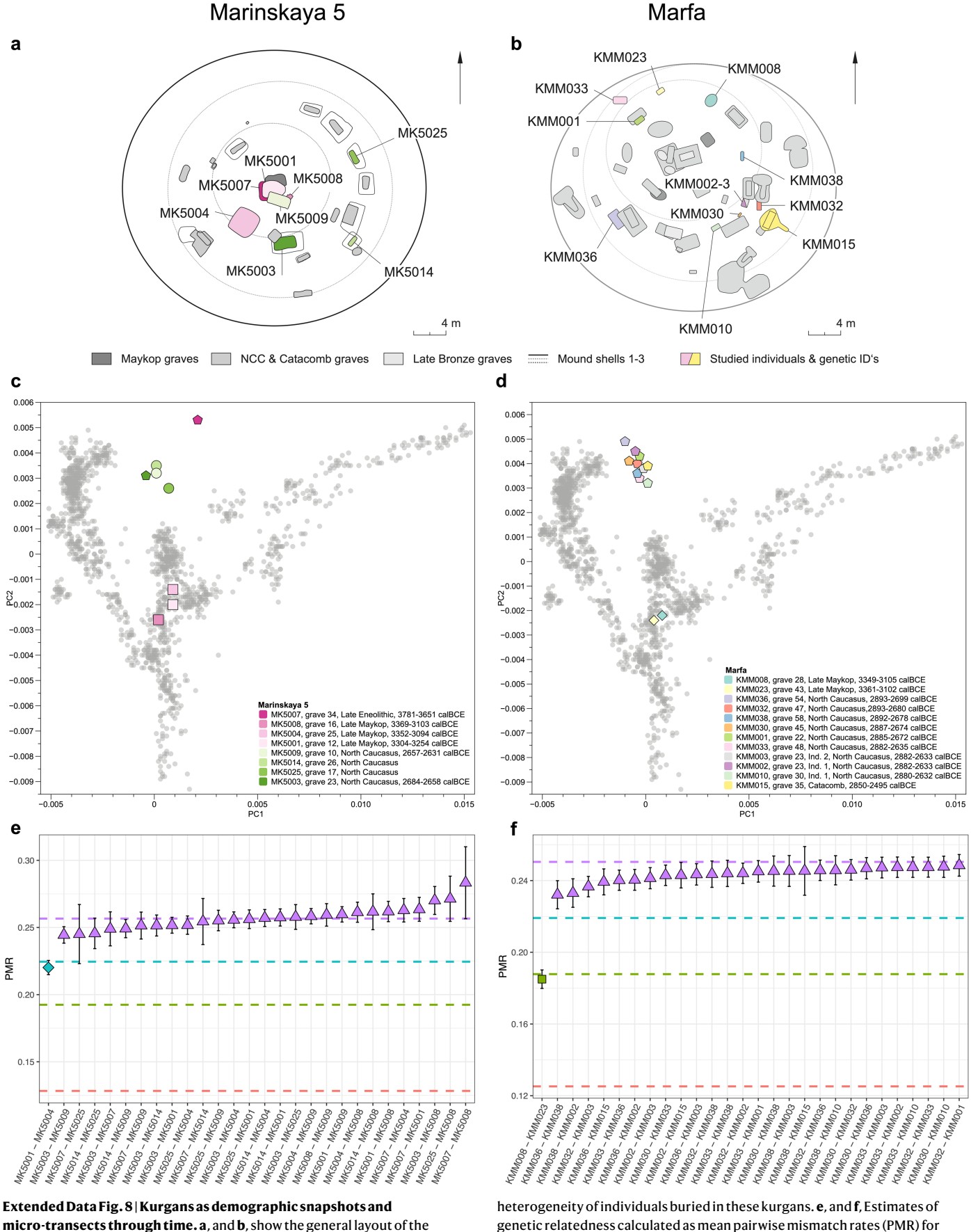

**Extended Data Fig. 8 | Kurgans as demographic snapshots and micro-transects through time. a**, and **b**, show the general layout of the Marinskaya 5 and Marfa kurgans (Bronze Age only). Analysed individuals are highlighted in different colours. **c**, and **d**, PCA plots highlighting the genetic heterogeneity of individuals buried in these kurgans. **e**, and **f**, Estimates of genetic relatedness calculated as mean pairwise mismatch rates (PMR) for individuals from Marinskaya 5 and Marfa, with error bars indicating ± 2 s.e. Dashed lines represent the expected PMR for each relatedness degree.

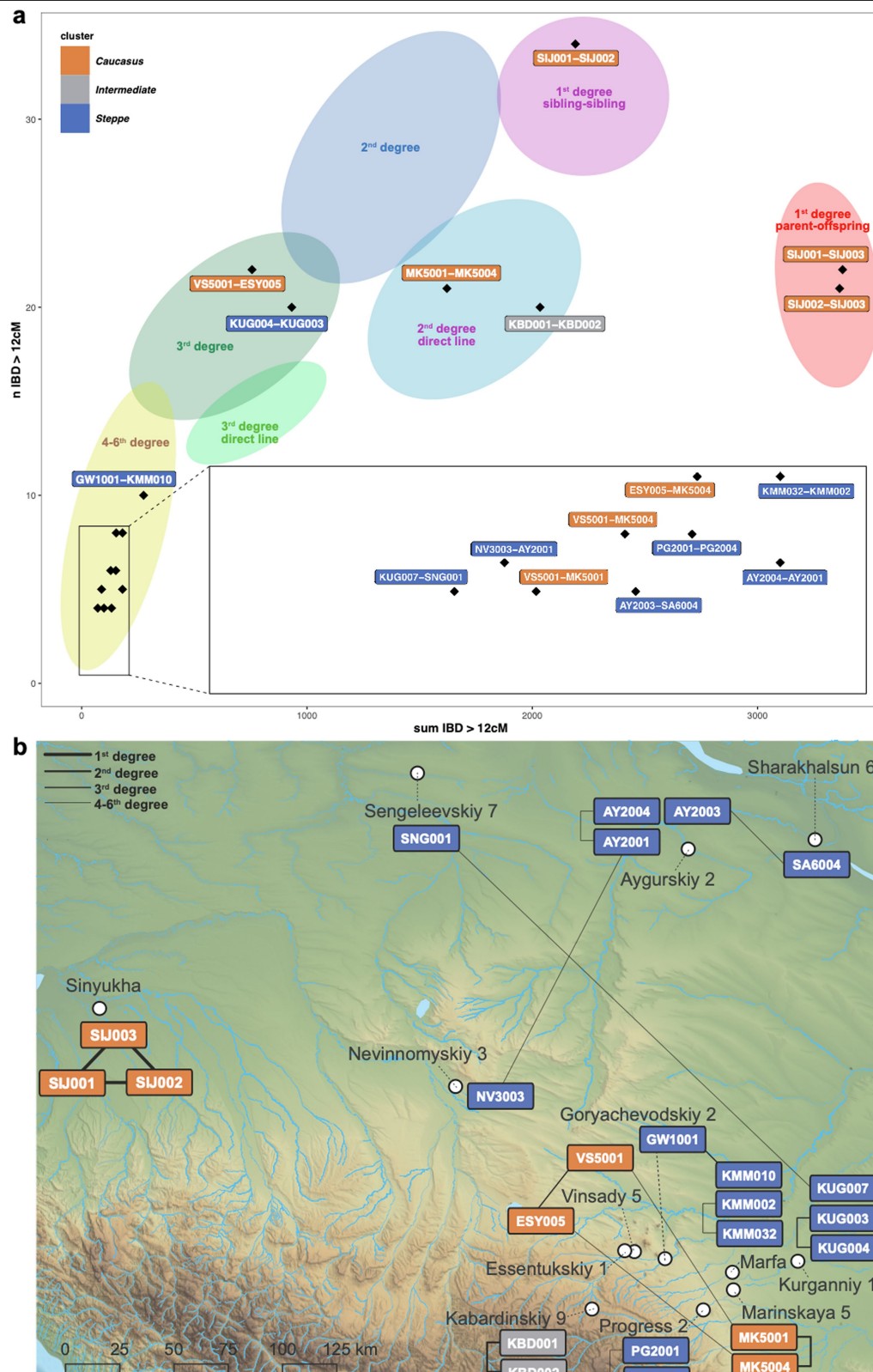

**Extended Data Fig. 9 | IBD analysis results per pair of individuals with ancIBD[27]. a**, Visualising the sum (*x* axis) and the number (*y* axis) of all IBD tracts with a length of at least 12cM. Only pairs with at least two shared chunks of 20cM length are shown. **b**, Map showing IBD networks of closely related individuals within and between sites of the piedmont and steppe zone.

The map was generated using Base Relief: Mapzen, OpenStreetMap, and rivers, lakes and borders were added using free vector and raster map data from Natural Earth (https://www.naturalearthdata.com). OpenStreetMap is open data, licensed under the Open Data Commons Open Database Licence by the OpenStreetMap Foundation.

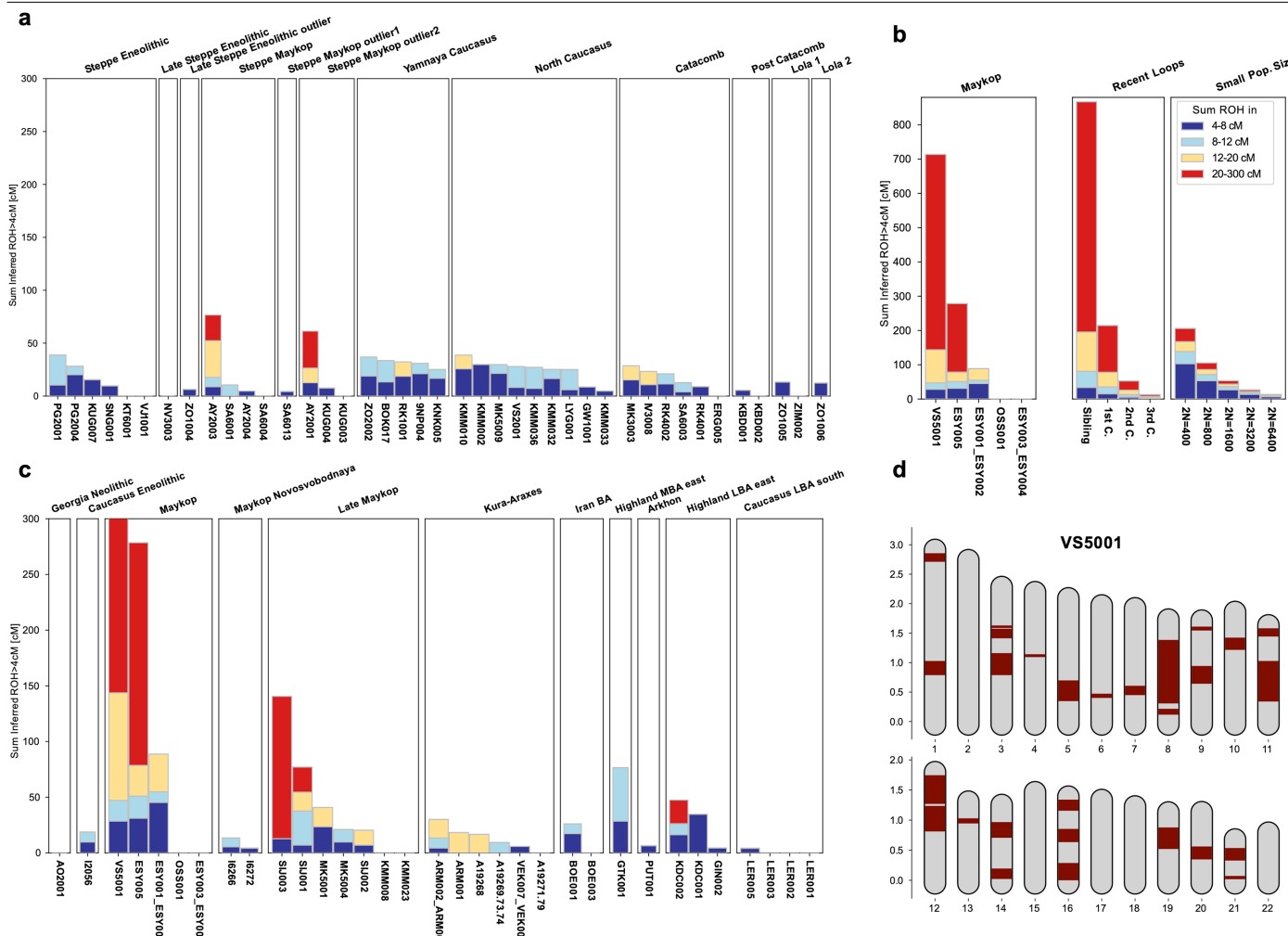

**Extended Data Fig. 10 | Assessment of runs of homozygosity and inbreeding. a**, **b**, and **c**, Inferred run of homozygosity ROH per individual in *Steppe* and *Caucasus* clusters, respectively. Results are plotted by genetic and cultural groups in relative chronological order from left to right. **b**, Full scale of inferred ROH for individuals in the Maykop group. The legend illustrates the expected ROH for offspring of small populations or closely related individuals. **d**, Karyogram of VS5001. The positions of ROH longer than 4 cM are marked on the 22 autosomes (maroon).

Wolfgang Haak

# Reporting Summary

## Statistics

For all statistical analyses, confirm that the following items are present in the figure legend, table legend, main text, or Methods section.

| n/a | Confirmed | |
|---|---|---|
| ☐ | ☒ | The exact sample size (*n*) for each experimental group/condition, given as a discrete number and unit of measurement |
| ☐ | ☒ | A statement on whether measurements were taken from distinct samples or whether the same sample was measured repeatedly |
| ☐ | ☒ | The statistical test(s) used AND whether they are one- or two-sided *Only common tests should be described solely by name; describe more complex techniques in the Methods section.* |
| ☒ | ☐ | A description of all covariates tested |
| ☐ | ☒ | A description of any assumptions or corrections, such as tests of normality and adjustment for multiple comparisons |
| ☐ | ☒ | A full description of the statistical parameters including central tendency (e.g. means) or other basic estimates (e.g. regression coefficient) AND variation (e.g. standard deviation) or associated estimates of uncertainty (e.g. confidence intervals) |
| ☐ | ☒ | For null hypothesis testing, the test statistic (e.g. $F$, $t$, $r$) with confidence intervals, effect sizes, degrees of freedom and $P$ value noted *Give P values as exact values whenever suitable.* |
| ☒ | ☐ | For Bayesian analysis, information on the choice of priors and Markov chain Monte Carlo settings |
| ☒ | ☐ | For hierarchical and complex designs, identification of the appropriate level for tests and full reporting of outcomes |
| ☒ | ☐ | Estimates of effect sizes (e.g. Cohen's *d*, Pearson's *r*), indicating how they were calculated |

*Our web collection on statistics for biologists contains articles on many of the points above.*

## Software and code

Policy information about availability of computer code

| Data collection | No specific software was used for sample and data collection. Software used for the processing of raw sequence data and genotype data is listed below. |
|---|---|
| Data analysis | We used the following freely available software for data analyses and preparation of figures and maps, and provide the corresponding citations in the Material & Methods section: EAGER (v2.4.0), nf core/eager v2.3.2 (https://nf-co.re/eager), (FastQC (v0.11.9), AdapterRemoval (v2.3.2), BWA (v0.7.17), CircularMapper (v1.93.5), MarkDuplicates (v2.26.0), MapDamage (v2.2.1), samtools (v1.3), pileupCaller (v1.5.2), bamUtils (v1.0.13), Sex.DetERRmine tool (v1.1.2), ANGSD (v0.935), contamMix (v1.0 12), Schmutzi (v1.0), ADMIXTOOLS (v7.0.2) (qp3Pop, qpDstats, qpWave and qpAdm [v1520]), admixr (v0.9.1), ADMIXTURE (v1.3.0), EIGENSOFT package (v7.2.1), smartpca (v16000), Haplogrep 2 (v2.4.0), ancIBD (v0.4), hapROH (v1.0), GLIMPSE (v1.0.1), DATES (v4010), OxCal (v4.4.2), READ2 (v2.0), BREADR (v1.0), Geneious (v2019.2.3), Sankeymatic (no versioning), Datagraph (v5.2), R (v4.3.3), R Studio (v2023.12.1+402), and QGIS (v3.36.0). The maps were generated using QGIS and data from Base Relief: © Mapzen, OpenStreetMap (no versioning), and rivers, lakes and border were added using free vector and raster map data from naturalearthdata.com, with final touches in Adobe Illustrator 28.3 and Photoshop 25.5. |

For manuscripts utilizing custom algorithms or software that are central to the research but not yet described in published literature, software must be made available to editors and reviewers. We strongly encourage code deposition in a community repository (e.g. GitHub). See the Nature Portfolio guidelines for submitting code & software for further information.

## Data

Policy information about availability of data

All manuscripts must include a data availability statement. This statement should provide the following information, where applicable:

- Accession codes, unique identifiers, or web links for publicly available datasets
- A description of any restrictions on data availability
- For clinical datasets or third party data, please ensure that the statement adheres to our policy

Genomic sequence data (fastq and BAM format) will be available at the European Nucleotide Archive under project accession number PRJEB73987. The published genotype data available in compiled and annotated format as Allen Ancient DNA Resource (AADR v44.3) as well as the POSEIDON repository was used for comparative analyses and is available here:
https://reich.hms.harvard.edu/allen-ancient-dna-resource-aadr-downloadable-genotypes-present-day-and-ancient-dna-data.
https://www.poseidon-adna.org/#/
The human mitochondrial revised Cambridge Reference Sequence (NC 012920.1): https://www.ncbi.nlm.nih.gov/nuccore/251831106.
The human reference genome GrCh38 (hg38): https://www.ncbi.nlm.nih.gov/datasets/genome/GCF_000001405.26/
The human reference genome GrCh37 (hg19): https://www.ncbi.nlm.nih.gov/datasets/genome/GCF_000001405.13/
hs37d5 is consistent with GRCh37, and contains the rCRS mitochondrial sequence, Human herpesvirus 4 type 1 and concatenated decoy sequences: https://ftp.1000genomes.ebi.ac.uk/vol1/ftp/technical/reference/phase2_reference_assembly_sequence/hs37d5.fa.gz
The 1000 human genomes reference panel: http://hgdownload.cse.ucsc.edu/gbdb/hg19/1000Genomes/phase3/

## Research involving human participants, their data, or biological material

Policy information about studies with human participants or human data. See also policy information about sex, gender (identity/presentation), and sexual orientation and race, ethnicity and racism.

| | |
|---|---|
| Reporting on sex and gender | not applicable |
| Reporting on race, ethnicity, or other socially relevant groupings | not applicable |
| Population characteristics | not applicable |
| Recruitment | not applicable |
| Ethics oversight | not applicable |

Note that full information on the approval of the study protocol must also be provided in the manuscript.

# Field-specific reporting

Please select the one below that is the best fit for your research. If you are not sure, read the appropriate sections before making your selection.

☒ Life sciences  ☐ Behavioural & social sciences  ☐ Ecological, evolutionary & environmental sciences

For a reference copy of the document with all sections, see nature.com/documents/nr-reporting-summary-flat.pdf

# Life sciences study design

All studies must disclose on these points even when the disclosure is negative.

| | |
|---|---|
| Sample size | We did not determine ancient DNA sample size a priori. Sample sizes for ancient groups/populations depend entirely on availability and preservation of human skeletal remains (and thus ancient DNA molecules) associated to archaeologically described cultures and/or techno-complexes. |
| Data exclusions | We processed and screened samples from 253 samples from 211 prehistoric individuals, of which 131 were deemed suitable for downstream analyses, following pre-defined data quality and authentication criteria, described in the Methods section. Data from specimens that showed insufficient levels of ancient DNA content or high levels of DNA contamination were excluded from further analyses. |
| Replication | We study unique entities of past populations and did not use different treatments or variations of data analyses. Experiments are carried out once and replication only occurs partially and randomly, e.g. by generating multiple DNA extracts and/or DNA libraries from the same sample or individuals, or when several samples turn out to belong to the same individual, as can be the case when dealing with commingled remains in collective burials. We recognize that individuals from the same region and time period of the past show similarities, and that their particular ancestry composition does not exist in the same form anymore today. Genome-wide data with hundreds of thousands of SNPs allows for multiple realisations of the sample history. |
| Randomization | Following model-free approaches such as principal component analysis, prehistoric individuals are grouped by chrono-cultural contexts, i.e. |

| Randomization | time period (archaeological culture, radiocarbon date), geographic region and genetic similarity. Randomisation is thus not relevant/ applicable to this study. |
|---|---|
| Blinding | The archaeological and anthropological context of our samples (date, location, material culture etc.) is critical to the interpretation of the data, blinding is not applicable to our study. |

# Reporting for specific materials, systems and methods

We require information from authors about some types of materials, experimental systems and methods used in many studies. Here, indicate whether each material, system or method listed is relevant to your study. If you are not sure if a list item applies to your research, read the appropriate section before selecting a response.

## Materials & experimental systems

| n/a | Involved in the study |
|---|---|
| ☒ ☐ | Antibodies |
| ☒ ☐ | Eukaryotic cell lines |
| ☐ ☒ | Palaeontology and archaeology |
| ☒ ☐ | Animals and other organisms |
| ☒ ☐ | Clinical data |
| ☒ ☐ | Dual use research of concern |
| ☒ ☐ | Plants |

## Methods

| n/a | Involved in the study |
|---|---|
| ☒ ☐ | ChIP-seq |
| ☒ ☐ | Flow cytometry |
| ☒ ☐ | MRI-based neuroimaging |

## Palaeontology and Archaeology

| Specimen provenance | All specimens were collected and analyzed with permissions from the respective local organizations for the handling of the archaeological material, and represented by local curators and collaboration partners who are listed among the co-authors of this study. All relevant organizations and contact persons are listed per site, including excavation licence numbers, are listed under paragraph 2 in the Supplementary Information. |
|---|---|
| Specimen deposition | Original specimens are stored at the Shirak Armenology Research Center of Armenian Academy of Sciences in Armenia, the Eurasia Department of the German Archaeological Institute in Germany, the  Georgian National Museum in Georgia, the Nasledie' Cultural Heritage Stavropol in the Russian Federation, and the Research Institute and Museum of Anthropology of the Lomonosov Moscow State University (RIMA) in the Russian Federation. Specimens will be returned to the respective heritage organization and museums after completion of the joint collaborations. DNA extract and libraries will remain stored at the ancient DNA laboratories of the Max Planck Institute for Evolutionary Anthropology, Jena & Leipizg, Germany. |
| Dating methods | New AMS 14C dates were obtained from ultra-filtrated collagen of 84 individuals. Collagen extraction and 14C measurements were carried out at the Curt-Engelhorn-Zentrum Archäometrie gGmbH, Mannheim, Germany. All new and published dates from the Caucasus were calibrated on the basis of the IntCal20 database and using OxCal v4.4.2. |

☒ Tick this box to confirm that the raw and calibrated dates are available in the paper or in Supplementary Information.

| Ethics oversight | No ethics oversight was required. Permission to work on the archaeological samples was granted by the respective excavators, archaeologist, and curators and museum directors of the sites, who are co-authoring the study and who approved and provided guidance on the study protocol. All steps in the analyses followed standard ethical guidelines with regards to respectful handling, documentation, storage, transport, sampling and processing of human skeletal elements. Excavation licence numbers for each site are provided in the Supplementary Information. |
|---|---|

Note that full information on the approval of the study protocol must also be provided in the manuscript.

## Plants

| Seed stocks | not applicable |
|---|---|
| Novel plant genotypes | not applicable |
| Authentication | not applicable |

