## [Peer Review File · Nature]

Manuscript Title: The rise and transformation of Bronze Age pastoralists in the Caucasus

Reviewer Comments & Author Rebuttals

Reviewer Reports on the Initial Version:

Referee #1 (Remarks to the Author):

A. Summary of the key results

In this study, Ghalichi et al. present a new genetic data set for 131 ancient human individuals and 1.24M SNP positions. The individuals analyzed come from 38 excavation sites spread across the Caucasus range and spanning a time line of 600 years. Combined with previously published data in the region, the study charts through space and time the genetic diversity of the diverse human groups once inhabiting the region, and the complex history of admixture accompanying major changes in lifestyle (from hunting-gathering to pastoralism) and/or the development key technological innovations, including wheeled-based transportation and possibly equestrianism. The authors reveal that those Mesolithic groups located on both sides of the Caucasian Mountain range comprised different genetic origins. The following Eneolithic period saw the emergence of the so-called west Eurasian ancestry and increasing exchange between the mountain and steppe ecozones. Then the genetic landscape of the region relatively stable until the Late Bronze Age started.

B. Originality and significance: if not novel, please include reference

The study is very timely and unfolds the formation and migration history of several human groups that developed key cultural and technological innovations. These questions are of critical importance for understanding the migration history of Eurasia, given the long-standing influence that those groups had, including down to the present-day. The study also comes at the time when another study presents similar findings for population groups inhabiting the Caucasian range between the 4th and early 2nd millennium BCE.

C. Data & methodology: validity of approach, quality of data, quality of presentation

Overall, the methodology is very sound, and quite standard in ancient DNA research. It is based mostly on PCA visualization to help identify (i) shifting genetic profiles over time and/or cultures, and (ii) candidate population sources for admixture modelling. The filters and parameters used in PCA reconstruction, qpADM modelling, pruning and ADMIXTURE analyses are well accepted. The maximum filter requirements used for disregarding potentially contaminated individuals (5% based on nuclear patterns of sequence variation on the X autosome in males) is on the high end, but generally the individuals concerned and belonging to similar sites and/or cultures seem to co-cluster with these least contaminated. Therefore, those permissive filters seem to have limited impact if at all.

D. Appropriate use of statistics and treatment of uncertainties

Yes.

E. Conclusions: robustness, validity, reliability

See section F. The authors contrast the number of pairs of related individuals identified in different time periods to speculate about changing social and/or funerary practices (e.g. line 522). Reporting N versus zero is, however, not sufficient to claim for statistical significance; appropriate statistical tests accounting for the different number of individuals genetically characterized in both time periods are required.

F. Suggested improvements: experiments, data for possible revision

Analyses of uniparental variation are simple and deliver highly-credible haplogroup assignments. However, this information is only presented in Table SA (incidentally, with notes for the Y chromosome that should be polished or removed; see 'column Y_notes'), and not discussed in the main text. Yet, some of the information presented is at the core of some debates about Yamnaya, e.g. whether they are as the source of Corded Ware Culture populations, and more. The pervasive presence of U mitochondrial haplogroups in some populations, and the temporal dynamics of the increasing presence of other mitochondrial variants, would be important to discuss. More importantly, while the authors spend substantial efforts into identifying proximal population sources for the reported admixture events, no attempts are made for estimating the respective dates of such events. As a result, admixture events are simply limited within the time boundaries defined by the individual sources. Yet, DATES modeling can be, and should be, used to more precisely obtain these temporal estimates. Such analyses would open to interesting narratives/interpretations, e.g. such as whether some ancestry sources contributed simultaneously to the descent of several population groups, as they migrated from the same location into different directions to meet and mix local populations of the way. In other words, whether there were temporal hotspots of admixture, versus gradual admixture in different regions over time. For example (but there are many more situations to investigate), whether the CHG contribution into all descending populations shown in Fig. 2c took place at the same time (suggesting one homogenous and fast migration wave in the region), or not (suggesting a gradual spread in the region, reaching some groups first, and other later).

Regarding qpADM modelling, I found the analyses well documented and very sound in most cases. There are a few cases, however, where the choice made by the authors should be clarified. For example, in Fig. 2a, Steppe_ENEO corresponds to a 45%/55% mixture of EHG and CHG. The more recent LateSteppe_ENEO (Fig. 2b) shows a 52%/48% mixture of the same source. An obvious modeling of LateSteppe_ENEO could be as the proximal Steppe_ENEO source mixing with a limited CHG contribution. The second scenario would probably imply a more recent mixing of CHG, hence, their survival for a bit longer and a second pulse of migration a bit later in the region. Following up on Fig. 3c, Catacomb and NCC populations are modelled again as mixtures of more distal sources, including Steppe_ENEO, but may as well have been modelled from more recent local groups, such as the LateSteppe_ENEO group discussed above. Another example was whether or not NCC could be a source (instead of Catacomb) for the formation of Lola, Arkhan, Srubnaya and Post-Catacomb groups. Same for NCC, and their contribution into Caucasus_MBA: could it be replaced by Catacomb?

A few typos should be fixed (e.g. line 117: 'Khvalysnk' is mis-spelled).

G. References: appropriate credit to previous work?

Yes.

H. Clarity and context: lucidity of abstract/summary, appropriateness of abstract, introduction and conclusions

The text is well-written, clear and relatively easy to follow, despite the complex cultural landscape and significant timeframe covered. The quality of the figures should be commended, as they (especially the various Sankey diagrams) considerably facilitate understanding of the evidence and modeling. Figure 1 should, however, clarify the geographic boundaries of the so-called Steppe, Intermediate and Caucasus ranges. I would expect that the modern extent of the Black Sea and the Caspian Sea may have changed due to past climatic conditions (e.g. the 4.2 KYBP aridification event). This may have opened new migration corridors worth considering, especially in periods of intensification of contact between the north and south sides of the mountain range. There are a few cases where the calendar years underlying some important sites/cultures (e.g. Lernakert) are not provided as results are presented. The authors should ensure to provide such information at first occurrence on relevant paragraphs. Lines 613-619 should be clarified, especially the underlying timelines of horse domestication. To the best of my knowledge, there is no evidence of horse-boosted mobility between the mid-4th and mid-3rd millennium BCE, either in Central Asia (Botai/Borly), or in the lower Don-Volga region (DOM2). The evidence available supports horse husbandry, milking and possibly husbandry, though this remains debated. It may be that Botai horses were used for long-distance mobility but this remains to be proven and cattle-driven wagons would have been equally likely to boost mobility at the time. It also seems that the ancestry cline shown in Fig. 2b comprises WSHG and Maykop, rather than Botai and Maykop, hence, the attempts to related the chain of population contacts identified and horse-based mobility appear rather loose.

Referee #2 (Remarks to the Author):

Review:

The authors present an important and well-executed study of the formation of late-prehistoric populations of the Caucasus and nearby 'steppe' region (between the Black Sea and North Caspian Sea). Overall the research execution is excellent, the findings are highly significant, and their conclusions are broadly merited, given the insights their complex genetic analysis provides, especially within the study zone. I also very much liked the graphics in this article, as they are an effective way of illustrating complex timing and admixtures of ancestry and are superior to pie charts, which somehow still remain common.

I do recommend the paper for publication, as it will provide an important benchmark in understanding the social and genetic dynamics of Western Eurasia and areas to the south and west.

However, I have a few small points of clarification and some thoughts for consideration that might improve the article on the broader interpretive perspective and which may allow this work to articulate with a wider range of archaeological models and data that are available outside of the genomics themselves. My points, corrections, concerns, and suggestions are itemized with page numbers for the author's consideration. Overall, these items are generally easy fixes or can be addressed through slightly more inclusive citations and a revised discussion throughout the text.

Line 98 – It might make sense to add Narasimhan et al. 2023 here (it is cited later), since this represents a comprehensive dataset on Eurasian genomics and pairs well with the statement. The current citation (Kohl) is a wonderful article for its time, but is quite out of date with regards to the genomics (if it even considers them), thus non-expert readers might be misled.

Line 137-138, and elsewhere. Throughout the article, the term "steppe" is used to describe geographic regions, ancestry, and regional communities. On line 137 I 'think' the authors mean the steppic region of the North Caucasus, but if misunderstood to mean the Eurasian "Steppe" then the area was far from "abandoned at 1700 BC". To avoid any misunderstanding, I would recommend clarifying which 'steppe' you mean in each case throughout the article since, as noted, Eurasia is a large and variable geography. I expect that simply using specific geographic locales and terms (North Caucasus, or Volga Basin) and then indicating these on the map is the clearest way forward in this regard. Indeed, many geographic locations throughout the so-called 'Eurasian Steppe' do not follow similar developmental paths – thus more specific geographic indications are helpful when we speak of possible regional processes. I found myself pausing and thinking at nearly every use of the word "steppe" to situate exactly where and what was being discussed.

In general, this also pertains to the genomic categories, since throughout the literature there are a range of interpretations of what is "steppe" ancestry and this paper seems to be adding yet another batch of genomic terms (e.g. Yamnaya_NC).

Line 365, Please ensure when you are referring to ancestry geographically situated in the north Caucasus steppe, for example, to clarify how different or similar it is to other steppe groups such as

“Steppe EMBA” (for Yamnaya/Afanasievo), Steppe -MLBA (for 2nd mill BC groups etc.). I think you are suggesting a nuanced differentiation within the Yamnaya cluster, but perhaps some additional clarification would be helpful here. Alas, I do wish there was a standard taxonomic standard for linking geographies to genomes, since at this point each team has their lexicon and not all are interchangeable and it is getting more confusing as to the variability within groups and regions.

Again, in this regard it will help to more clearly indicate which territories are considered steppe, intermediate etc. on Figure 1., since the dots are not color-coded according to the taxonomy within the map itself. Again, I would recommend a different term than steppe all-together. Simply North, Central and Southern Caucasus perhaps?

Line 182: The ‘intermediate group’ (which has a chronological range of 2200-1600 BCE (roughly) is implicated in the statement that this is a “dynamic phase of interaction (yes, agreed) resulting in the “establishment and intensification of pastoralism” in the Eurasian steppe. First, which steppe (again), but more importantly, it seems the latter portion of the sentence is quite incorrect based on existing archaeological knowledge. Intensification is a particular word in archaeology, and thus should be used critically. Elsewhere you state elsewhere that the Yamnaya were ‘100% pastoralists’ (line 634) at 3000 BCE (so “intensified” 1000 years before the ‘intermediate’ group) and there are many other sites across the local steppe (!) and broader steppes and mountain regions of Central / Inner Asia that reveal the establishment and consolidation of pastoralist lifeways in the 4th and 3rd mill. BCE (e.g. Frachetti 2012 article in Current Anthro, Hermes et al 2019 on early IAMC economies). In the Altai there is also strong evidence for the establishment of agro-pastoralism by 3000 BC, and fully diversified agro-pastoralist economies along what Frachetti calls the Inner Asian Mountain Corridor (IAMC) where by 2800 BCE there are abundant cases of established multi-domesticate pastoralism, totally unrelated (genetically or culturally) to Afanasievo (i.e. Yamnaya migration). Taylor et al 2022 also offer suggestive evidence that the IAMC may have been a path for pastoralism northward into eastern Eurasia even by the Neolithic. Indeed even at Botai, 500 years earlier than the Yamnaya, there was still quite an “intensive and established” pastoralist strategy at 3500 BCE, albeit horse-focused and not multi-animal. Please use care in making these broad conclusions, as they can obscure decades of important research on the regional forms of pastoralism that existed before the 2nd mill. BCE.

This also raises a question about the discussion of Eurasian pastoralism more generally throughout the article. In a range of places, the argument is made (to paraphrase) that the genetic formulation of the Yamnaya from a range of prior Eneolithic admixtures (***) which is an important insight from this paper**) is key for understanding the rise and consolidation of pastoralism across Central and Eastern Eurasia (specific lines listed below). However, this model is far from established fact in terms of the archaeological record, indeed much of today’s archaeological evidence does NOT support that process in terms of the economic interactions documented.

1) The Yamnaya migration to the east, according to many genetic papers (Allentoft et al., Narasimhan et al, Jeong et al, Zhang et al. Wang et al.) all demonstrate that the Afanasievo populations (ie. Steppe-EMBA) had an extremely *limited genetic impact on the regional populations during and immediately following the migration. A number of authors on the current paper are better aware of the genomics on this point.

2) What's more, the economic similarity between the Afanasievo and Yamnaya is not a one-to-one match. They exhibit fundamentally different herd structures, a total lack of grains among the Yamnaya (but evidence for limited use of grains in the Altai, ca. 3000-2800 BCE (see isotopic work by Chinese teams, as well as macro-botanical grains at Tongtian cave (Zhou et al.) - as well as goat/sheep genetics (see Hermes et al. 2021), all which point to economic links with the Inner Asian Mt. Corridor, but without significant human admixture. (Only limited Afanasievo admixture is found among some Chemurchek groups). So, the human genetics of the Afanasievo are indicative of a burst of Yamnaya population to the Altai without a huge genetic impact, while the two regional lifeways share only broad contours.

3) The preconditions for pastoralist "transformations" of Eurasia cannot yet be linked to the Yamnaya. Those Yamnaya populations who did migrate to the Altai appear to have fundamentally changed their modes of pastoralism, from open-steppe cattle herders (with some sheep) to vertical, seasonal mountain pastoralists relying heavily on sheep/goats (with fewer cattle). To my understanding, there remains a 3000 km gap in evidence that significantly questions the link between any "spread of pastoralism" across the Eurasian steppe to the episodic migration of Yamnaya groups to the Altai.

There are alternative models for the spread of pastoralism and the transformative role that domesticated economies had across Eurasia, some citations noted above. What is clear, however, is that it appears to be far more regionally punctuated and multi-directional than the model of simple derivation from Yamnaya herders, spreading eastward.

Line 336-338, 346,

This may be an overly reliant link to Botai. In this case, Botai may work as a site for modeling, but the presence of WSHG ancestry likely does not indicate a direct linkage with Botai itself. This is more likely the presence of a deeply ancient ancestry in the western Eurasian geographic orbit. There is little archaeology (to my knowledge) linking the Tobol basin to the north Caspian in the 4th mill. BC, if it does exist please cite the known archaeological ties. (At this time it would be with the Tersek cultures I imagine, who were regionally quite unique).

Also line 615-619: This is archaeologically not well supported. At this point (4th mill. BCE) the economies of these communities are completely different, and the WSHG ancestry is almost certainly far more geographically widespread to the borders of the north Caspian. Picking Botai - among the most unique of Eneolithic cultures - as a link is not well supported.

Line 619-621: Again, this is a vast overstatement of the wider Eurasian implications of 4th mill., local genomic processes within the study region. The sentence "...affinity of Steppe_Maykop to eastern groups seems to reflect the opening of the Eurasian Steppe....[when] horses, other technology spread...grassland adopted sheep, wheels and wagons, wool etc...". This is quite anachronistic and ignores a wealth of data that suggests a significantly different geographic range of communities and processes that shaped the diverse trajectories of regional pastoralism across the Eurasian steppe. E.g. Wheels are virtually unknown in the Eurasian steppe for nearly a 1000 years after the "Yamnaya migration". If this period was such an awakening of economy and technology spurred by the Yamnaya (or Pre-yamnaya even) why does it take over a millennium for these technologies to be evident across the Eurasian steppes? Many suggest it is not until the 2nd mill. BC that wheels make their way to the east, and horses perhaps even later. The diversity of sheep herding has already

been mentioned, and cannot be directly linked to the Caucasus. Thus, the sentence on 619-620, should be rectified to consider the alternative explanations that exist, the significant anachronisms, either discussed with the full weight of counter-arguments or simply reconsidered as a major wider impact of the research.

Line 628-630. "...combined innovations in the North Caucasus enabled the emergence of pastoralism and the dynamic modes of interaction, connectivity, and mobility that subsequently spread across larger geographic regions, bridging the greater Caucasus region, Pontic steppe, and Europe with lands further east in Central and Inner Asia"

Again, this is an over-glossed conclusion and archaeologically unsubstantiated statement. The modes of mobility evident in the steppe zones of the Caucasus around 3000 BC are completely different in a strategic sense than other documented pastoralists of the same time-period, and the "dynamic modes of interaction and connectivity" that emerged in the later BA had little if nothing to do with the Yamnaya mobility in the 3rd mill. BC., or only in the broadest deep time historical sense.

The formation of the 3rd mill. BCE Yamnaya pastoralist strategy DOES appear to have been important for Europe in the immediate afterglow, but this is far less so in the Eurasian steppe. The subsequent re-integration of admixed Yamnaya ancestry (admixed with Corded ware) does coincide with some technology changes, extending to roughly the Urals, ca. 2100 BCE. But this is 800-1000 years later. Beyond the Urals, the genetic percentages of admixed Yamnaya ancestry among 2nd mill BC Eurasian steppe groups is far less than the authors document in the Caucasus themselves and the populations east of the Urals appear far more integrated with local populations to the east and the south. This is made clear in Narasimhan et al, where over a span of ca. 1000 years of multi-regional dynamics there is admixed 10-40% Western Steppe ancestry to the east, hitting a max geographic range by ca. 1000 BC.

The old model of a wide arrow striking across the steppe and changing the economy, society and genomics is simply not scientifically evident – neither in the 3rd mill. BC with the initial Yamnaya migration, nor in the 2nd mill. BC. With the prolonged (and multidirectional) interactions of Bronze Age steppe groups. (It is sadly ironic, however, that Narasimhan et al. still felt compelled to draw such arrows, in spite of their own abundant genomic evidence that the process was much more intricate and multidirectional).

****This point has implications for the present article in so far as the authors make this paper about the formation of the communities that set the stage for transformative economies and societies of the Eurasian steppe and more. To be clear – I don't necessarily think this gesturing to the "sources of Eurasian pastoralism" throughout the article is particularly necessary. The well-documented social matrix that emerges from WITHIN the study zone is itself of major importance and worthy of publication.

However, if the authors are dedicated to making this point a central one, they must not rely on overly simplified narratives. Rather, please do get into the weeds of the argument or at least make the point that the field is much weedier than these decades' old models could reveal.

Line 646: What does sustainable, permanent and self-supporting mean in this context? Sustainable and permanent where? If the idea is that Yamnaya economy was resilient, then one might question why in the span of only 250 years, the basis of Yamnaya economy was fundamentally changed in almost all other regions where they migrated? E.g. 3000-2700 BCE in Europe, it was immediately integrated with farming. In the Altai, it was quickly transformed in terms of local ecology and seasonal mobility strategy, novel animal herd composition, and novel use of grains...By 2200 BCE whatever legacy of Yamnaya pastoralism was left was significantly transformed and adapted to local conditions and economic transformations (see point above about high regional diversity of herd composition and integrations (or not) of farming, as well as the complexity of admixtures that underly later Bronze Age Eurasian steppe). Please take care to reduce hyperbole and adhere to the archaeological evidence.

The points being made in this article about the formation of the Yamnaya as a genetic population are important, but the notion that the Yamnaya somehow transformed Central Eurasia are not backed by evidence. Whatever legacy of Yamnaya genetics that did underpin Eurasian populations came at or after 2000 BCE, and this was a protracted process of admixture taking place over a 1000+ years after their formation.

To this reader, the point made about the complex, multi-regional admixtures across the Caucasus around 4700-3500 BCE is itself a huge contribution, but the argument gets far less granular when it is linked to general narratives about the spread of pastoralism eastward across the steppe. At very least, the authors should engage with alternative models or take the opportunity to establish a line where the data presented provides clear insights and across which, the data does not offer concrete model testing.

The risk, as I see it, is linking data and solid genomics to broader processes that are only distantly related in archaeological terms, without delving into a thorough discussion of the actual data amassed now about the diversity of Bronze Age Eurasian pastoralism and regional ecologies (e.g. Bendrey 2011, Haruda 2019?).

Lines 682-692:

I do expect that the complex genomic admixtures within local and regional contexts helped innovate economic and social conditions and certainly impacted later populations – and those in the Caucasus are important. However, to argue that the admixture dynamics of the 5-3rd mill. Caucasus was the sole engine behind the transformation of economies and societies of the whole Eurasian steppe, is wildly overstated. Even in articles restricted by length, we should not fall into simplified conclusions without considering the breadth and diversity of both data and alternative explanatory models.

I highly support this article for publication (after considering these points) but I think that ultimately, the broader impacts of the 'prehistoric genomics of the Caucasus' offer a chance to evolve beyond the narrative of the "source" of all things Eurasian. At present, the data (both genomic and archaeological) do not directly support that conclusion in more than the broadest sense, but do allow for more nuanced and equally interesting narratives to emerge.

Referee #3 (Remarks to the Author):

There are 78 new radiocarbon dates in this paper, and as the primary publication of these data the paper should include all the information specified in the conventions for reporting radiocarbon determinations (Millard 2014).

No methods are described or cited for the production of the dates. No quality control measures, such as C/N ratios, %C, %N, or yields, are reported nor whether they were used to reject dates that are not published. $\delta^{13}\text{C}$ and $\delta^{15}\text{N}$ ratios are also routinely reported by radiocarbon labs and can be useful in identifying potential reservoir effects, but are not reported here. The quality of these dates cannot be judged unless all this information is included.

There are 139 radiocarbon dates reported in Supp Table A, but it is not clear which 78 of them are new. Nor is this clear in the text. For example, OxA and Hd dates must be from previous studies but no citations are given for them.

Why report “95.4% (2σ) calibration interval and median value” (Supp. p.4, Supp Table A) but plot mean (Figure 1b)? And in text give “ 1σ ” and “ 2σ ”? It should be made clear that the OxCal ‘Whole range’ option has been used rather than reporting all the separate sub-ranges. The ranges given are 68.3% and 95.4% ranges not 1σ and 2σ as the probability distributions are not normally distributed (see Millard 2014).

Dates reported in the text should be rounded out appropriately, probably to the next 10 years. The OxCal ‘Round by’ option will do this.

Supp p.6 “Modelling of radiocarbon dates from these graves result in a chronological sequence with a transitional period where both shaft and catacomb graves were used. This confirms earlier observations from neighbouring mounds²⁹ (Extended Data Fig. 8a).” However, fig 8a does not show any such chronological model, and in the cited reference I could only find mention of chronological modelling in the context of Rasshevatskiy 1, Russia.

Supp p.13 -14 calibrated age ranges are given without specifying the probability or σ .

Supp p. 14 “Grave 7 is a late Yamnaya grave interned into the first mounds hell” the final word must be wrong. ‘interned’ should be ‘interred’.

Supp p. 16 Lab-code ГИH: all lab codes should be in Latin letters. See https://radiocarbon.org/sites/default/files/2023-01/Labs-2023_01_11.pdf

Supp p. 18 interned → interred

Supp p. 22 ‘35272 σ ’ missing a space

Supp p.21 has Lab-Id #####: (including the highlight). The date of 3810 BP does not appear in Table A where the dating for this sample is said to be contextual.

Supp p.28: ‘fayence(?)’: is faience meant?

Supp p. 29 ‘40662 σ ’ missing a space

p.37-40 Marinskaya 5: I don’t understand dendrochronological dates with \pm . They do not follow a normal distribution, but are usually reported as a range when a sapwood estimate has been added to the last surviving ring.

p.39 MK5009 “dendro-dated 2644+13 BC” : + should be \pm

p.45 “This assemblage was not in atomic order” should be “This assemblage was not in anatomical order”

p. 47 SUS005 Context information should just be See above. The rest is repeated from above.

p.51 “half-desert” should be “semi-desert”

Table A

Three radiocarbon dated samples are missing the material: A19269.73.74, A19272, I2055

Why do lab numbers for dates appear in parentheses? Lab numbers (KIA-?) and (DAI) seem to be incomplete.

Table A, Date_Note: some of these are transparent, others obscure such as 'organics', 'hb', 'both highlighted', 'Remodeling for Marinskaya. Presumably 'hb' is human bone as in the text, but animal bones are not indicated.

Presumably 'source tissue' refers to the DNA extraction. Should we presume that the radiocarbon date came from the same sample unless indicated in Date_Note?

HD-29619

Referee #4 (Remarks to the Author):

The manuscript "The rise and transformation of Bronze Age pastoralists in the Caucasus" represents an archaeogenomic study from Caucasus. Authors have generated new genome-wide data for 131 ancient genomes (102 out of them were suitable for the analyses) spanning 6000 years, focusing on Bronze Age. New data analyzed together with published genomes from Caucasus and adjacent regions and/or relevant time periods allowed detailing the demographic processes in the region at the onset, during and decline of the Bronze Age.

The conclusions of the manuscript are based on the new data and are original. The new data contributes to our understanding of the Steppe ancestry formation which had a strong impact on genetic landscape of human populations across Eurasia. However, the study in the current format might rather be of significant interest for those specifically interested in the Caucasus prehistory: demographic events are analyzed within sub-periods (an array of overlapping periods of Eneolith-Bronza Age), there are numerous region-specific details, archaeological sites, outlier information etc. that might be difficult to follow for a wider audience.

Methodological section of the manuscript indicates that ancient DNA extraction/processing and contamination assessment followed requirements to work with highly degraded biomolecules hence suggest that data generated is reliable. Authors use established in human population genetics and archaeogenomics methods based on allele-frequencies (PCA, Admixture, f-statistics, qpAdm) and haplotype-based analyses (hapRoH).

I do not have any major comments, below are some minor things addressing them might improve the manuscript.

- Authors use Steppe and Caucasus clusters when describing the genetic patterns they observe within different periods they single out. It might be useful to specifically indicate who is part of which cluster according to them for each period as "clustering" based on PCA might be subjective and people tend to see different clusters.

- Readv2 – 3rd degree relationship should be treated with cautious.

- Line 117 – Khyvalynsk (typo?)

- Lines 440-450: why Lola ancestry is modelled using Steppe_Maykop individuals (it is rejected, and final model included Catacomb). From archaeological perspective, Catacomb would rather be the first choice for modelling genetic ancestry in Lola.

Figures

Figure 1: would benefit if the color codes between the panels a and b matched; the red color of the new genomes merges with the mountain color.

Figure 2: the Sankey diagrams look like a good choice for visualization of the ancestry modelling results.

Author Rebuttals to Initial Comments:

Referee #1 (Remarks to the Author):

We thank referee 1 for the positive and constructive feedback.

A. Summary of the key results

In this study, Ghalichi et al. present a new genetic data set for 131 ancient human individuals and 1.24M SNP positions. The individuals analyzed come from 38 excavation sites spread across the Caucasus range and spanning a time line of 600 years. Combined with previously published data in the region, the study charts through space and time the genetic diversity of the diverse human groups once inhabiting the region, and the complex history of admixture accompanying major changes in lifestyle (from hunting-gathering to pastoralism) and/or the development key technological innovations, including wheeled-based transportation and possibly equestrianism. The authors reveal that those Mesolithic groups located on both sides of the Caucasian Mountain range comprised different genetic origins. The following Eneolithic period saw the emergence of the so-called west Eurasian ancestry and increasing exchange between the mountain and steppe ecozones. Then the genetic landscape of the region relatively stable until the Late Bronze Age started.

B. Originality and significance: if not novel, please include reference

The study is very timely and unfolds the formation and migration history of several human groups that developed key cultural and technological innovations. These questions are of critical importance for understanding the migration history of Eurasia, given the long-standing influence that those groups had, including down to the present-day. The study also comes at the time when another study presents similar findings for population groups inhabiting the Caucasian range between the 4th and early 2nd millennium BCE.

C. Data & methodology: validity of approach, quality of data, quality of presentation

Overall, the methodology is very sound, and quite standard in ancient DNA research. It is based mostly on PCA visualization to help identify (i) shifting genetic profiles over time and/or cultures, and (ii) candidate population sources for admixture modelling. The filters and parameters used in PCA reconstruction, qpADM modelling, pruning and ADMIXTURE analyses are well accepted. The maximum filter requirements used for disregarding potentially contaminated individuals (5% based on nuclear patterns of sequence variation on the X autosome in males) is on the high end, but generally the individuals concerned and belonging to similar sites and/or cultures seem to co-cluster with these least contaminated. Therefore, those permissive filters seem to have limited impact if at all.

Yes, agreed. We have kept an eye on those samples that have elevated levels of contamination and monitored their behaviour in downstream analyses. Those that were excluded were flagged as such in the Supplementary Table A, columns BH-BJ.

D. Appropriate use of statistics and treatment of uncertainties

Yes.

E. Conclusions: robustness, validity, reliability

See section F. The authors contrast the number of pairs of related individuals identified in different time periods to speculate about changing social and/or funerary practices (e.g. line 522). Reporting N versus zero is, however, not sufficient to claim for statistical significance; appropriate statistical tests accounting for the different number of individuals genetically characterized in both time periods are required.

We agree in principle. However, the number of individuals in the two multi-phase kurgans is too small to allow for meaningful statistical hypothesis testing. A chi-squared test lacked statistical power to detect any statistical significance, and a beta regression approach yielded wide error bars, also indicating a lack of statistical power. Consequently, we are limited to cautious, descriptive inference, and we make no claims of statistical significance in the main text. To make this clearer, we have added the following statement to the Methods section:

“To investigate whether we had the statistical power required to perform rigorous hypothesis testing for a change in relatedness within multi-phase kurgans, we calculated the statistical power of a beta regression using the pwr package. Using the standard significance level of $\alpha=0.05$, and power of $\beta=0.8$, we can only reliably detect an effect size of $f^2=1.64$ (greater than the possible effect size of 1.0 for beta regression). Similarly, if we use a chi-squared test, we can only reliably detect an effect size of greater than 0.305, which is considered greater than a “medium effect size.”

F. Suggested improvements: experiments, data for possible revision

Analyses of uniparental variation are simple and deliver highly-credible haplogroup assignments. However, this information is only presented in Table SA (incidentally, with notes for the Y chromosome that should be polished or removed; see ‘column Y_notes’), and not discussed in the main text. Yet, some of the information presented is at the core of some debates about Yamnaya, e.g. whether they are as the source of Corded Ware Culture populations, and more. The pervasive presence of U mitochondrial haplogroups in some populations, and the temporal dynamics of the increasing presence of other mitochondrial variants, would be important to discuss.

Thank you for the suggestion. We have included a brief description of the Y chromosome haplogroup results as paragraph ‘4.6 Y-chromosome diversity through time’ in the Supplementary Information. We also include a summary plot of both uniparentally inherited marker systems in Extended Data Fig. 7 b and c. However, we fully agree that a more detailed description of the results is warranted and have added these now in the revised Supplementary Information ‘4.6 Y-chromosome diversity through time’ and ‘4.7 Mitochondrial haplogroup diversity through time’, as well as Supplementary Fig. S2.

We had noticed the comments being left in the table, too, and have removed them in the updated version.

More importantly, while the authors spend substantial efforts into identifying proximal population sources for the reported admixture events, no attempts are made for estimating the respective dates of such events. As a result, admixture events are simply limited within the time boundaries defined by the individual sources. Yet, DATES modeling can be, and should be, used to more precisely obtain these temporal estimates.

We agree that obtaining admixture dates as precisely as possible is important, but we caution that DATES is equally constrained by the respective sources, and can often be unreliable when proximal and more complex mixtures or common ancestries are involved. On the other hand, the estimated mixture dates of very distal sources can be irrelevant for the actual processes of admixture for those in question at later stages.

Such analyses would open to interesting narratives/interpretations, e.g. such as whether some ancestry sources contributed simultaneously to the descent of several population groups, as they migrated from the same location into different directions to meet and mix local populations of the way. In other words, whether there were temporal hotspots of admixture, versus gradual admixture in different regions over time. For example (but there are many more situations to investigate), whether the CHG contribution into all descending populations shown in Fig. 2c took place at the same time (suggesting one homogenous and fast migration wave in the region), or not (suggesting a gradual spread in the region, reaching some groups first, and other later).

We agree and have estimated the admixture date for the main clines, such as EHG + CHG north of the Caucasus, and Anatolia_N + CHG south of the Caucasus. Using the exact same sources as in qpadm (Fig. 2c and d) did not result in reliable date estimates (extremely wide error margins and low Z scores). However, following the model for steppe pastoralist used in the manuscript by the developers of DATES (Chintalapati et al. 2022, eLife, Figure 3) we substantially extended the number of individuals in the respective sources. For example, instead of only two CHG individuals, we also included Iranian Neolithic and Chalcolithic individuals (see Supplementary Table K). The same applies to the extended set of individuals representing EHG and Anatolia_N ancestries. This improved the estimates considerably.

We now include a new panel in Extended Data Figure 5c and integrate the results of the admixture date estimates in the main text and the discussion, albeit briefly given the length restrictions. The full set of analyses and individuals included as sources are given in Supplementary Table K.

Currently, only two immigration scenarios are evidenced by archaeological findings: (1) the immigration of North Mesopotamian populations at the beginning of the Neolithisation in the South Caucasus in the 6th millennium BC, and (2) the immigration of South Caucasian populations to the northern side of the Caucasus in the 5th millennium BC. We lack archaeological data for a more precise determination of the genetic profiles of the hunter-gatherer groups living in the North Caucasus prior to this. The fact that two genetically completely different groups can be identified at two sites that are only a few centuries and

approx. 250 kilometres apart (Kotias kldc & Satanaj grotto) suggests that this region was inhabited by more than just one hunter-gatherer population. This is also indicated by the recurrent genetic CHG signature in the formation of the steppe ancestry. Since it has no admixture of Neolithic components, it must be assumed that genetically unmixed groups persisted. We refer to both in lines 110-120 (archaeology), lines 189-201 (genetics) and address this difficult situation in the discussion in lines 551-570. We also address the question of the origin of different HG populations in the discussion, but have no evidence for any external immigration prior to the Neolithic due to a lack of archaeological finds.

Regarding qpADM modelling, I found the analyses well documented and very sound in most cases. There are a few cases, however, where the choice made by the authors should be clarified. For example, in Fig. 2a, Steppe_ENEO corresponds to a 45%/55% mixture of EHG and CHG. The more recent LateSteppe_ENEO (Fig. 2b) shows a 52%/48% mixture of the same source. An obvious modeling of LateSteppe_ENEO could be as the proximal Steppe_ENEO source mixing with a limited CHG contribution. The second scenario would probably imply a more recent mixing of CHG, hence, their survival for a bit longer and a second pulse of migration a bit later in the region.

Considering qpAdm results and their standard errors, we observe that both Steppe_Eneolithic and Late_Steppe_Eneolithic have very similar proportions of EHG and CHG ancestries. We tested whether the Late_Steppe_Eneolithic can be modelled with Steppe_Eneolithic as a single source but this model lacks strong support ($p=0.0179$; Supplementary Table M). However, since the CHG proportion in Late_Steppe_Eneolithic is lower, the limited CHG contribution can also be viewed as an additional EHG contribution in the later period which diluted the CHG proportion. To test this, we included a well-fit additional model for Late_Steppe_Eneolithic with Steppe_Eneolithic and EHG as a second source ($p=0.331896$, Supplementary Table M).

By and large, given the small EHG contribution, we interpret this rather as a gradual dilution than a second pulse of migration to the region.

Following up on Fig. 3c, Catacomb and NCC populations are modelled again as mixtures of more distal sources, including Steppe_ENEO, but may as well have been modelled from more recent local groups, such as the LateSteppe_ENEO group discussed above.

We have tested both distal and proximal sources which included Late_Steppe_Eneolithic as one of the sources for various two-way models (lines 373-380, Supplementary Table O). Both NCC and Catacomb can be modelled successfully with preceding Yamnaya groups from the North Caucasus, Samara region and Ukraine. As mentioned in the main text (lines 406-408), NCC can be modelled with Ukraine_EBA_Yamnaya, while Catacomb can be modelled with Yamnaya_NC.

Another example was whether or not NCC could be a source (instead of Catacomb) for the formation of Lola, Arkhan, Srubnaya and Post-Catacomb groups. Same for NCC, and their contribution into Caucasus_MBA: could it be replaced by Catacomb?

Yes, NCC and Catacomb are genetically near-identical and can be used interchangeably as sources in ancestry modelling, as shown in Supplementary Table Q. This is also stated in the main text (e.g, lines 454-456). We have now also included models in which we rotate one of each to the outgroups, respectively, to test whether this leads to model rejection.

However, since this is not the case for both, we conclude that NCC and Catacomb descend from the same population, but exhibit archaeologically distinct material cultures.

For the purpose of illustration in the Sankey plots, we preferred to display Catacomb as the proximal sources on the basis of cultural similarities in burial practices.

A few typos should be fixed (e.g. line 117: 'Khvalysnk' is mis-spelled).

Thanks. This is fixed.

G. References: appropriate credit to previous work?

Yes.

H. Clarity and context: lucidity of abstract/summary, appropriateness of abstract, introduction and conclusions

The text is well-written, clear and relatively easy to follow, despite the complex cultural landscape and significant timeframe covered. The quality of the figures should be commended, as they (especially the various Sankey diagrams) considerably facilitate understanding of the evidence and modeling.

Thank you. Much appreciated.

Figure 1 should, however, clarify the geographic boundaries of the so-called Steppe, Intermediate and Caucasus ranges.

Thank you. We have frequently used maps showing ecozones of the Caucasus region in publications of our team (e.g. Wang et al. 2019; Knipper et al. 2020, Reinhold 2024). We had considered using such a version in Figure 1, but decided against it for aesthetic reasons, because the map is already very busy and has been revised to improve the links between sites and genetic ancestries.

However, we have added a map with the eco-regions according to Dinerstein et al. 2017 as Supplementary Figure S1 in the respective chapter 2 Archaeological site and sample descriptions in the Supplementary Information.

I would expect that the modern extent of the Black Sea and the Caspian Sea may have changed due to past climatic conditions (e.g. the 4.2 KYBP aridification event). This may have opened new migration corridors worth considering, especially in periods of intensification of contact between the north and south sides of the mountain range.

Thank you for raising this point. We are aware of recent work on the shoreline fluctuations of both the Black Sea (e.g. Kelterbaum et al. 2015, Laermanns et al. 2019) and the Caspian Sea (Krooneberg et al. 2008, Oliver et al. 2015, Bezrodnykh et al. 2020 or recently subsumed in Leroy et al. 2022). These fluctuations have continuously reshaped coastlines during the Holocene. As such, it is not possible to select one shoreline that is applicable to the entire time transect presented in our study. Thus, we decided to keep the modern-day shorelines as a reference that most readers would be familiar with. However, even though the width of the shoreline corridors fluctuated periodically, the even distribution of sites along

both sides of the inner mountain ranges (from east to west) shows that the mountains themselves were not a significant barrier to human movement. Similarly, climatic fluctuations as observed in the pollen record indicate that the usage of routes across the mountains did not change during these times (see Connor et al. 2009). The archaeological sites also suggest that many passes documented in use by shepherds in the 19th century (Merzbacher 1901) were already in use in prehistoric times. Thus, while shoreline fluctuations occurred through time, we do not find evidence that this fluctuation substantially affected migration or mobility through the Caucasus during the time periods of our study.

There are a few cases where the calendar years underlying some important sites/cultures (e.g. Lernakert) are not provided as results are presented. The authors should ensure to provide such information at first occurrence on relevant paragraphs.

Thank you for bringing this to our attention. We went through the manuscript carefully and added calendar years for sites and cultures where appropriate, including in Armenia (e.g. Lernakert) and Iran.

Lines 613-619 should be clarified, especially the underlying timelines of horse domestication. To the best of my knowledge, there is no evidence of horse-boasted mobility between the mid-4th and mid-3rd millennium BCE, either in Central Asia (Botai/Borly), or in the lower Don-Volga region (DOM2). The evidence available supports horse husbandry, milking and possibly husbandry, though this remains debated. It may be that Botai horses were used for long-distance mobility but this remains to be proven and cattle-driven wagons would have been equally likely to boost mobility at the time. It also seems that the ancestry cline shown in Fig. 2b comprises WSHG and Maykop, rather than Botai and Maykop, hence, the attempts to related the chain of population contacts identified and horse-based mobility appear rather loose.

Thank you for raising this point, and it was not our intent to over-emphasize horses. We see where the misunderstanding might originate and have add further clarification to the paragraph (highlighted in yellow), now reading in lines 623-634:

“At this time, innovations such as cattle-drawn wheeled transport and initial steps towards horse domestication gradually boosted mobility and herd management, respectively^{15,16}. In this regard, the clockwise tilt in the admixture cline of Late_Steppe_Eneolithic individuals from an EHG/CHG axis to an WSHG/Maykop axis is intriguing, as it encompasses Steppe_Maykop individuals who carry additional WSHG ancestry, an ancestry that is also found at Botai in Central Asia, another area of incipient equid domestication^{16,34,51}. The genetic affinity of the Steppe_Maykop to such eastern groups seems to reflect the opening of the Eurasian steppe as a habitat and a communication space, even though this link is enigmatic and not related to any archaeological phenomenon known from this epoch and region. Besides emerging horse husbandry, other technological innovations started to spread, such as grassland-adapted sheep for dairying², wheels and wagons^{15,19}, and possibly wool as a material for insulating clothes and mobile architecture⁵².”

We have not made any strong claims about horse-based mobility and importantly, at no point did we intend to relate our observations to mobility via horse riding alone. This is just one of the arguments that may have gradually led to greater mobility in the grasslands. However, it remains undeniable that the horse will have played a crucial role

in herd management, facilitating the range expansions that become visible during the 3rd millennium BC in the archaeological as well as the archaeological record.

Evidence for cattle-drawn mobility during the Maykop period and later has been discussed intensively by members of our team (e.g. Reinhold, S. et al. 2017; S. Hansen & F. Klimscha, Digital Atlas of Innovation, <https://atlas-innovations.de/de/wheeled-vehicles>). Furthermore, the earliest of the candidate precursor forms of the DOM2 lineage (BZNK1002x4_Rus_m3450; Librado et al. 2021) was contributed by members of our team and comes from Steppe Maykop-associated contexts at the site Aygurskiy 2, which is also represented in our manuscript (AY2004, mound 22, grave 16). We are aware that the increased mobility associated with horses did not become relevant before the end of the 3rd millennium BC (Librado et al. 2024). Nevertheless, we found the observed tilt to ANE-rich groups, of which Botai in Kazakhstan is also one, worth mentioning, not only because Botai were experimenting with horses (Librado et al. 2021, Librado et al. 2024), but also because the affinity of populations north of the Caucasus, the Caspian Sea, and the western parts of central Asia is archaeologically elusive and not well understood or studied yet. As such, and while being admittedly loose, this observation still presents a notion worth considering in future studies.

Referee #2 (Remarks to the Author):

Review:

The authors present an important and well-executed study of the formation of late-prehistoric populations of the Caucasus and nearby 'steppe' region (between the Black Sea and North Caspian Sea). Overall the research execution is excellent, the findings are highly significant, and their conclusions are broadly merited, given the insights their complex genetic analysis provides, especially within the study zone. I also very much liked the graphics in this article, as they are an effective way of illustrating complex timing and admixtures of ancestry and are superior to pie charts, which somehow still remain common.

I do recommend the paper for publication, as it will provide an important benchmark in understanding the social and genetic dynamics of Western Eurasia and areas to the south and west.

We thank the reviewer for the positive evaluation and detailed feedback!

We do wish to note, however, that it seems from the points raised below that parts of our manuscript were confused with arguments made in two other studies that are currently available as pre-prints (Lazaridis et al.

<https://www.biorxiv.org/content/10.1101/2024.04.17.589597v1>; and Nikitin et al.

<https://www.biorxiv.org/content/10.1101/2024.04.17.589600v1>), both of which have a much

more pronounced emphasis on the Yamnaya phenomenon specifically and the Eurasian steppe zone generally.

We would like to stress that our manuscript focuses in detail on the Caucasus and its interface with the Eurasian steppe region, and is intended to provide a high-resolution genetic history of this key interaction zone spanning 6,000 years of time. Our study is less focused/fixed on the archaeological cultures of the 3rd millennium BC and instead attempts to take a more holistic approach to understanding the emergence, establishment, and transformation of pastoralist groups in the western part of the Eurasian steppe and neighbouring mountains (see also Reinhold 2024).

However, I have a few small points of clarification and some thoughts for consideration that might improve the article on the broader interpretive perspective and which may allow this work to articulate with a wider range of archaeological models and data that are available outside of the genomics themselves. My points, corrections, concerns, and suggestions are itemized with page numbers for the author's consideration. Overall, these items are generally easy fixes or can be addressed through slightly more inclusive citations and a revised discussion throughout the text.

Line 98 – It might make sense to add Narasimhan et al. 2023 here (it is cited later), since this represents a comprehensive dataset on Eurasian genomics and pairs well with the statement. The current citation (Kohl) is a wonderful article for its time, but is quite out of date with regards to the genomics (if it even considers them), thus non-expert readers might be misled.

We added Narasimhan et al. 2019, Science, but also kept Kohl as a critical archaeology reference because it provides important information not included in the genetics study.

Line 137-138, and elsewhere. Throughout the article, the term “steppe” is used to describe geographic regions, ancestry, and regional communities. On line 137 I ‘think’ the authors mean the steppic region of the North Caucasus, but if misunderstood to mean the Eurasian “Steppe” then the area was far from “abandoned at 1700 BC”. To avoid any misunderstanding, I would recommend clarifying which ‘steppe’ you mean in each case throughout the article since, as noted, Eurasia is a large and variable geography. I expect that simply using specific geographic locales and terms (North Caucasus, or Volga Basin) and then indicating these on the map is the clearest way forward in this regard. Indeed, many geographic locations throughout the so-called ‘Eurasian Steppe’ do not follow similar developmental paths – thus more specific geographic indications are helpful when we speak of possible regional processes. I found myself pausing and thinking at nearly every use of the word “steppe” to situate exactly where and what was being discussed.

Yes, this is correct. We have changed the sentence in lines 137-138 as follows:

“However, the lasting effects of overexploitation and climatic stress resulted in a decline of sites and the abandonment of the steppe zone between the North Caucasus and the Lower Don and Volga rivers after 1700 BC.”

We are aware of the problems associated with the terminology of 'steppe' as an ambiguous term for both an ecological biome and a cultural space, as well as the approach taken by the reviewer in 2012 (Frachetti 2012). The terms 'steppe' or 'Caucasus/Caucasia' are in some respects metaphors for large areas of interaction that encompass far more than the

geographical terms or even the biome that defines them (see Reinhold 2024). To clarify this, we have now added a map in the Supplementary Information (Supplementary Fig. S1) that links our studied sites to the steppe biome as an ecozone, based on the classification of Dinerstein et al. 2017 and the resulting maps from <https://ecoregions.appspot.com/>.

We agree that the use of the term 'steppe' or 'Eurasian steppe' requires careful consideration and, if necessary, geographical clarification. We have now reconsidered at each phrase whether such a delimitation is necessary or whether the statements do not apply to the entire area of the Eurasian steppe in total.

In general, this also pertains to the genomic categories, since throughout the literature there are a range of interpretations of what is “steppe” ancestry and this paper seems to be adding yet another batch of genomic terms (e.g. Yamnaya_NC).

‘Steppe-related’ ancestry has been previously defined as a near equal mix of Eastern European hunter-gatherer (EHG) and Caucasus hunter-gatherer (CHG) ancestry. This ancestry mix was formed during the Eneolithic and persisted throughout the Bronze Age in the steppe and forest steppe zones of western and central Eurasia (Haak et al. 2015, Allentoft et al. 2015; 2024, Jones et al. 2015, Mathieson et al. 2015; 2018, Narasimhan et al. 2019, Wang et al. 2019, among others). With Yamnaya_North_Caucasus (Yamnaya_NC) we label a group of Yamnaya-associated individuals from our study region for the simple purpose of comparative analyses. This is common practice and not intended to inflate the number of genomic terms.

We now address this issue of mixing genetic and archaeological terms in SI section 1.3 “Archaeological Cultures and Genetic Groupings”. We are unable to solve the problem of previously established terms, but we have proposed a practicable compromise.

Line 365, Please ensure when you are referring to ancestry geographically situated in the north Caucasus steppe, for example, to clarify how different or similar it is to other steppe groups such as “Steppe EMBA”(for Yamnaya/Afanasievo), Steppe -MLBA (for 2nd mill BC groups etc.). I think you are suggesting a nuanced differentiation within the Yamnaya cluster, but perhaps some additional clarification would be helpful here.

We show this in Fig. 3 (where the new data and the published EMBA populations are shown), as well as in the subsequent sentence: “...who also fall on the EHG-CHG cline of the Steppe groups in PC space (Fig. 3a, Extended Data Fig. 1). They form a tight cluster with published individuals from the Black Sea, Samara, and North Caucasus regions, and represent a mixture of the two distal ancestry sources^{4,5} (Supplementary Table N).” We also added a reference to Extended Data Fig. 1 and note the overlap of EMBA and MLBA individuals. For more nuanced proximal qpadm models we refer to Supplementary Table O (see lines 375-385 in the main text).

Alas, I do wish there was a standard taxonomic standard for linking geographies to genomes, since at this point each team has their lexicon and not all are interchangeable and it is getting more confusing as to the variability within groups and regions.

We agree in part, but want to acknowledge that the majority of colleagues in our field do agree on most of these terms. It is difficult to achieve complete consistency and agreement across multiple research teams in a fast-paced field, and the same criticism could be made about the naming of archaeological cultures/complexes and groups: the terminologies often

differ between regions, state borders, and/or schools of thought despite addressing highly similar phenomena. However, we do recognize that this is an important issue for interoperability between studies, and we now address these issues of terminology in detail in SI chapter 1.3.

Again, in this regard it will help to more clearly indicate which territories are considered steppe, intermediate etc. on Figure 1., since the dots are not color-coded according to the taxonomy within the map itself. Again, I would recommend a different term than steppe all-together. Simply North, Central and Southern Caucasus perhaps?

We would like to clarify that the groupings *Steppe*, *Caucasus*, and *Intermediate* (denoted in Italics) describe genomic clusters that are introduced and described in lines 179-184 on page 7, and are not territories. The naming of these clusters was introduced in Wang et al. 2019, and for reasons also laid out in the critique above, we would like to keep the naming scheme and not introduce a new label that could cause further confusion.

Of note, we revised Figure 1 in which the sampled sites are now colour-coded according to the chronology in panel B and consistent with all other figures and legends. Further, we now also include another map as Supplementary Figure S1, which shows the location of the sampled sites in the ecozones of the wider Caucasus region.

Line 182: The 'intermediate group' (which has a chronological range of 2200-1600 BCE (roughly) is implicated in the statement that this is a "dynamic phase of interaction (yes, agreed) resulting in the "establishment and intensification of pastoralism" in the Eurasian steppe. First, which steppe (again), but more importantly, it seems the latter portion of the sentence is quite incorrect based on existing archaeological knowledge.

There appears to be a conceptual misunderstanding. The individuals labelled '*Intermediate*' carry ancestry that is a mix of *Steppe*-related and *Caucasus*-related ancestries. Importantly, they are neither a cohesive genetic group (note that we use the plural 'groups') nor do they reflect a single temporal horizon. As shown in the chronology in Figure 1b, individuals with mixed (i.e. intermediate) ancestry occur repeatedly at various points in time in our transect, covering a range from 4800 to 1600 BC. They represent individuals whose mixed genetic profiles provide evidence of contact between the otherwise genetically distinct populations. For this reason, they are of great importance to our reasoning. As a consequence of the observation of mixed ancestries, resulting from contact and gene flow between groups from the steppe zone north of the Caucasus and those from the mountainous regions of the Caucasus in the mid-5th millennium BC – a time when we observe the emergence of pastoral dairy technology in the region for the first time (Scott et al. 2022) – we feel confident in claiming that this contact also facilitated the transfer of cultural knowledge related to pastoralist activities.

Intensification is a particular word in archaeology, and thus should be used critically. We agree and have changed the term 'intensification' to 'spread'.

Elsewhere you state elsewhere that the Yamnaya were '100% pastoralists' (line 634) at 3000 BCE (so "intensified" 1000 years before the 'intermediate' group) and there are many other sites across the local steppe (!) and broader steppes and mountain regions of Central / Inner Asia that reveal the establishment and consolidation of pastoralist lifeways in the 4th

and 3rd mill. BCE (e.g. Frachetti 2012 article in Current Anthro, Hermes et al 2019 on early IAMC economies). In the Altai there is also strong evidence for the establishment of agro-pastoralism by 3000 BC, and fully diversified agro-pastoralist economies along what Frachetti calls the Inner Asian Mountain Corridor (IAMC) where by 2800 BCE there are abundant cases of established multi-domesticate pastoralism, totally unrelated (genetically or culturally) to Afanasievo (i.e. Yamnaya migration). Taylor et al 2022 also offer suggestive evidence that the IAMC may have been a path for pastoralism northward into eastern Eurasia even by the Neolithic. Indeed even at Botai, 500 years earlier than the Yamnaya, there was still quite an "intensive and established" pastoralist strategy at 3500 BCE, albeit horse-focused and not multi-animal. Please use care in making these broad conclusions, as they can obscure decades of important research on the regional forms of pastoralism that existed before the 2nd mill. BCE.

We are unable to find a reference to the quantification of Yamnaya subsistence in our manuscript at line 634 or elsewhere. To avoid confusion, we have removed the term intensification from our discussion of the spread and adoption of pastoralism. We agree that the development of pastoralism in the Eurasian steppes was multifactorial and proceeded along different regional trajectories. Overall, this comment seems to arise from a misunderstanding that an intermediate group "intensified" pastoralism after 2200 BC. This is not what we say in the text. Our reasoning in the paragraphs about the 'Eneolithic' (page 9) and 'Late Eneolithic and Early Bronze Age (page 11), as well as the developments depicted in Figure 2 describe the emergence, gradual establishment and consolidation of populations for which we were able to establish in previous isotope and proteomic studies (Knipper et al. 2018, Knipper et al. 2020, Scott et al. 2022) that their way of life was based on a (more or less) mobile pastoral economy. In parallel to the developments along the IAMC (Hermes et al. 2019, Taylor et al. 2021) these developments paved the way for the full exploitation of the Eurasian steppe habitat during the 3rd millennium BC. However, this broader macroregional story is not one that we address in our paper. Focusing instead on events in the greater Caucasus region specifically, we examine one of the scenarios in which cultural ties between groups with a pastoral dairy economy (as evidenced by proteomics and stable isotope analysis) coalesce and lead to a trajectory for the spread of pastoralism across the steppe. However, importantly, we do not claim that this was the only route.

Importantly, the intermediate individuals with mixed ancestry from the time period after 2200-1200 BC demarcate the tail end of the Bronze Age peak era for mobile pastoralism, as we are alluding to in the paragraph 'Final Middle and Late Bronze Age' (page 15). This interaction resulted in the establishment of highly productive stationary pastoral economies, a process that can also be observed more broadly in the Eurasian steppe region (e.g. among Sintashta, Andronovo, etc.). The early genetic observations regarding the Sintashta by Allentoft et al. 2015 suggest similar consolidation processes there, but this also goes far beyond the scope of the article.

This also raises a question about the discussion of Eurasian pastoralism more generally throughout the article. In a range of places, the argument is made (to paraphrase) that the genetic formulation of the Yamnaya from a range of prior Eneolithic admixtures (**which is an important insight from this paper**) is key for understanding the rise and consolidation of pastoralism across Central and Eastern Eurasia (specific lines listed below). However, this model is far from established fact in terms of the archaeological record, indeed much of

today's archaeological evidence does NOT support that process in terms of the economic interactions documented.

To clarify this point, our manuscript describes “the formation of the characteristic Western Eurasian steppe ancestry” (abstract, lines 77-80), which is fully in accordance with the archaeological literature of the Caucasus and the North Pontic region (Reinhold et al. 2024; Scott et al. 2022). Importantly, *Steppe*-related ancestry is not a synonym for Yamnaya, as its formation predates the latter by at least 1000-2000 years. Regarding the rise and consolidation of pastoralism across Central and Eastern Eurasia, the eastern movement or non-movement of Yamnaya groups is not the focus of this article.

1) The Yamnaya migration to the east, according to many genetic papers (Allentoft et al., Narasimhan et al, Jeong et al, Zhang et al. Wang et al.) all demonstrate that the Afanasievo populations (ie. Steppe-EMBA) had an extremely *limited genetic impact on the regional populations during and immediately following the migration. A number of authors on the current paper are better aware of the genomics on this point.

Yes, this is correct. However, the fact that Afanasievo-associated individuals from the Altai region carry allochthonous, *Steppe*-related ancestry is indicative of the dynamic expansions of genetically-defined EMBA steppe groups to the east. Whether or not they left an enduring archaeological impact in these eastern regions is a separate question. The same applies to contact zones in the west, i.e. southeastern Europe.

2) What's more, the economic similarity between the Afanasievo and Yamnaya is not a one-to-one match. They exhibit fundamentally different herd structures, a total lack of grains among the Yamnaya (but evidence for limited use of grains in the Altai, ca. 3000-2800 BCE (see isotopic work by Chinese teams, as well as macro-botanical grains at Tongtian cave (Zhou et al.) - as well as goat/sheep genetics (see Hermes et al. 2021), all which point to economic links with the Inner Asian Mt. Corridor, but without significant human admixture. (Only limited Afanasievo admixture is found among some Chemurchek groups). So, the human genetics of the Afanasievo are indicative of a burst of Yamnaya population to the Altai without a huge genetic impact, while the two regional lifeways share only broad contours.

This is correct, but the topic falls outside the scope of our manuscript. We fully agree that the archaeological arguments linking Afanasievo and Yamnaya are weak, but this is likely due in part to the fact that the Afanasievo were a small population that was rapidly adapting to new environments and emerging local needs, as well as interacting with new cultural groups. In such a scenario, it is not surprising that the Yamnaya and Afanasievo show strong genetic links but weaker cultural links. Furthermore, as there are almost no Yamnaya animal bone assemblages in the core region of this group, economic arguments about herd structures cannot be directly substantiated. Subsistence-related Yamnaya animal bone assemblages comparable to those found at the Afanasievo sites in the Altai only exist for the 5th millennium BC (e.g. Vybronov et al. 2015: Caspi sites; Morgunova et al. 2018: Turganik) and, with a few exceptions, for the early 4th millennium BC (see overview by Kaiser 2019), if Repin Chutor is assigned to Yamnaya.

3) The preconditions for pastoralist "transformations" of Eurasia cannot yet be linked to the Yamnaya. Those Yamnaya populations who did migrate to the Altai appear to have fundamentally changed their modes of pastoralism, from open-steppe cattle herders (with

some sheep) to vertical, seasonal mountain pastoralists relying heavily on sheep/goats (with fewer cattle). To my understanding, there remains a 3000 km gap in evidence that significantly questions the link between any “spread of pastoralism” across the Eurasian steppe to the episodic migration of Yamnaya groups to the Altai.

There seems to be misunderstanding that pastoralist transformation on the Eurasian steppe is entirely linked to the Yamnaya period, but this is not what we argue in our paper, and events occurring in the Altai region are not the focus of our study. In a recent dietary proteomics study by our group, we investigated dairy pastoralist diets in the North Caucasus and found no evidence that Yamnaya diets were specifically cattle-focused. Instead, we found that the diets of all studied steppe pastoralist groups in the North Caucasus prior to ca. 2800 BC (e.g., Maykop, Early Yamnaya) focused heavily on sheep dairying, whereas groups after ca. 2800 BC (Late Yamnaya, NCC, Catacomb) developed a more diversified form of dairy pastoralism that included sheep, cattle, and goat milking (Scott et al. 2022). Thus, while we did observe a dietary shift among pastoralists, it did not occur with the emergence of Yamnaya groups, but rather occurred simultaneously among many different pastoralist groups during the mid-3rd millennium BC. Critically, sheep were important livestock for all of the pastoralists in our study, including Yamnaya groups.

In addition, our new genomic study tries to move away from simplistic prescripts in showing that the transition to pastoralism in the Eurasian steppe emerged from innovations in transport, herding, and dairying practices. We explicitly avoid narrowly attributing these innovations to the Yamnaya groups and the hypothesis of their spread as a simplistic migration scenario. We deliberately use terms such as 'emergence' or 'spread', which can be conceptualised both as a transfer in the form of migration/expansion and as a transfer of knowledge or cultural practices. In fact, the aim of the discussion in lines 615-622 is to use the example of the Maykop groups to show that technological innovations can also be mediated primarily by the transfer of knowledge and with only limited or no gene flow.

What we conclude, based on the synthesis of previous studies and the present genetic study, is that the emergence of mobile pastoral economies began at the northern frontier of the Caucasus in the mid-/late-5th millennium BC and spread from there to the west and east. The beginning is thus at least 1000 years earlier than the start of the Yamnaya phenomenon. This timing also fits quite well with the hypothesis of Nobert Benecke (2017), who suggests a rapid replacement of the Botai-horse economy with an economy based on Western domesticates for Central Asia in the late 4th millennium BC.

There are alternative models for the spread of pastoralism and the transformative role that domesticated economies had across Eurasia, some citations noted above. What is clear, however, is that it appears to be far more regionally punctuated and multi-directional than the model of simple derivation from Yamnaya herders, spreading eastward.

As stated above, we do not address the eastward spread of Yamnaya-related groups. By contrast, the main focus and novelty of our manuscript revolves around the genetic transformations and innovation horizons of the preceding Eneolithic periods of the 5th and 4th millennium BC in the Caucasus region and the Western Eurasian steppe zone.

Line 336-338, 346,

This may be an overly reliant link to Botai. In this case, Botai may work as a site for modeling, but the presence of WSHG ancestry likely does not indicate a direct linkage with

Botai itself. This is more likely the presence of a deeply ancient ancestry in the western Eurasian geographic orbit. There is little archaeology (to my knowledge) linking the Tobol basin to the north Caspian in the 4th mill. BC, if it does exist please cite the known archaeological ties. (At this time it would be with the Tersek cultures I imagine, who were regionally quite unique).

Also line 615-619: This is archaeologically not well supported. At this point (4th mill. BCE) the economies of these communities are completely different, and the WSHG ancestry is almost certainly far more geographically widespread to the borders of the north Caspian. Picking Botai - among the most unique of Eneolithic cultures – as a link is not well supported.

We address both points raised above in conjunction and added in line 630-631 “even though this link is enigmatic and currently not related to any archaeological phenomenon known from this epoch and region“ to refer to the uncertainties of the archaeological record.

We currently have no evidence for the distribution of WSHG ancestry during the 4th millennium BC. Botai are used as a genetic proxy in our modelling and are certainly not the direct or primary source. As with most genetic ancestry modelling it is likely that an as yet unsampled, geographically more proximal source is an equally good or even better fit. Importantly, we do not postulate an archaeological link between Steppe Maykop and Botai, but merely state the genetic affinity to regions northeast of Steppe Maykop are archaeologically less well understood and thus remain enigmatic.

However, what we find intriguing is that both Steppe Maykop and Botai represent early, independent attempts of horse management/domestication, which suggest that the regions between the two (and also beyond) likely also harboured Eneolithic groups with similar intention and/or potential. As such, despite no current archaeological support, the region appears understudied and we thus point to a possible avenue of future investigation.

Line 619-621: Again, this is a vast overstatement of the wider Eurasian implications of 4th mill., local genomic processes within the study region. The sentence "...affinity of Steppe_Maykop to eastern groups seems to reflect the opening of the Eurasian Steppe....[when] horses, other technology spread...grassland adopted sheep, wheels and wagons, wool etc...". This is quite anachronistic and ignores a wealth of data that suggests a significantly different geographic range of communities and processes that shaped the diverse trajectories of regional pastoralism across the Eurasian steppe. E.g. Wheels are virtually unknown in the Eurasian steppe for nearly a 1000 years after the “Yamnaya migration”. If this period was such an awakening of economy and technology spurred by the Yamnaya (or Pre-yamnaya even) why does it take over a millennium for these technologies to be evident across the Eurasian steppes? Many suggest it is not until the 2nd mill. BC that wheels make their way to the east, and horses perhaps even later. The diversity of sheep herding has already been mentioned, and cannot be directly linked to the Caucasus. Thus, the sentence on 619-620, should be rectified to consider the alternative explanations that exist, the significant anachronisms, either discussed with the full weight of counter-arguments or simply reconsidered as a major wider impact of the research.

Here we must respectfully refute the allegation of overstatement and anachronism. The findings in lines 618-620 and 623-628 are well documented with references to specific studies. The literature cited unfolds the vectors of the spread of the technologies mentioned.

In our recent proteomics study of subsistence in the Caucasus region (Scott et al. 2022), we show a long-standing practice of sheep-focused dairy pastoralism, including among the early Yamnaya until ca. 2800 BC, overturning previous views that they were more cattle-focused (a mistaken view that had been based on very limited zooarchaeological evidence).

Covering the period of the 5th-3rd millennium BC, Scott et al. 2022 and Wilkin et al. 2021 further document the spread of dairy technologies from the Caucasus to the southern Urals. While the Caucasus data revealed sheep dairy product consumption as early as the late-5th and 4th mill. BC, the data from Khvalynsk and other Caspian (Vybronov et al. 2018) and Black Sea (Mileto et al. 2019) region sites failed to detect dairy products. Only in the 3rd millennium BC is such evidence positive for these regions, which provides a chronological vector. For the spread of wagonry to the west, see Burmeister 2019 or the Digital Atlas of Innovations (<https://atlas-innovations.de/en/>). To the east, the remains of a wagon chase with crossed floorboards was identified at Kurak gobi 1 (2876–2482 calBC, Kovalev, Erdenebaatar 2010) and thus provides evidence of a technical link to the North Caucasus in the early 3rd millennium BC. For comparisons see Gey 2009. The spread of metal to the west in the 5th-3rd millennium BC is documented by Hansen 2021, to the east in the early 3rd millennium BC by Kusnecov 2009 and others. It is correct that the spread of some of these innovations from the Caucasus to the Altai or Mongolia may have taken 1000 years or more. However, this should not come as a surprise as these are culturally managed adaptation processes. The much earlier timing not only loosens the ties between the transfer of innovations and the Yamnaya groups as (sole) transporters (which are vastly overemphasised by some scholars in our opinion) but also adds more temporal flexibility to the Eurasian perspective.

Since our paper cannot possibly discuss the spread of technologies and their archaeological evidence in all detail (but rather is limited to only providing a population-historical context), we also cannot list all the necessary citations. A fuller discussion of these archaeological processes is provided in Reinhold 2024, and this source is cited where relevant multiple times in the text.

Line 628-630. "...combined innovations in the North Caucasus enabled the emergence of pastoralism and the dynamic modes of interaction, connectivity, and mobility that subsequently spread across larger geographic regions, bridging the greater Caucasus region, Pontic steppe, and Europe with lands further east in Central and Inner Asia"

Again, this is an over-glossed conclusion and archaeologically unsubstantiated statement. The modes of mobility evident in the steppe zones of the Caucasus around 3000 BC are completely different in a strategic sense than other documented pastoralists of the same time-period, and the "dynamic modes of interaction and connectivity" that emerged in the later BA had little if nothing to do with the Yamnaya mobility in the 3rd mill. BC., or only in the broadest deep time historical sense.

Please refer to our comment on the archaeological and bioarchaeological evidence for the referred processes above. We believe the critique may be based on a misunderstanding of our genetic intermediate groups as a cohesive cultural or temporal entity. We address this in the comment above and in lines 180-186.

The formation of the 3rd mill. BCE Yamnaya pastoralist strategy DOES appear to have been important for Europe in the immediate afterglow, but this is far less so in the Eurasian

steppe. The subsequent re-integration of admixed Yamnaya ancestry (admixed with Corded ware) does coincide with some technology changes, extending to roughly the Urals, ca. 2100 BCE. But this is 800-1000 years later. Beyond the Urals, the genetic percentages of admixed Yamnaya ancestry among 2nd mill BC Eurasian steppe groups is far less than the authors document in the Caucasus themselves and the populations east of the Urals appear far more integrated with local populations to the east and the south. This is made clear in Narasimhan et al, where over a span of ca. 1000 years of multi-regional dynamics there is admixed 10-40% Western Steppe ancestry to the east, hitting a max geographic range by ca. 1000 BC.

Of note, we do not contradict this finding. Our study documents the spread of a lifestyle facilitated by technological innovations. The scope of our manuscript is the western start of these processes and not its eastern consequences. Again, we fully agree that there have certainly been more pathways along which western domesticates and pastoral techniques were transferred to the East than by Yamnaya groups alone.

The old model of a wide arrow striking across the steppe and changing the economy, society and genomics is simply not scientifically evident – neither in the 3rd mill. BC with the initial Yamnaya migration, nor in the 2nd mill. BC. With the prolonged (and multidirectional) interactions of Bronze Age steppe groups. (It is sadly ironic, however, that Narasimhan et al. still felt compelled to draw such arrows, in spite of their own abundant genomic evidence that the process was much more intricate and multidirectional).

We agree.

****This point has implications for the present article in so far as the authors make this paper about the formation of the communities that set the stage for transformative economies and societies of the Eurasian steppe and more. To be clear – I don't necessarily think this gesturing to the "sources of Eurasian pastoralism" throughout the article is particularly necessary. The well-documented social matrix that emerges from WITHIN the study zone is itself of major importance and worthy of publication.

However, if the authors are dedicated to making this point a central one, they must not rely on overly simplified narratives. Rather, please do get into the weeds of the argument or at least make the point that the field is much weedier than these decades' old models could reveal.

Please refer to our responses above. We do not pursue out-dated models or unilinear, genetically underpinned dispersal scenarios. On the contrary, in line 657-658 we explicitly and cautiously formulate: "The Western Eurasian steppe pastoralist groups of this period, best represented by the Yamnaya culture...". This leaves it open as to who the actors in the processes were. The extended timeline alone makes it clear that these groups can at best be responsible for a limited transfer of innovations during a limited time window. The region itself is the origin of the specifically mobile forms of dairy pastoralism and associated technology that later spread via multiple vectors throughout the steppe and beyond. This subsistence strategy did not begin with the Yamnaya, but rather developed in the North Caucasus region during the millennium after the Eneolithic introduction of animal husbandry from agricultural populations south of the Caucasus. By the Early Bronze Age Maykop period, mobile dairy pastoralism was practised by nearly all populations in the North

Caucasus steppe zones, and it continued to be the dominant subsistence strategy of all studied groups (including the Yamnaya) in the North Caucasus up until the region's abandonment ca. 1700 BC. Mobile dairy pastoralism was a transformative technology that had a major and enduring impact on Eurasian steppe populations that continued well into the Soviet period throughout Central and Inner Asia. Consequently, we feel it is important to discuss this technology in the context of our study populations.

Line 646: What does sustainable, permanent and self-supporting mean in this context? Sustainable and permanent where? If the idea is that Yamnaya economy was resilient, then one might question why in the span of only 250 years, the basis of Yamnaya economy was fundamentally changed in almost all other regions where they migrated? E.g. 3000-2700 BCE in Europe, it was immediately integrated with farming. In the Altai, it was quickly transformed in terms of local ecology and seasonal mobility strategy, novel animal herd composition, and novel use of grains...By 2200 BCE whatever legacy of Yamnaya pastoralism was left was significantly transformed and adapted to local conditions and economic transformations (see point above about high regional diversity of herd composition and integrations (or not) of farming, as well as the complexity of admixtures that underly later Bronze Age Eurasian steppe). Please take care to reduce hyperbole and adhere to the archaeological evidence.

Sustainable, permanent and self-supporting in this context is argued from the evidence of dairy product consumption (Scott et al. 2022) and the finding that the isotopic signals of the steppe and foothill/mountain groups indicate two clearly defined and separate economic areas in different ecological habitats (Knipper et al. 2020). The first suggests a sustainable and renewable food base, the second shows that the utilisation of the steppe zone was permanent and not seasonal (as e.g. assumed by Shishlina 2008). The first date to the late-5th millennium BC, the second to the mid-4th millennium BC. The Yamnaya groups adopted this existing 'steppe' economy and, as the increasing number of animal species in the dairy products shows, adapted it during the 3rd millennium BC.

That the adaptation of pastoral economies varied in each region and required/underwent local adaptation is not in doubt. For Western Siberia and Central Asia, the zoological analyses by Kosincev 2003 for example argue for extremely variable and local livestock breeding strategies up to the 2nd millennium BC. Benecke (2017) argues similarly on the admittedly poorly dated archaeozoological material.

The points being made in this article about the formation of the Yamnaya as a genetic population are important, but the notion that the Yamnaya somehow transformed Central Eurasia are not backed by evidence. Whatever legacy of Yamnaya genetics that did underpin Eurasian populations came at or after 2000 BCE, and this was a protracted process of admixture taking place over a 1000+ years after their formation.

We would like to point out again that the Yamnaya groups are not at the centre of this article. We believe this critique may be more relevant for the two above-mentioned manuscripts available in preprint that focus more specifically on the Yamnaya and their influence outside the North Caucasus region.

To this reader, the point made about the complex, multi-regional admixtures across the Caucasus around 4700-3500 BCE is itself a huge contribution, but the argument gets far less granular when it is linked to general narratives about the spread of pastoralism

eastward across the steppe. At very least, the authors should engage with alternative models or take the opportunity to establish a line where the data presented provides clear insights and across which, the data does not offer concrete model testing.

The risk, as I see it, is linking data and solid genomics to broader processes that are only distantly related in archaeological terms, without delving into a thorough discussion of the actual data amassed now about the diversity of Bronze Age Eurasian pastoralism and regional ecologies (e.g. Bendrey 2011, Haruda 2019?).

The comment on the extensive data on the economy in the central and eastern steppe and the neighbouring areas in the south and north is definitely correct. However, we only mention the dynamics to the east twice in our text, in lines 130-134 and 662-663, and it is not at the centre of our discussion. Here the reviewer overinterprets our text in a direction that we do not intend.

Lines 682-692:

I do expect that the complex genomic admixtures within local and regional contexts helped innovate economic and social conditions and certainly impacted later populations – and those in the Caucasus are important. However, to argue that the admixture dynamics of the 5-3rd mill. Caucasus was the sole engine behind the transformation of economies and societies of the whole Eurasian steppe, is wildly overstated. Even in articles restricted by length, we should not fall into simplified conclusions without considering the breadth and diversity of both data and alternative explanatory models.

I highly support this article for publication (after considering these points) but I think that ultimately, the broader impacts of the 'prehistoric genomics of the Caucasus' offer a chance to evolve beyond the narrative of the "source" of all things Eurasian. At present, the data (both genomic and archaeological) do not directly support that conclusion in more than the broadest sense, but do allow for more nuanced and equally interesting narratives to emerge.

Thank you!

M. Frachetti

Thank you. It was a pleasure to respond to the many critical points. This is one of the advantages of the peer review process.

Referee #3 (Remarks to the Author):

Reviewer's report on "The rise and transformation of Bronze Age pastoralists in the Caucasus"

There are 78 new radiocarbon dates in this paper, and as the primary publication of these data the paper should include all the information specified in the conventions for reporting radiocarbon determinations (Millard 2014).

No methods are described or cited for the production of the dates. No quality control measures, such as C/N ratios, %C, %N, or yields, are reported nor whether they were used

to reject dates that are not published. $\delta^{13}\text{C}$ and $\delta^{15}\text{N}$ ratios are also routinely reported by radiocarbon labs and can be useful in identifying potential reservoir effects, but are not reported here. The quality of these dates cannot be judged unless all this information is included.

We agree and have updated our manuscript to now include all measures routinely reported by radiocarbon labs to the “Supplementary Table A. General Information”. These include $\delta^{13}\text{AMS}$ ‰, C/N ratio, C ‰, and collagen ‰. The recommended thresholds were used to include the new dates presented in this study. In the Supplementary Information we also provide additional information about the AMS dating laboratory that we used for radiocarbon dating.

In addition, we added new radiocarbon dates that were not yet available at the time of the first submission, and we used these to replace contextual dates.

There are 139 radiocarbon dates reported in Supp Table A, but it is not clear which 78 of them are new. Nor is this clear in the text. For example, OxA and Hd dates must be from previous studies but no citations are given for them.

We added a column indicating which radiocarbon dates are newly reported in this study, and we provided references for published dates in Supplementary Table A. These publications are also now referenced in the catalogue of archaeological complexes and site descriptions in the Supplementary Information Text.

Why report “95.4% (2a) calibration interval and median value” (Supp. p.4, Supp Table A) but plot mean (Figure 1b)?

Thank you for spotting this. It should read median in the latter case, as it was in fact plotted as the median. We corrected the figure legend accordingly.

And in text give “1a” and “2a”? It should be made clear that the OxCal ‘Whole range’ option has been used rather than reporting all the separate sub-ranges. The ranges given are 68.3% and 95.4% ranges not 1a and 2a as the probability distributions are not normally distributed (see Millard 2014).

Many thanks for the comment. We have expanded the information in the Supplementary Information Text and adjusted the reported data accordingly. The link to the data ranges in Supplementary Table A is also noted in this section.

Dates reported in the text should be rounded out appropriately, probably to the next 10 years. The OxCal ‘Round by’ option will do this.

We considered this point but decided to keep the dates in the text exactly as reported in Supplementary Table A, so that they are consistent with the original radiocarbon lab reports.

Supp p.6 “Modelling of radiocarbon dates from these graves result in a chronological sequence with a transitional period where both shaft and catacomb graves were used. This confirms earlier observations from neighbouring mounds²⁹ (Extended Data Fig. 8a).” However, fig 8a does not show any such chronological model, and in the cited reference I could only find mention of chronological modelling in the context of Rasshevatskiy 1, Russia.

You are correct. The complete radiocarbon data sequences for the two mounds are currently being prepared for a separate publication. We have therefore adapted our text accordingly and changed the reference to Reinhold et al. 2017, where the first radiocarbon dates from Marinskaya 5 (unmodeled) are discussed:

“Radiocarbon dates from these graves result in a chronological sequence with a transitional period where both shaft and catacomb graves were used. This confirms earlier observations in these mounds (Reinhold et al. 2017) (Extended Data Fig. 8a).”

Supp p.13 -14 calibrated age ranges are given without specifying the probability or σ .

We have updated the text for radiocarbon data and revised the reported data, avoiding the 1- and 2-sigma definitions.

Supp p. 14 “Grave 7 is a late Yamnaya grave interned into the first mounds hell” the final word must be wrong. ‘interned’ should be ‘interred’.

Thank you for spotting these typos. It should indeed read: “interred into the first mound shell”.

Supp p. 16 Lab-code ГИИ: all lab codes should be in Latin letters. See https://radiocarbon.org/sites/default/files/2023-01/Labs-2023_01_11.pdf

Supp p. 18 interned → interred

Supp p. 22 ‘35272 σ ’ missing a space

Thanks. All such instances in the Supplementary Information Text and Supplementary Table A are now corrected.

Supp p.21 has Lab-Id 11111111: (including the highlight). The date of 3810 BP does not appear in Table A where the dating for this sample is said to be contextual.

After consultation with the excavators of this site, we have deleted the date. It does not correspond to the dates from the more recent excavations, although the inventories are very similar.

Supp p.28: ‘fayence(?)’: is faience meant?

Supp p. 29 ‘40662 σ ’ missing a space

Thanks. Both typos are now corrected.

p.37-40 Marinskaya 5: I don’t understand dendrochronological dates with \pm . They do not follow a normal distribution, but are usually reported as a range when a sapwood estimate has been added to the last surviving ring.

No sapwood was preserved in this case. The date is based on a floating chronology using wiggle matching as described in Kantorovich et al. 2013.

p.39 MK5009 “dendro-dated 2644+13 BC” : + should be \pm

Thanks, this is now corrected.

p.45 “This assemblage was not in atomic order” should be “This assemblage was not in anatomical order”

Thank you. This is now corrected.

p. 47 SUS005 Context information should just be See above. The rest is repeated from above.

p.51 "half-desert" should be "semi-desert"

Thanks. Both typos are now corrected.

Table A

Three radiocarbon dated samples are missing the material: A19269.73.74, A19272, I2055

Thanks, these are now corrected.

Why do lab numbers for dates appear in parentheses?

There was no particular reason. We have now removed the

parentheses. Lab numbers (KIA-?) and (DAI) seem to be incomplete.

Thank you. The KIA ID has been added. DAI dendrochronological recordings do not have a specific ID as they are based on several series of measurements with internal numbers.

Table A, Date_Note: some of these are transparent, others obscure such as 'organics', 'hb', 'both highlighted', 'Remodeling for Marinskaya. Presumably 'hb' is human bone as in the text, but animal bones are not indicated.

Presumably 'source tissue' refers to the DNA extraction. Should we presume that the radiocarbon date came from the same sample unless indicated in Date_Note?

Yes, source tissue in column Z refers to the sample for DNA extraction as stated in the header. We added a column indicating the material dated as accurately as possible.

The shortcuts are the same as those used in Supplementary Information Text.

HD-29619

This is a date published in Wang et al. 2019. The dated material comes from the wooden burial chamber, the wood could not be determined. It is now included in Supplementary Table A.

Referee #4 (Remarks to the Author):

The manuscript "The rise and transformation of Bronze Age pastoralists in the Caucasus" represents an archaeogenomic study from Caucasus. Authors have generated new genome-wide data for 131 ancient genomes (102 out of them were suitable for the analyses) spanning 6000 years, focusing on Bronze Age. New data analyzed together with published genomes from Caucasus and adjacent regions and/or relevant time periods allowed detailing the demographic processes in the region at the onset, during and decline of the Bronze Age.

The conclusions of the manuscript are based on the new data and are original. The new data contributes to our understanding of the Steppe ancestry formation which had a strong impact on genetic landscape of human populations across Eurasia. However, the study in

the current format might rather be of significant interest for those specifically interested in the Caucasus prehistory: demographic events are analyzed within sub-periods (an array of overlapping periods of Eneolith-Bronza Age), there are numerous region-specific details, archaeological sites, outlier information etc. that might be difficult to follow for a wider audience.

Methodological section of the manuscript indicates that ancient DNA extraction/processing and contamination assessment followed requirements to work with highly degraded biomolecules hence suggest that data generated is reliable. Authors use established in human population genetics and archaeogenomics methods based on allele-frequencies (PCA, Admixture, f-statistics, qpAdm) and haplotype-based analyses (hapRoH).

I do not have any major comments, below are some minor things addressing them might improve the manuscript.

We thank referee #4 for the positive feedback.

- Authors use Steppe and Caucasus clusters when describing the genetic patterns they observe within different periods they single out. It might be useful to specifically indicate who is part of which cluster according to them for each period as “clustering” based on PCA might be subjective and people tend to see different clusters.

We indicate this in Extended Data Figure 2 and in Supplementary Table A. General Information, column D. We also would like to point the reviewer to Extended Data Figure 1 where data from all time periods are presented on the same plots, and in which the two main point clouds are clearly visible. Of note, we use these two cluster terms mainly as descriptive entities, while each chrono-cultural group is characterised individually.

- Readv2 – 3rd degree relationship should be treated with cautious.

We are aware of this and always cross-check using different methods. For example, with BREADR (<https://github.com/jonotuke/BREADR>) it is possible to test whether the observed PMR would be consistent with the expected distribution for third degree relatives (or any other degree) using a binomial test.

- Line 117 – Khyvalynsk (typo?)

Thank you. This is now fixed.

- Lines 440-450: why Lola ancestry is modelled using Steppe_Maykop individuals (it is rejected, and final model included Catacomb). From archaeological perspective, Catacomb would rather be the first choice for modelling genetic ancestry in Lola.

We usually approach the ancestry modelling via qpadm from a neutral genetic perspective, i.e., based on first-round qualitative observations from e.g. PCA or ADMIXTURE analysis. Here, Lola-associated individuals fall very close to Steppe Maykop individuals and therefore we first tested whether the preceding Steppe Maykop would be supported as a single source despite the chronological gap in our dataset. This single source model is rejected, as is a single-source model with Catacomb. As we show in the manuscript, Lola-associated ancestry can be modelled successfully as a two-source mix with preceding Catacomb as local source and additional ancestry from Steppe Maykop and/or Kazakhstan Kumsay EBA.

The latter two provide fitting sources for the excess ANE-related ancestry observed in Lola individuals.

Figures

Figure 1: would benefit if the color codes between the panels a and b matched; the red color of the new genomes merges with the mountain color.

We agree and have revised Figure 1. We had experimented with ways to use the same symbols as used in the chronology in panel b as well as in the other figures. The problem is that many sites/kurgans contain individuals from different chrono-cultural periods, which made the resulting figure far too busy. than it currently is. We thus opted to use pie charts for each site and use site code labels which are consistent with the individual IDs throughout the manuscript.

Figure 2: the Sankey diagrams look like a good choice for visualization of the ancestry modelling results.

Thank you, much appreciated.

Reviewer Reports on the First Revision:

Referee #1 (Remarks to the Author):

I am satisfied with the answers that the authors have made on all the points that I had made and those that the other reviewers made. I, thus, recommend the manuscript for publication.

Referee #2 (Remarks to the Author):

I have carefully read the revised manuscript and feel the authors have comprehensively addressed the questions and points raised by all reviewers. I fully recommend this paper for publication and congratulate all the authors on an impressive collaboration and multi-faceted study that will serve as a benchmark in Eurasian prehistory and genomics for many years to come.

Referee #3 (Remarks to the Author):

The authors have systematically dealt with all the issues that I raised and I have no further comments.

Referee #4 (Remarks to the Author):

No additional comments.